# Optimal Time Complexities of Parallel Stochastic Optimization Methods Under a Fixed Computation Model

**Alexander Tyurin**
KAUST
Saudi Arabia
alexandertiurin@gmail.com

**Peter Richtárik**
KAUST
Saudi Arabia
richtarik@gmail.com

## Abstract

Parallelization is a popular strategy for improving the performance of iterative algorithms. Optimization methods are no exception: design of efficient parallel optimization methods and tight analysis of their theoretical properties are important research endeavors. While the minimax complexities are well known for sequential optimization methods, the theory of parallel optimization methods is less explored. In this paper, we propose a new protocol that generalizes the classical oracle framework approach. Using this protocol, we establish *minimax complexities for parallel optimization methods* that have access to an unbiased stochastic gradient oracle with bounded variance. We consider a fixed computation model characterized by each worker requiring a fixed but worker-dependent time to calculate stochastic gradient. We prove lower bounds and develop optimal algorithms that attain them. Our results have surprising consequences for the literature of *asynchronous* optimization methods.

## 1 Introduction

We consider the nonconvex optimization problem

$$\min_{x \in Q} \left\{ f(x) := \mathbb{E}_{\xi \sim \mathcal{D}} \left[ f(x; \xi) \right] \right\}, \tag{1}$$

where $f : \mathbb{R}^d \times \mathbb{S}_\xi \to \mathbb{R}$, $Q \subseteq \mathbb{R}^d$, and $\xi$ is a random variable with some distribution $\mathcal{D}$ on $\mathbb{S}_\xi$. In machine learning, $\mathbb{S}_\xi$ could be the space of all possible data, $\mathcal{D}$ is the distribution of the training dataset, and $f(\cdot, \xi)$ is the loss of a data sample $\xi$. In this paper we address the following natural setup:

   (i) $n$ workers are available to work in parallel,

   (ii) the $i^{\text{th}}$ worker requires $\tau_i$ seconds[1] to calculate a stochastic gradient of $f$.

The function $f$ is $L$–smooth and lower-bounded (see Assumptions 7.1–7.2), and stochastic gradients are unbiased and $\sigma^2$-variance-bounded (see Assumption 7.3).

### 1.1 Classical theory

In the nonconvex setting, gradient descent (GD) is an optimal method with respect to the number of gradient ($\nabla f$) calls (Lan, 2020; Nesterov, 2018; Carmon et al., 2020) for finding an approximately stationary point of $f$. Obviously, a key issue with GD is that it requires access to the exact gradients

---

[1]Or any other unit of time.

37th Conference on Neural Information Processing Systems (NeurIPS 2023).

$\nabla f$ of the function $f$. However, in many practical applications, it can be infeasible to calculate the gradient of $\mathbb{E}\left[f(\cdot;\xi)\right]$ analytically. Moreover, even if this is possible, e.g., if the distribution $\mathcal{D}$ is described by $m$ possible samples, so that $\mathbb{E}_{\xi \sim \mathcal{D}}\left[f(\cdot;\xi)\right] = (1/m)\sum_{i=1}^{m} f(\cdot;\xi_i)$, $m$ can be huge (Krizhevsky et al., 2017), and gradient evaluation can be arbitrarily expensive.

**Stochastic Gradient Descent.** Due to the above-mentioned problem, machine learning literature is preoccupied with the study of algorithms that can work with stochastic gradients instead (Lan, 2020; Ghadimi and Lan, 2013). For all $x \in \mathbb{R}^d$, we assume that the $n$ workers have access to independent, unbiased, and $\sigma^2$-variance-bounded stochastic gradients $\widehat{\nabla} f(x, \xi)$ (see Assumption 7.3), where $\xi$ is a random sample from $\mathcal{D}$. Under such assumptions, **with one worker**, stochastic gradient descent (SGD), i.e., the method $x^{k+1} = x^k - \gamma\widehat{\nabla} f(x^k;\xi^k)$, where $\xi^k$ are i.i.d. random samples from $\mathcal{D}$, is known to be optimal with respect to the number of stochastic gradient calls (Ghadimi and Lan, 2013; Arjevani et al., 2022). SGD guarantees convergence to an $\varepsilon$–stationary point in expectation after $\mathrm{O}\left(L\Delta/\varepsilon + \sigma^2 L\Delta/\varepsilon^2\right)$ stochastic gradient evaluations, where $\Delta := f(x^0) - f^*$ and $x^0 \in \mathbb{R}^d$ is a starting point.

## 1.2 Parallel optimization methods

Using the bounds from Section 1.1, one can easily *estimate* the performance of these algorithms in real systems. For instance, if it takes $\tau_1$ *seconds* to calculate a stochastic gradient **with one worker**, then SGD guarantees to return a solution after

$$\mathrm{O}\left(\tau_1\left(\frac{L\Delta}{\varepsilon} + \frac{\sigma^2 L\Delta}{\varepsilon^2}\right)\right)$$

seconds. If instead of a single worker we can access $n$ workers that can calculate stochastic gradients in parallel, we can consider the following classical parallel methods:

**Minibatch SGD.** The minibatch SGD method (Minibatch SGD), i.e., the iterative process

$$x^{k+1} = x^k - \gamma\frac{1}{n}\sum_{i=1}^{n}\widehat{\nabla} f(x^k;\xi_i^k),$$

where $\gamma$ is a stepsize, $\xi_i^k$ are i.i.d. samples from $\mathcal{D}$, and the gradients $\widehat{\nabla} f(x^k;\xi_i^k)$ are calculated in parallel. This method converges after $\mathrm{O}\left(L\Delta/\varepsilon + \sigma^2 L\Delta/n\varepsilon^2\right)$ iterations (Cotter et al., 2011; Goyal et al., 2017; Gower et al., 2019) and after

$$\mathrm{O}\left(\tau_{\max}\left(\frac{L\Delta}{\varepsilon} + \frac{\sigma^2 L\Delta}{n\varepsilon^2}\right)\right) \tag{2}$$

seconds, where $\tau_{\max} := \max_{i \in [n]} \tau_i$ is the processing time associated with the *slowest* machine[2].

Although the time complexity (2) of Minibatch SGD improves with the number of workers $n$, in general, this does *not* guarantee better performance due to the delay $\tau_{\max}$. In real systems, parallel computations can be very chaotic, e.g., they can be slow due to inconsistent network communications, or GPU computation delays (Dutta et al., 2018; Chen et al., 2016).

**Asynchronous SGD.** We now consider the asynchronous SGD method (Asynchronous SGD) (Recht et al., 2011; Nguyen et al., 2018; Arjevani et al., 2020; Feyzmahdavian et al., 2016) described by

1. Receive $\widehat{\nabla} f(x^{k-\delta_k};\xi^{k-\delta_k})$ from a worker,
2. $x^{k+1} = x^k - \gamma^k\widehat{\nabla} f(x^{k-\delta_k};\xi^{k-\delta_k})$,
3. Ask the worker to calculate $\widehat{\nabla} f(x^{k+1};\xi^{k+1})$,

where $\xi^k$ are i.i.d. samples from $\mathcal{D}$, and $\delta_k$ are gradient iteration delays. This is an *asynchronous* method: the workers work independently, finish calculations of stochastic gradients with potentially large and chaotic delays $\delta_k$, and the result of their computation is applied as soon as it is ready, without having to wait for other workers. Asynchronous SGD was also considered in the heterogeneous setting (see details in Section A.2).

---

[2] Further, we assume that the last $n^{\text{th}}$ worker is the slowest one: $\tau_n = \tau_{\max}$.

Table 1: **Homogeneous and Heterogeneous Case.** The required time to get an $\varepsilon$-stationary point ($\mathbb{E}[\|\nabla f(\widehat{x})\|^2] \leq \varepsilon$) in the nonconvex setting, where $i^{\text{th}}$ worker requires $\tau_i$ seconds to calculate a stochastic gradient. We assume that $0 < \tau_1 \leq \cdots \leq \tau_n$.

| Homogeneous Case | | Heterogeneous Case | |
|---|---|---|---|
| **Method** | **Time Complexity** | **Method** | **Time Complexity** |
| Minibatch SGD | $\tau_n \left( \frac{L\Delta}{\varepsilon} + \frac{\sigma^2 L\Delta}{n\varepsilon^2} \right)$ | Minibatch SGD | $\tau_n \left( \frac{L\Delta}{\varepsilon} + \frac{\sigma^2 L\Delta}{n\varepsilon^2} \right)$ |
| Asynchronous SGD (Cohen et al., 2021) (Koloskova et al., 2022) (Mishchenko et al., 2022) | $\left( \frac{1}{n} \sum_{i=1}^{n} \frac{1}{\tau_i} \right)^{-1} \left( \frac{L\Delta}{\varepsilon} + \frac{\sigma^2 L\Delta}{n\varepsilon^2} \right)$ | Malenia SGD (Theorem A.4) | $\tau_n \frac{L\Delta}{\varepsilon} + \left( \frac{1}{n} \sum_{i=1}^{n} \tau_i \right) \frac{\sigma^2 L\Delta}{n\varepsilon^2}$ |
| Rennala SGD (Theorem 7.5) | $\min_{m \in [n]} \left[ \left( \frac{1}{m} \sum_{i=1}^{m} \frac{1}{\tau_i} \right)^{-1} \left( \frac{L\Delta}{\varepsilon} + \frac{\sigma^2 L\Delta}{m\varepsilon^2} \right) \right]$ | Lower Bound (Theorem A.2) | $\tau_n \frac{L\Delta}{\varepsilon} + \left( \frac{1}{n} \sum_{i=1}^{n} \tau_i \right) \frac{\sigma^2 L\Delta}{n\varepsilon^2}$ |
| Lower Bound (Theorem 6.4) | $\min_{m \in [n]} \left[ \left( \frac{1}{m} \sum_{i=1}^{m} \frac{1}{\tau_i} \right)^{-1} \left( \frac{L\Delta}{\varepsilon} + \frac{\sigma^2 L\Delta}{m\varepsilon^2} \right) \right]$ | | |

Cohen et al. (2021); Mishchenko et al. (2022); Koloskova et al. (2022) provide the current state-of-the-art analysis of Asynchronous SGD. In particular, they prove that Asynchronous SGD converges after $\mathrm{O}\left( nL\Delta/\varepsilon + \sigma^2 L\Delta/\varepsilon^2 \right)$ iterations. To show the superiority of Asynchronous SGD, Mishchenko et al. (2022) consider the following *fixed computation model*: the $i^{\text{th}}$ worker requires $\tau_i$ seconds to calculate stochastic gradients. In this setting, Asynchronous SGD converges after

$$\mathrm{O}\left( \left( \frac{1}{n} \sum_{i=1}^{n} \frac{1}{\tau_i} \right)^{-1} \left( \frac{L\Delta}{\varepsilon} + \frac{\sigma^2 L\Delta}{n\varepsilon^2} \right) \right) \tag{3}$$

seconds (we reprove this fact in Section L). Thus, Asynchronous SGD can be $(1/n) \sum_{i=1}^{n} \tau_{\max}/\tau_i$ times faster than Minibatch SGD.

Besides Asynchronous SGD, many other strategies utilize parallelization (Dutta et al., 2018; Woodworth et al., 2020; Wu et al., 2022), and can potentially improve over Minibatch SGD.

## 2 Problem and Contribution

In this paper, we seek to find the optimal time complexity in the setting from Section 1: our goal is to provide a lower bound and a method that attains it. Our main contributions are:

(i) **Lower bound.** In Sections 4 & 5, we define new Protocols 2 & 3, and a new complexity measure $\mathfrak{m}_{\text{time}}$ (see (6)), which we believe are more appropriate for the analysis of parallel optimization algorithms. In Section 6, we prove the time complexity lower bound for (nonconvex) functions and algorithms that work with parallel asynchronous oracles.

(i) **Optimal method.** In Section 7, we develop a minimax optimal method—Rennala[3] SGD—that attains this lower bound.

In addition, we investigate several other related questions. As an independent result, in Section 8 we prove that all methods which synchronize workers in each iteration (e.g., Minibatch SGD) have *provably* worse time complexity than asynchronous methods (e.g., Rennala SGD (see Method 4), Asynchronous SGD). In Section A, we extend our theory to the *heterogeneous* case, in which the workers have access to different distributions (datasets), and provide a lower bound and a new method that attains it. In Section B, we provide the optimal time complexities in the *convex* setting.

---

[3] https://eldenring.wiki.fextralife.com/Rennala+Queen+of+the+Full+Moon: Rennala, Queen of the Full Moon is a Legend Boss in Elden Ring. Though not a demigod, Rennala is one of the shardbearers who resides in the Academy of Raya Lucaria. Rennala is a powerful sorceress, head of the Carian Royal family, and erstwhile leader of the Academy.

# 3 Classical Oracle Protocol

Let us recall the classical approach to obtaining lower bounds for optimization algorithms. We need to define a *function class* $\mathcal{F}$, an *oracle class* $\mathcal{O}$, and an *algorithm class* $\mathcal{A}$. We then analyze the complexity of an algorithm $A = \{A^k\}_{k=0}^{\infty} \in \mathcal{A}$, using the following protocol:

---
**Protocol 1** Classical Oracle Protocol

---
1: **Input:** function $f \in \mathcal{F}$, oracle and distribution $(O, \mathcal{D}) \in \mathcal{O}(f)$, algorithm $A \in \mathcal{A}$
2: **for** $k = 0, \ldots, \infty$ **do**
3:    $x^k = A^k(g^1, \ldots, g^k)$                                                     $\triangleright \; x^0 = A^0$ for $k = 0$.
4:    $g^{k+1} = O(x^k, \xi^{k+1}), \quad \xi^{k+1} \sim \mathcal{D}$
5: **end for**

---

More formally, in first-order stochastic optimization, the oracle class $\mathcal{O}$ returns a random mapping $O : \mathbb{R}^d \times \mathbb{S}_\xi \to \mathbb{R}^d$ based on a function $f \in \mathcal{F}$ and a distribution $\mathcal{D}$; we use the notation $(O, \mathcal{D}) \in \mathcal{O}(f)$. An algorithm $A = \{A^k\}_{k=0}^{\infty} \in \mathcal{A}$ is a sequence such that

$$A^k : \underbrace{\mathbb{R}^d \times \cdots \times \mathbb{R}^d}_{k \text{ times}} \to \mathbb{R}^d \;\; \forall k \geq 1, \text{ and } A^0 \in \mathbb{R}^d. \tag{4}$$

Typically, an oracle $O$ returns an unbiased stochastic gradient that satisfies Assumption 7.3: $O(x, \xi) = \widehat{\nabla} f(x; \xi)$ for all $x \in \mathbb{R}^d$ and $\xi \in \mathbb{S}_\xi$. Let us fix an oracle class $\mathcal{O}$. Then, in the nonconvex first-order stochastic setting, we analyze the complexity measure

$$\mathfrak{m}_{\text{oracle}}(\mathcal{A}, \mathcal{F}) := \inf_{A \in \mathcal{A}} \sup_{f \in \mathcal{F}} \sup_{(O, \mathcal{D}) \in \mathcal{O}(f)} \inf \left\{ k \in \mathbb{N} \,\middle|\, \mathbb{E}\left[\left\|\nabla f(x^k)\right\|^2\right] \leq \varepsilon \right\}, \tag{5}$$

where the sequence $\{x^k\}_k$ is generated by Protocol 1. Virtually all previous works are concerned with lower bounds of optimization problems using Protocol 1 and the complexity measure (5) (Nemirovskij and Yudin, 1983; Carmon et al., 2020; Arjevani et al., 2022; Nesterov, 2018).

# 4 Time Oracle Protocol

In the previous sections, we discuss the classical approach to estimating the complexities of algorithms. Briefly, these approaches seek to quantify the worst-case number of iterations or oracle calls that are required to find a solution (see (5)), which is very natural for *sequential* methods. However, and this is a key observation of our work, this approach is not convenient if we want to analyze *parallel* methods. We now propose an alternative protocol that can be more helpful in this situation:

---
**Protocol 2** Time Oracle Protocol

---
1: **Input:** functions $f \in \mathcal{F}$, oracle and distribution $(O, \mathcal{D}) \in \mathcal{O}(f)$, algorithm $A \in \mathcal{A}$
2: $s^0 = 0$
3: **for** $k = 0, \ldots, \infty$ **do**
4:    $(t^{k+1}, x^k) = A^k(g^1, \ldots, g^k)$,                                        $\triangleright \; t^{k+1} \geq t^k$
5:    $(s^{k+1}, g^{k+1}) = O(t^{k+1}, x^k, s^k, \xi^{k+1}), \quad \xi^{k+1} \sim \mathcal{D}$
6: **end for**

---

Protocol 2 is almost identical to Protocol 1 except for one key detail: Protocol 2 requires the algorithms to return a sequence $\{t^{k+1}\}_{k=1}^{\infty}$ such that $t^{k+1} \geq t^k \geq 0$ for all $k \geq 0$. We assume that $t^0 = 0$. We also assume that the oracles take to the input the states $s^k$ and output them (the role of these states will be made clear later). In this case, we provide the following definition of an algorithm.

**Definition 4.1.** An algorithm $A = \{A^k\}_{k=0}^{\infty}$ is a sequence such that

$$A^k : \underbrace{\mathbb{R}^d \times \cdots \times \mathbb{R}^d}_{k \text{ times}} \to \mathbb{R}_{\geq 0} \times \mathbb{R}^d \quad \forall k \geq 1, A^0 \in \mathbb{R}_{\geq 0} \times \mathbb{R}^d,$$

and, for all $k \geq 1$ and $g^1, \ldots, g^k \in \mathbb{R}^d$, $t^{k+1} \geq t^k$, where $t^{k+1}$ and $t^k$ are defined as $(t^{k+1}, \cdot) = A^k(g^1, \ldots, g^k)$ and $(t^k, \cdot) = A^{k-1}(g^1, \ldots, g^{k-1})$.

Let us explain the role of the sequence $\{t^k\}_k$. In Protocol 1, an algorithm outputs a point $x^k$ and then asks the oracle: *Provide me a gradient at the point $x^k$*. In contrast, in Protocol 2 an algorithm outputs a point $x^k$ and a time $t^{k+1}$, and asks the oracle: *Start calculating a gradient at the point $x^k$ at a time $t^{k+1}$*. We have a constraint that $t^{k+1} \geq t^k$ for all $k \geq 0$, which means that the algorithm is not allowed to travel into the past.

Using Protocol 2, we propose to use another complexity measure instead of (5):

$$\mathfrak{m}_{\text{time}}(\mathcal{A}, \mathcal{F}) := \inf_{A \in \mathcal{A}} \sup_{f \in \mathcal{F}} \sup_{(O,\mathcal{D}) \in \mathcal{O}(f)} \inf \left\{ t \geq 0 \,\middle|\, \mathbb{E}\left[ \inf_{k \in S_t} \left\| \nabla f(x^k) \right\|^2 \right] \leq \varepsilon \right\},$$
$$S_t := \left\{ k \in \mathbb{N}_0 \middle| t^k \leq t \right\},$$
(6)

where the sequences $t^k$ and $x^k$ are generated by Protocol 2. In (5), we seek to find the *worst-case number of iterations $k$* required to get $\mathbb{E}\left[ \left\| \nabla f(x^k) \right\|^2 \right] \leq \varepsilon$ for any $A \in \mathcal{A}$. In (6), we seek to find the *worst-case case time $t$* required to find an $\varepsilon$-stationary point for any $A \in \mathcal{A}$.

We now provide an example, considering an oracle that calculates a stochastic gradient in $\tau$ seconds. Let us define the appropriate oracle for this problem:

$$O_\tau^{\widehat{\nabla} f} : \underbrace{\mathbb{R}_{\geq 0}}_{\text{time}} \times \underbrace{\mathbb{R}^d}_{\text{point}} \times \underbrace{(\mathbb{R}_{\geq 0} \times \mathbb{R}^d \times \{0,1\})}_{\text{input state}} \times \mathbb{S}_\xi \to \underbrace{(\mathbb{R}_{\geq 0} \times \mathbb{R}^d \times \{0,1\})}_{\text{output state}} \times \mathbb{R}^d$$

such that $O_\tau^{\widehat{\nabla} f}(t, x, (s_t, s_x, s_q), \xi) = \begin{cases} ((t, x, 1), & 0), & s_q = 0, \\ ((s_t, s_x, 1), & 0), & s_q = 1 \text{ and } t < s_t + \tau, \\ ((0, 0, 0), & \widehat{\nabla} f(s_x; \xi)), & s_q = 1 \text{ and } t \geq s_t + \tau, \end{cases}$ (7)

and $\widehat{\nabla} f$ is a mapping such that $\widehat{\nabla} f : \mathbb{R}^d \times \mathbb{S}_\xi \to \mathbb{R}^d$. Further, we additionally assume that $\widehat{\nabla} f$ is an unbiased $\sigma^2$-variance-bounded stochastic gradient (see Assumption 7.3).

Note that the oracle $O_\tau^{\widehat{\nabla} f}$ emulates the behavior of a real worker. Indeed, the oracle can return three different outputs. If $s_q = 0$, it means that the oracle has been idle, then "starts the calculation" of the gradient at the point $x$, and changes the state $s_q$ to 1. Also, using the state, it remembers the time moment $t$ when the calculation began and the point $x$. Next, if $s_q = 1$ and $t < s_t + \tau$, it means the oracle is still calculating the gradient, so if an algorithm sends time $t$ such that $t < s_t + \tau$, then it receives the zero vector. Finally, if $s_q = 1$, as soon as an algorithm sends time $t$ such that $t \geq s_t + \tau$, then the oracle will be ready to provide the gradient. Note that the oracle provides the gradient calculated at the point $x$ that was requested when the oracle was idle. Thus, the time between the request of an algorithm to get the gradient and the time when the algorithm gets the gradient is at least $\tau$ seconds.

In Protocol 2, we have a game between an algorithm $A \in \mathcal{A}$ and an oracle class $\mathcal{O}$, where algorithms can decide the sequence of times $t^k$. Thus, an algorithm wants to find enough information from an oracle as soon as possible to obtain $\varepsilon$–stationary point.

Let us consider an example. For the oracle class $\mathcal{O}$ that generates the oracle from (7), we can define the SGD method in the following way. We take any starting point $x^0 \in \mathbb{R}^d$, a step size $\gamma = \min\{1/L, \varepsilon/2L\sigma^2\}$ (see Theorem D.8) and define $A^k : \underbrace{(\mathbb{R}^d \times \cdots \times \mathbb{R}^d)}_{k \text{ times}} \to \mathbb{R}_{\geq 0} \times \mathbb{R}^d$ such that

$$A^k(g^1, \ldots, g^k) = \begin{cases} \left( \tau \lfloor k/2 \rfloor, x^0 - \gamma \sum_{j=1}^k g^k \right), & k \pmod 2 = 0, \\ \left( \tau (\lfloor k/2 \rfloor + 1), 0 \right), & k \pmod 2 = 1, \end{cases}$$
(8)

for all $k \geq 1$, and $A^0 = (0, x^0)$. Let us explain the behavior of the algorithm. In the first step of Protocol 2, when $k = 0$, the algorithm requests the gradient at the point $x^0$ at the time $t^1 = 0$ since

$A^0 = (0, x^0)$. The oracle $O$ changes the state from $s_q^0 = 0$ to $s_q^1 = 1$ and remembers the point $x^0$ in the state $s_x^1$. In the second step of the protocol, when $k = 1$, the algorithm calls the oracle at the time $\tau(\lfloor k/2 \rfloor + 1) = \tau$. In the oracle, the condition $t^2 \geq s_t^1 + \tau \Leftrightarrow \tau \geq 0 + \tau$ is satisfied, and it returns the gradient at the point $x^0$. Note that this can only happen if an algorithm does the second call at a time that is greater or equal to $\tau$.

One can see that after $\tau K$ seconds, the algorithm returns the point $x^{2K} = x^0 - \gamma \sum_{j=0}^{K-1} \widehat{\nabla} f(x^{2j}; \xi^{2j+1})$, where $\xi^j \sim \mathcal{D}$ are i.i.d. random variables. The algorithm is equivalent to the SGD method that converges after $K = O\left(L\Delta/\varepsilon + \sigma^2 L\Delta/\varepsilon^2\right)$ steps for the function class $\mathcal{F}_{\Delta,L}$ (see Definition 6.1) for $x^0 = 0$. Thus, the complexity $\mathfrak{m}_{\text{time}}\left(\{A\}, \mathcal{F}_{\Delta,L}\right)$ equals $O\left(\tau \times \left(L\Delta/\varepsilon + \sigma^2 L\Delta/\varepsilon^2\right)\right)$.

Actually, any algorithm that was designed for Protocol 1 can be used in Protocol 2 with the oracle (7). Assuming that we have mappings $A^k : \mathbb{R}^d \times \cdots \times \mathbb{R}^d \to \mathbb{R}^d$ for all $k \geq 1$, we can define mappings $\widehat{A}^k : \mathbb{R}^d \times \cdots \times \mathbb{R}^d \to \mathbb{R}_{\geq 0} \times \mathbb{R}^d$ via

$$\widehat{A}^k(g^1, \ldots, g^k) = \begin{cases} \left(\tau \lfloor k/2 \rfloor, A^{\lfloor k/2 \rfloor}(g^2, g^4, \ldots, g^{2k})\right), & k \pmod 2 = 0, \\ \left(\tau\left(\lfloor k/2 \rfloor + 1\right), 0\right), & k \pmod 2 = 1. \end{cases}$$

For $k = 0$, we define $\widehat{A}^0 = (0, A^0)$.

# 5   Time Multiple Oracles Protocol

The protocol framework from the previous section does not seem to be very powerful because one can easily find the time complexity (6) by knowing (5) and the amount of time that oracle needs to calculate a gradient. In fact, we provide Protocol 2 for simplicity only. We now consider a protocol that works with multiple oracles:

---

**Protocol 3** Time Multiple Oracles Protocol

---

1: **Input:** function(s) $f \in \mathcal{F}$, oracles and distributions $((O_1, ..., O_n), (\mathcal{D}_1, ..., \mathcal{D}_n)) \in \mathcal{O}(f)$, algorithm $A \in \mathcal{A}$
2: $s_i^0 = 0$ for all $i \in [n]$
3: **for** $k = 0, \ldots, \infty$ **do**
4:     $(t^{k+1}, i^{k+1}, x^k) = A^k(g^1, \ldots, g^k)$,                              $\triangleright\ t^{k+1} \geq t^k$
5:     $(s_{i^{k+1}}^{k+1}, g^{k+1}) = O_{i^{k+1}}(t^{k+1}, x^k, s_{i^{k+1}}^k, \xi^{k+1})$,     $\xi^{k+1} \sim \mathcal{D}_{i^{k+1}}$    $\triangleright\ s_j^{k+1} = s_j^k\ \ \forall j \neq i^{k+1}$
6: **end for**

---

Compared to Protocol 2, Protocol 3 works with multiple oracles, and algorithms return the indices $i^{k+1}$ of the oracle they want to call. This minor add-on to the protocol enables the possibility of analyzing parallel optimization methods. Also, each oracle $O_i$ can have its own distribution $\mathcal{D}_i$.

Let us consider an example with two oracles $O_1 = O_{\tau_1}^{\widehat{\nabla} f}$ and $O_2 = O_{\tau_2}^{\widehat{\nabla} f}$ from (7). One can see that a "wise" algorithm will first call the oracle $O_1$ with the time $t^0 = 0$, and then, in the second step, it will call the oracle $O_2$ also with the time $t^1 = 0$. Note that it is impossible to do the following steps: in the first step an algorithm calls the oracle $O_1$ with the time $t^0 = 0$, in the second step, the algorithm calls the oracle $O_1$ with the time $t^1 = \tau_1$ and receives the gradient, in the third step, the algorithm calls the oracle $O_2$ with the time $t^2 = 0$. Indeed, this can't happen because $t^2 < t^1$.

An example of a "non-wise" algorithm is an algorithm that, in the first step, calls the oracle $O_1$ with the time $t^0 = 0$. In the second step, the algorithm calls the oracle $O_1$ with the time $t^1 = \tau_1$ and receives the gradient. In the third step, the algorithm calls the oracle $O_2$ with the time $t^2 = \tau_1$. It would mean that the "non-wise" algorithm did not use the oracle $O_2$ for $\tau_1$ seconds. Consequently, the "wise" algorithm can receive two gradients after $\max\{\tau_1, \tau_2\}$ seconds, while the "non-wise" algorithm can only receive two gradients after $\tau_1 + \tau_2$ seconds.

We believe that Protocol 3 and the complexity (6) is a better choice for analyzing the complexities of parallel methods than the classical Protocol 1. In the next section, we will use Protocol 3 to obtain lower bounds for parallel optimization methods.

# 6 Lower Bound for Parallel Optimization Methods

Considering Protocol 3, we define a special function class $\mathcal{F}$, oracle class $\mathcal{O}$, and algorithm class $\mathcal{A}$. We consider the same function class as Nesterov (2018); Arjevani et al. (2022); Carmon et al. (2020):

**Definition 6.1** (Function Class $\mathcal{F}_{\Delta,L}$)**.** We assume that function $f : \mathbb{R}^d \to \mathbb{R}$ is differentiable, $L$-smooth, i.e., $\|\nabla f(x) - \nabla f(y)\| \leq L \|x - y\| \quad \forall x, y \in \mathbb{R}^d$, and $\Delta$-bounded, i.e., $f(0) - \inf_{x \in \mathbb{R}^d} f(x) \leq \Delta$. A set of all functions with such properties we denote by $\mathcal{F}_{\Delta,L}$.

In this paper, we analyze the class of "zero-respecting" algorithms, defined next.

**Definition 6.2** (Algorithm Class $\mathcal{A}_{\mathrm{zr}}$)**.** Let us consider Protocol 3. We say that an algorithm $A$ from Definition 4.1 is a zero-respecting algorithm, if $\mathrm{supp}\left(x^k\right) \subseteq \bigcup_{j=1}^{k} \mathrm{supp}\left(g^j\right)$ for all $k \in \mathbb{N}_0$, where $\mathrm{supp}(x) := \{i \in [d] \,|\, x_i \neq 0\}$. A set of all algorithms with this property we define as $\mathcal{A}_{\mathrm{zr}}$.

A zero-respecting algorithm does not try to change the coordinates for which no information was received from oracles. This family is considered by Arjevani et al. (2022); Carmon et al. (2020), and includes SGD, Minibatch and Asynchronous SGD, and Adam (Kingma and Ba, 2014).

**Definition 6.3** (Oracle Class $\mathcal{O}_{\tau_1,\ldots,\tau_n}^{\sigma^2}$)**.** Let us consider an oracle class such that, for any $f \in \mathcal{F}_{\Delta,L}$, it returns oracles $O_i = O_{\tau_i}^{\widehat{\nabla} f}$ and distributions $\mathcal{D}_i$ for all $i \in [n]$, where $\widehat{\nabla} f$ is an unbiased $\sigma^2$-variance-bounded mapping (see Assumption 7.3). The oracles $O_{\tau_i}^{\widehat{\nabla} f}$ are defined in (7). We define such oracle class as $\mathcal{O}_{\tau_1,\ldots,\tau_n}^{\sigma^2}$. Without loss of generality, we assume that $0 < \tau_1 \leq \cdots \leq \tau_n$.

We take $\mathcal{O}_{\tau_1,\ldots,\tau_n}^{\sigma^2}$ because it emulates the behavior of workers in real systems, where workers can have different processing times (delays) $\tau_i$. Note that $\mathcal{O}_{\tau_1,\ldots,\tau_n}^{\sigma^2}$ has the freedom to choose a mapping $\widehat{\nabla} f$. We only assume that the mapping is unbiased and $\sigma^2$-variance-bounded. We are now ready to present our first result; a lower bound:

**Theorem 6.4.** *Let us consider the oracle class $\mathcal{O}_{\tau_1,\ldots,\tau_n}^{\sigma^2}$ for some $\sigma^2 > 0$ and $0 < \tau_1 \leq \cdots \leq \tau_n$. We fix any $L, \Delta > 0$ and $0 < \varepsilon \leq c'L\Delta$. In view Protocol 3, for any algorithm $A \in \mathcal{A}_{\mathrm{zr}}$, there exists a function $f \in \mathcal{F}_{\Delta,L}$ and oracles and distributions $((O_1, \ldots, O_n), (\mathcal{D}_1, \ldots, \mathcal{D}_n)) \in \mathcal{O}_{\tau_1,\ldots,\tau_n}^{\sigma^2}(f)$ such that $\mathbb{E}\left[\inf_{k \in S_t} \left\|\nabla f(x^k)\right\|^2\right] > \varepsilon$, where $S_t := \left\{k \in \mathbb{N}_0 \big| t^k \leq t\right\}$, and*

$$t = c \times \min_{m \in [n]} \left[\left(\frac{1}{m}\sum_{i=1}^{m}\frac{1}{\tau_i}\right)^{-1}\left(\frac{L\Delta}{\varepsilon} + \frac{\sigma^2 L\Delta}{m\varepsilon^2}\right)\right].$$

*The quantities $c'$ and $c$ are universal constants.*

Theorem 6.4 states that

$$\mathfrak{m}_{\mathrm{time}}\left(\mathcal{A}_{\mathrm{zr}}, \mathcal{F}_{\Delta,L}\right) = \Omega\left(\min_{m \in [n]}\left[\left(\frac{1}{m}\sum_{i=1}^{m}\frac{1}{\tau_i}\right)^{-1}\left(\frac{L\Delta}{\varepsilon} + \frac{\sigma^2 L\Delta}{m\varepsilon^2}\right)\right]\right). \tag{9}$$

The interpretation behind this complexity will be discussed later in Section 7.3. No algorithms known to us attain (9). For instance, Asynchronous SGD has the time complexity (3). Let us assume that $\sigma^2/\varepsilon \leq p$ and $p \in [n]$. Then (lower bound from (9)) $= \mathrm{O}\left(\left(\frac{1}{p}\sum_{i=1}^{p}\frac{1}{\tau_i}\right)^{-1}\left(\frac{L\Delta}{\varepsilon}\right)\right)$. In this case, the lower bound in (9) will be at least $\left(\frac{1}{n}\sum_{i=1}^{n}\frac{1}{\tau_i}\right)^{-1}/\left(\frac{1}{p}\sum_{i=1}^{p}\frac{1}{\tau_i}\right)^{-1}$ times smaller. It means that either the obtained lower bound is not tight, or Asynchronous SGD is a suboptimal method. In the following section we provide a method that attains the lower bound. The obtained lower bound is valid even if an algorithm has the freedom to interrupt oracles. See details in Section F.

## 6.1 Related work

For convex problems, Woodworth et al. (2018) proposed the graph oracle, which generalizes the classical gradient oracle (Nemirovskij and Yudin, 1983; Nesterov, 2018), and provided lower bounds

for a rather general family of parallel methods. Arjevani et al. (2020) analyzed the delayed gradient descent method, which is Asynchronous SGD when all iteration delays $\delta_k = \delta$ are a constant.

As far as we know, Woodworth et al. (2018) provide the most suitable and tightest prior framework for analyzing lower bound complexities for problem (1). However, as we shall see, our framework us more powerful. Moreover, they only consider the convex case. In Section M, we use the framework of Woodworth et al. (2018) and analyze the fixed computation model, where $i^{\text{th}}$ worker requires $\tau_i$ seconds to calculate stochastic gradients. In Section B, we consider the convex setting and show that the lower bound obtained by their framework is not tight and can be improved. While the graph oracle framework by Woodworth et al. (2018) is related to the classical oracle protocol (Section 3) and also calculates the number of oracle calls in order to get lower bounds, our approach directly estimates the required time. For more details, see Section B and the discussion in Section B.1.1.

---

**Method 4** Rennala SGD

1: **Input:** starting point $x^0$, stepsize $\gamma$, batch size $S$
2: Run Method 5 in all workers
3: **for** $k = 0, 1, \ldots, K - 1$ **do**
4:     Init $g^k = 0$ and $s = 1$
5:     **while** $s \leq S$ **do**
6:         Wait for the next worker
7:         Receive gradient and iteration index $(g, k')$
8:         **if** $k' = k$ **then**
9:             $g^k = g^k + \frac{1}{S} g; \quad s = s + 1$
10:         **end if**
11:         Send $(x^k, k)$ to the worker
12:     **end while**
13:     $x^{k+1} = x^k - \gamma g^k$
14: **end for**

**Method 5** Worker's Infinite Loop

1: Init $g = 0$ and $k' = -1$
2: **while** True **do**
3:     Send $(g, k')$ to the server
4:     Receive $(x^k, k)$ from the server
5:     $k' = k$
6:     $g = \widehat{\nabla} f(x^k; \xi), \quad \xi \sim \mathcal{D}$
7: **end while**

---

## 7 Minimax Optimal Method

We now propose and analyze a new method: Rennala SGD (see Method 4). Methods with a similar structure were proposed previously (e.g., (Dutta et al., 2018)), but we are not aware of any method with precisely the same parameters and structure. For us, in this paper, the theoretical bounds are more important than the method itself.

Let us briefly describe the structure of the method. At the start, Method 4 asks all workers to run Method 5. Method 5 is a standard routine: the workers receive points $x^k$ from the server, calculate stochastic gradients, and send them back to the server. Besides that, the workers receive and send the iteration counter $k$ of the received points $x^k$. At the server's side, in each iteration $k$, Method 4 calculates $g^k$ and performs the standard gradient-type step $x^{k+1} = x^k - \gamma g^k$. The calculation of $g^k$ is done in a loop. The server waits for the workers to receive a stochastic gradient and an iteration index. The most important part of the method is that the server ignores a stochastic gradient if its iteration index is not equal to the current iteration index. In fact, this means that $g^k = (1/s) \sum_{i=1}^{S} \widehat{\nabla} f(x^k; \xi_i)$, where $\xi_i$ are i.i.d. samples. In other words, the server ignores all stochastic gradients that were calculated at the points $x^0, \cdots, x^{k-1}$.

It may seem that Method 4 does not fully use the information due to ignoring some stochastic gradients. That contradicts the philosophy of Asynchronous SGD, which tries to use all stochastic gradients calculated in the previous points. Nevertheless, we show that Rennala SGD has *better time complexity* than Asynchronous SGD, and this complexity matches the lower bound from Theorem 6.4. The fact that Rennala SGD ignores the previous iterates is motivated by the proof of the lower bound in Section 6. In the proof, any algorithm, on the constructed "worst case" function, does not progress to a stationary point if it calculates a stochastic gradient at a non-relevant point. This suggested to us to construct a method that would focus all workers on the last iterate.

### 7.1 Assumptions

Let us consider the following assumptions.

**Assumption 7.1.** $f$ is differentiable & $L$–smooth, i.e., $\|\nabla f(x) - \nabla f(y)\| \leq L \|x - y\|, \forall x, y \in \mathbb{R}^d$.

**Assumption 7.2.** There exist $f^* \in \mathbb{R}$ such that $f(x) \geq f^*$ for all $x \in \mathbb{R}^d$.

**Assumption 7.3.** For all $x \in \mathbb{R}^d$, stochastic gradients $\widehat{\nabla} f(x; \xi)$ are unbiased and $\sigma^2$-variance-bounded, i.e., $\mathbb{E}_\xi \left[ \widehat{\nabla} f(x; \xi) \right] = \nabla f(x)$ and $\mathbb{E}_\xi \left[ \left\| \widehat{\nabla} f(x; \xi) - \nabla f(x) \right\|^2 \right] \leq \sigma^2$, where $\sigma^2 \geq 0$.

## 7.2 Analysis of Rennala SGD

**Theorem 7.4.** *Assume that Assumptions 7.1, 7.2 and 7.3 hold. Let us take the batch size* $S = \max \left\{ \lceil \sigma^2/\varepsilon \rceil, 1 \right\}$, *and* $\gamma = \min \left\{ \frac{1}{L}, \frac{\varepsilon S}{2L\sigma^2} \right\} = \Theta(1/L)$ *in Method 4. Then after*

$$K \geq \frac{24\Delta L}{\varepsilon}$$

*iterations, the method guarantees that* $\frac{1}{K} \sum_{k=0}^{K-1} \mathbb{E} \left[ \left\| \nabla f(x^k) \right\|^2 \right] \leq \varepsilon.$

In the following theorem, we provide the time complexity of Method 4.

**Theorem 7.5.** *Consider Theorem 7.4. We assume that* $i^{th}$ *worker returns a stochastic gradient every* $\tau_i$ *seconds for all* $i \in [n]$. *Without loss of generality, we assume that* $0 < \tau_1 \leq \cdots \leq \tau_n$. *Then after*

$$96 \times \min_{m \in [n]} \left[ \left( \frac{1}{m} \sum_{i=1}^m \frac{1}{\tau_i} \right)^{-1} \left( \frac{L\Delta}{\varepsilon} + \frac{\sigma^2 L\Delta}{m\varepsilon^2} \right) \right] \tag{10}$$

*seconds, Method 4 guarantees to find an* $\varepsilon$-*stationary point.*

This result with Theorem 6.4 state that

$$\mathfrak{m}_{\text{time}} \left( \mathcal{A}_{\text{zr}}, \mathcal{F}_{\Delta, L} \right) = \Theta \left( \min_{m \in [n]} \left[ \left( \frac{1}{m} \sum_{i=1}^m \frac{1}{\tau_i} \right)^{-1} \left( \frac{L\Delta}{\varepsilon} + \frac{\sigma^2 L\Delta}{m\varepsilon^2} \right) \right] \right) \tag{11}$$

for Protocol 3 and and the oracle class $\mathcal{O}_{\tau_1, \ldots, \tau_n}^{\sigma^2}$ from Definition 6.3.

## 7.3 Discussion

Theorem 7.5 and Theorem 6.4 state that Method 4 is *minimax optimal* under the assumption that the delays of the workers are fixed and equal to $\tau_i$. Note that this assumption is required only in Theorem 7.5, and Theorem 7.4 holds without it.

In the same setup, the previous works (Cohen et al., 2021; Mishchenko et al., 2022; Koloskova et al., 2022) obtained the weaker time complexity (3). We do not rule out that it might be possible for the analysis, the parameters or the structure of Asynchronous SGD to be improved and obtain the optimal time complexity (10). We leave this to future work. However, instead, we developed Method 4 that has not only the optimal time complexity, but also a very simple structure and analysis (see Section D.4.1). Our claims are supported by experiments in Section J.

The reader can see that we provide the complexity in a nonconstructive way, as the minimization over the parameter $m \in [n]$. Note that Method 4 *automatically finds the optimal* $m$ in (7.5), and it does not require the knowledge of the delays $\tau_i$ to do so! Let us explain the intuition behind the complexity (10). Let $m^*$ be the optimal parameter of (10) with the smallest index. In Section D.4.3, we show that all workers with the delays $\tau_i$ for all $i > m^*$ can be simply ignored since their delays are too large, and their inclusion would only harm the convergence time of the method. So, the method *automatically* ignores them! However, in Asynchronous SGD, these harmful workers can contribute to the optimization process, which can be the reason for the suboptimality of Asynchronous SGD.

In general, there are two important regimes: $\sigma^2/\varepsilon \ll n$ ("low noise/large # of workers") and $\sigma^2/\varepsilon \gg n$ ("high noise/small # of workers"). Intuitively, in the "high noise/small # of workers" regime, (11) is minimized when $m$ is close to $n$. However, in the "low noise/large # of workers", the optimal $m$ can be much smaller than $n$.

# 8 Synchronized Start of Workers

In the previous sections, we obtain the time complexities for the case when the workers asynchronously compute stochastic gradients. It is important that the complexities are obtained assuming that the workers *can start* their calculations asynchronously. However, in practice, it is common to train machine learning models with multiple workers/GPUs, so that all workers are *synchronized* after each stochastic gradient calculation (Goyal et al., 2017; Sergeev and Balso, 2018). The simplest example of such a strategy is Minibatch SGD (see Section 1.2). We want to find an answer to the question: what is the best time complexity we can get if we assume that the workers start simultaneously? In Section G, we formalize this setting, and show that the time complexity equals to

$$\mathfrak{m}_{\text{time}}\left(\mathcal{A}_{\text{zr}}, \mathcal{F}_{\Delta,L}\right) = \Theta\left(\min_{m \in [n]}\left[\tau_m\left(\frac{L\Delta}{\varepsilon} + \frac{\sigma^2 L\Delta}{m\varepsilon^2}\right)\right]\right) \tag{12}$$

for Protocol 2 and the oracle class $\mathcal{O}_{\tau_1,\dots,\tau_n}^{\sigma^2,\text{sync}}$ from Definition G.1. Comparing (11) and (12), one can see that *methods that start the calculations of workers simultaneously are provably worse than methods that allow workers to start the calculations asynchronously.*

# 9 Future Work

In this work, we consider the setup where the times $\tau_i$ are fixed. In future work, one can consider natural, important, and more general scenarios where they can be random, follow some distribution, and/or depend on the random variables $\xi$ from Assumption 7.3 (be correlated with stochastic gradients).

### Acknowledgements

This work of P. Richtárik and A. Tyurin was supported by the KAUST Baseline Research Scheme (KAUST BRF) and the KAUST Extreme Computing Research Center (KAUST ECRC), and the work of P. Richtárik was supported by the SDAIA-KAUST Center of Excellence in Data Science and Artificial Intelligence (SDAIA-KAUST AI).

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

# Contents

Table 2: **Heterogeneous Case.** The required time to get an $\varepsilon$-stationary point in the nonconvex setting, where $i^{\text{th}}$ worker requires $\tau_i$ seconds to calculate a stochastic gradient. We assume that $0 < \tau_1 \leq \cdots \leq \tau_n$.

| Heterogeneous Case | |
| --- | --- |
| **Method** | **Time Complexity** |
| Minibatch SGD | $\tau_n \left( \frac{L\Delta}{\varepsilon} + \frac{\sigma^2 L\Delta}{n\varepsilon^2} \right)$ |
| Malenia SGD (Theorem A.4) | $\tau_n \frac{L\Delta}{\varepsilon} + \left( \frac{1}{n} \sum_{i=1}^n \tau_i \right) \frac{\sigma^2 L\Delta}{n\varepsilon^2}$ |
| Lower Bound (Theorem A.2) | $\tau_n \frac{L\Delta}{\varepsilon} + \left( \frac{1}{n} \sum_{i=1}^n \tau_i \right) \frac{\sigma^2 L\Delta}{n\varepsilon^2}$ |

## A   Heterogeneous Regime

Up to this point, we discussed the regime when all workers calculate i.i.d. stochastic gradients. In distributed optimization and federated learning (Konečný et al., 2016), it can be possible that the workers hold different datasets. Let us consider the following optimization problem:

$$\min_{x \in \mathbb{R}^d} \left\{ f(x) := \frac{1}{n} \sum_{i=1}^n \mathbb{E}_{\xi_i \sim \mathcal{D}_i} \left[ f_i(x; \xi_i) \right] \right\}, \tag{13}$$

where $f_i : \mathbb{R}^d \times \mathbb{S}_{\xi_i} \to \mathbb{R}^d$ and $\xi_i$ are random variables with some distributions $\mathcal{D}_i$ on $\mathbb{S}_{\xi_i}$. Problem (13) generalizes problem (1). Here we have the same goals as in the previous sections. We want to obtain the minimax complexities for the case when the workers contain different datasets.

### A.1   Lower bound

For the heterogeneous case, we modify Definition 6.3:

**Definition A.1** (Oracle Class $\mathcal{O}_{\tau_1,\ldots,\tau_n}^{\sigma^2,heterog}$)**.**
Let us consider an oracle class such that, for any $f = \frac{1}{n} \sum_{i=1}^n f_i \in \mathcal{F}_{\Delta,L}$, it returns oracles $O_i = O_{\tau_i}^{\widehat{\nabla} f_i}$ and distributions $\mathcal{D}_i$ for all $i \in [n]$, where $\widehat{\nabla} f_i$ is an unbiased $\sigma^2$-variance-bounded mapping for all $i \in [n]$ (see Assumption 7.3). The oracles $O_{\tau_i}^{\widehat{\nabla} f_i}$ are defined in (7). We define such oracle class as $\mathcal{O}_{\tau_1,\ldots,\tau_n}^{\sigma^2,heterog}$. Without loss of generality, we assume that $0 < \tau_1 \leq \cdots \leq \tau_n$.

**Theorem A.2.** *Let us consider the oracle class $\mathcal{O}_{\tau_1,\ldots,\tau_n}^{\sigma^2,heterog}$ for some $\sigma^2 > 0$ and $0 < \tau_1 \leq \cdots \leq \tau_n$. We fix any $L, \Delta > 0$ and $0 < \varepsilon \leq c' L\Delta$. In the view Protocol 3, for any algorithm $A \in \mathcal{A}_{\text{zr}}$, there exists a function $f = \frac{1}{n} \sum_{i=1}^n f_i \in \mathcal{F}_{\Delta,L}$ and oracles and distributions $((O_1, \ldots, O_n), (\mathcal{D}_1, \ldots, \mathcal{D}_n)) \in \mathcal{O}_{\tau_1,\ldots,\tau_n}^{\sigma^2,heterog}(f_1, \ldots, f_n)$ such that $\mathbb{E}\left[ \inf_{k \in S_t} \left\| \nabla f(x^k) \right\|^2 \right] > \varepsilon$, where $S_t := \left\{ k \in \mathbb{N}_0 \big| t^k \leq t \right\}$, and*

$$t = c \times \left( \tau_n \frac{L\Delta}{\varepsilon} + \left( \frac{1}{n} \sum_{i=1}^n \tau_i \right) \frac{\sigma^2 L\Delta}{n\varepsilon^2} \right). \tag{14}$$

*The quantity $c'$ and $c$ are universal constants.*

Theorem A.2 states that

$$\mathfrak{m}_{\text{time}} \left( \mathcal{A}_{\text{zr}}, \mathcal{F}_{\Delta,L} \right) = \Omega \left( \tau_n \frac{L\Delta}{\varepsilon} + \left( \frac{1}{n} \sum_{i=1}^n \tau_i \right) \frac{\sigma^2 L\Delta}{n\varepsilon^2} \right).$$

One can see that the lower bound for the heterogeneous case is larger than (9). In Section A.3, we provide a method that attains the lower bound.

## A.2 Related work and discussion

The optimization problem (13) is well-investigated by many papers, including (Aytekin et al., 2016; Mishchenko et al., 2018; Nguyen et al., 2022; Wu et al., 2022; Koloskova et al., 2022; Mishchenko et al., 2022). There were attempts to analyze Asynchronous SGD in the heterogeneous regime. For instance, Mishchenko et al. (2022) proved the convergence to a neighborhood of a solution only. In general, it is quite challenging to get good rates for Asynchronous SGD without additional assumptions about the similarity of the functions $f_i$ (Koloskova et al., 2022; Mishchenko et al., 2022).

In the non-stochastic case, when $\sigma^2 = 0$, Wu et al. (2022) analyzed the PIAG method in the non-stochastic heterogeneous regime and showed convergence. Although the performance of PIAG can be good in practice, in the worst case PIAG requires $O\left(\tau_n \widehat{L}\Delta/\varepsilon\right)$ seconds to converge, where $\tau_n$ is the time delay of the slowest worker, $\widehat{L} := \sqrt{\sum_{i=1}^n L_i^2}$, and $L_i$ is a Lipschitz constant of $\nabla f_i$. Note that the synchronous Minibatch SGD (see Section 1.2) method has the complexity $O\left(\tau_n L\Delta/\varepsilon\right)$, which is always better.[4]

Our lower bound in Theorem A.2 does not leave hope of breaking the dependence on the worst straggler in the heterogeneous case. In the stochastic case, the lower bound is slightly more optimistic in the regimes when the statistical term (the second term in (14)) is large. If the stragglers do not have too large delays, then their contributions to the *arithmetic* mean can be small. Note that in Theorem 6.4 in the homogeneous case, we have the *harmonic* mean of the delays instead.

## A.3 Minimax optimal method

In this section, we provide Malenia[5] SGD (see Method 6) that is slightly different from Rennala SGD (Method 4). There are two main differences: the first one is that Method 6 has different gradients estimators $g_i^k$ for each worker, and the second one is the constraint $\left(\frac{1}{n}\sum_{i=1}^n 1/B_i\right)^{-1} < S/n$ in the inner loop[6]. The more gradients we get from the workers, the larger the term $\left(\frac{1}{n}\sum_{i=1}^n 1/B_i\right)^{-1}$.

---

**Method 6** Malenia SGD

1: **Input:** starting point $x^0$, stepsize $\gamma$, parameter $S$
2: Run Method 7 in all workers
3: **for** $k = 0, 1, \ldots, K-1$ **do**
4:     Init $g_i^k = 0$ and $B_i = 0$
5:     **while** $\left(\frac{1}{n}\sum_{i=1}^n \frac{1}{B_i}\right)^{-1} < \frac{S}{n}$ **do**
6:         Wait for the next worker
7:         Receive gradient, iteration index, worker's index $(g, k', i)$
8:         **if** $k' = k$ **then**
9:             $g_i^k = g_i^k + g$
10:             $B_i = B_i + 1$
11:         **end if**
12:         Send $(x^k, k)$ to the worker
13:     **end while**
14:     $g^k = \frac{1}{n}\sum_{i=1}^n \frac{1}{B_i} g_i^k$
15:     $x^{k+1} = x^k - \gamma g^k$
16: **end for**

---

As in Section 7.2, we can provide the convergence theorems.

**Theorem A.3.** *Assume that Assumptions 7.1 and 7.2 hold for the function $f$. Assumption 7.3 holds for the function $f_i$ for all $i \in [n]$. Let us take the parameter $S = \max\left\{\left\lceil \sigma^2/\varepsilon\right\rceil, n\right\}$, and*

---

[4]In the nonconvex case, $\widehat{L}$ can be arbitrarily larger than $L$.

[5]https://eldenring.wiki.fextralife.com/Malenia+Blade+of+Miquella: Malenia, Blade of Miquella and Malenia, Goddess of Rot is two-phase a Demigod Boss in Elden Ring. She's the twin of Miquella, the most powerful of the Empyreans, and gained renown for her legendary battle against Starscourge Radahn during the Shattering, in which she unleashed the power of the Scarlet Rot and reduced Caelid to ruins.

[6]We assume that $\left(\frac{1}{n}\sum_{i=1}^n 1/B_i\right)^{-1} = 0$ if exists $i \in [n]$ such that $B_i = 0$.

**Method 7** Worker's Infinite Loop
___
1: Init $g = 0$, $k' = -1$, and worker's index $i$
2: **while** True **do**
3:     Send $(g, k', i)$ to the server
4:     Receive $(x^k, k)$ from the server
5:     $k' = k$
6:     $g = \widehat{\nabla} f_i(x^k; \xi), \quad \xi \sim \mathcal{D}$
7: **end while**
___

$\gamma = \min \left\{ \frac{1}{L}, \frac{\varepsilon S}{2L\sigma^2} \right\} = \Theta\left( 1/L \right)$ *in Method 6, then after* $K \geq {24\Delta L}/{\varepsilon}$ *iterations the method guarantees that* $\frac{1}{K} \sum_{k=0}^{K-1} \mathbb{E}\left[ \left\| \nabla f(x^k) \right\|^2 \right] \leq \varepsilon$.

**Theorem A.4.** *Let us consider Theorem A.3. We assume that $i^{th}$ worker returns a stochastic gradient every $\tau_i$ seconds for all $i \in [n]$. Without loss of generality, we assume that $0 < \tau_1 \leq \cdots \leq \tau_n$. Then after*

$$96 \left( \tau_n \frac{L\Delta}{\varepsilon} + \left( \frac{1}{n} \sum_{i=1}^{n} \tau_i \right) \frac{\sigma^2 L\Delta}{n\varepsilon^2} \right) \tag{15}$$

*seconds, Method 6 guarantees to find an $\varepsilon$-stationary point.*

Comparing Theorem A.2 and Theorem A.4, one can see that the complexity (15) is optimal. Note that Theorem A.3 holds without assumptions that the delays $\tau_i$ are fixed.

### A.4 Discussion

Unlike Asynchronous SGD and PIAG, Malenia SGD ignores all stochastic gradients that were calculated in the previous iterations, which appears to be counterproductive. Nevertheless, we show that Malenia SGD converges, and the time complexity is optimal with respect to all parameters. Note that Malenia SGD does not require the Lipschitz smoothness of the local functions $f_i$, does not depend on the time delays $\tau_i$, does not need any similarity assumptions about the functions $f_i$, and can be applied to problems where the function is not Lipschitz (does not have bounded gradients). The analysis of the method is elementary and does not go far away from the theory of the classical SGD method. When the ratio $\sigma^2/\varepsilon$ is large, Malenia SGD is better than Minibatch SGD (see (2)) by $\Theta\left( \tau_n / \left( \frac{1}{n} \sum_{i=1}^{n} \tau_i \right) \right)$ times.

## B Convex Case

### B.1 Lower Bound

Let us consider the optimization problem (1) in the case when the function $f$ is convex. For the convex case, using Protocol 2, we propose to use another complexity measure instead of (6):

$$\mathfrak{m}_{\text{time}} (\mathcal{A}, \mathcal{F}) := \inf_{A \in \mathcal{A}} \sup_{f \in \mathcal{F}} \sup_{(O, \mathcal{D}) \in \mathcal{O}(f)} \inf \left\{ t \geq 0 \,\middle|\, \mathbb{E}\left[ \inf_{k \in S_t} f(x^k) \right] - \inf_{x \in Q} f(x) \leq \varepsilon \right\},$$
$$S_t := \left\{ k \in \mathbb{N}_0 \,\middle|\, t^k \leq t \right\}, \tag{16}$$

where the sequences $t^k$ and $x^k$ are generated by Protocol 2. Let us consider the following class of convex functions:

**Definition B.1** (Function Class $\mathcal{F}_{R,M,L}^{\text{conv}}$)**.**
Let us define $B_2(0, R) := \left\{ x \in \mathbb{R}^d \,\middle|\, \|x\| \leq R \right\}$. We assume that a function $f : \mathbb{R}^d \to \mathbb{R}$ is convex, differentiable, $L$-smooth on the set $B_2(0, R)$, i.e.,

$$\|\nabla f(x) - \nabla f(y)\| \leq L \|x - y\| \quad \forall x, y \in B_2(0, R),$$

and $M$-Lipschitz on the set $B_2(0, R)$, i.e.,

$$|f(x) - f(y)| \leq M \|x - y\| \quad \forall x, y \in B_2(0, R).$$

A set of all functions with such properties we define as $\mathcal{F}_{R,M,L}^{\text{conv}}$.

Table 3: **Convex Homogeneous Case.** The required time to get an $\varepsilon$-solution in the convex setting, where $i^{\text{th}}$ worker requires $\tau_i$ seconds to calculate a stochastic gradient. We assume that $0 < \tau_1 \leq \cdots \leq \tau_n$.

| Method | Time Complexity |
|---|---|
| Minibatch SGD | $\tau_n \left( \min \left\{ \frac{\sqrt{L}R}{\sqrt{\varepsilon}}, \frac{M^2 R^2}{\varepsilon^2} \right\} + \frac{\sigma^2 R^2}{n\varepsilon^2} \right)$ |
| Asynchronous SGD (Mishchenko et al., 2022) | $\left( \frac{1}{n} \sum_{i=1}^{n} \frac{1}{\tau_i} \right)^{-1} \left( \frac{LR^2}{\varepsilon} + \frac{\sigma^2 R^2}{n\varepsilon^2} \right)$ |
| (Accelerated) Rennala SGD (Theorems B.9 and B.11) | $\min_{m \in [n]} \left[ \left( \frac{1}{m} \sum_{i=1}^{m} \frac{1}{\tau_i} \right)^{-1} \left( \min \left\{ \frac{\sqrt{L}R}{\sqrt{\varepsilon}}, \frac{M^2 R^2}{\varepsilon^2} \right\} + \frac{\sigma^2 R^2}{m\varepsilon^2} \right) \right]$ |
| Lower Bound (Theorem B.4) | $\min_{m \in [n]} \left[ \left( \frac{1}{m} \sum_{i=1}^{m} \frac{1}{\tau_i} \right)^{-1} \left( \min \left\{ \frac{\sqrt{L}R}{\sqrt{\varepsilon}}, \frac{M^2 R^2}{\varepsilon^2} \right\} + \frac{\sigma^2 R^2}{m\varepsilon^2} \right) \right]$ |
| Lower Bound (Section M) (Woodworth et al., 2018) | $\tau_1 \min \left\{ \frac{\sqrt{L}R}{\sqrt{\varepsilon}}, \frac{M^2 R^2}{\varepsilon^2} \right\} + \left( \frac{1}{n} \sum_{i=1}^{n} \frac{1}{\tau_i} \right)^{-1} \frac{\sigma^2 R^2}{n\varepsilon^2}$ |

For the convex case, we analyze the following class of algorithms:

**Definition B.2** (Algorithm Class $\mathcal{A}_{\text{zr}}^R$)**.**
Let us consider Protocol 3. We say that an algorithm $A$ from Definition 4.1 belongs to a class $\mathcal{A}_{\text{zr}}^R$ iff $A \in \mathcal{A}_{\text{zr}}$ and $x^k \in B_2(0, R)$ for all $k \geq 0$.

We also define an oracle class:

**Definition B.3** (Oracle Class $\mathcal{O}_{\tau_1,\ldots,\tau_n}^{\text{conv},\sigma^2}$)**.**
Let us consider an oracle class such that, for any $f \in \mathcal{F}_{R,M,L}^{\text{conv}}$, it returns oracles $O_i = O_{\tau_i}^{\widehat{\nabla} f}$ and distributions $\mathcal{D}_i$ for all $i \in [n]$, where $\widehat{\nabla} f$ is an unbiased $\sigma^2$-variance-bounded mapping on the set $B_2(0, R)$. The oracles $O_{\tau_i}^{\widehat{\nabla} f}$ are defined in (7). We define such oracle class as $\mathcal{O}_{\tau_1,\ldots,\tau_n}^{\text{conv},\sigma^2}$. Without loss of generality, we assume that $0 < \tau_1 \leq \cdots \leq \tau_n$.

For this setup, we provide the lower bound for the class of convex functions in the next theorem.

**Theorem B.4.** *Let us consider the oracle class $\mathcal{O}_{\tau_1,\ldots,\tau_n}^{\text{conv},\sigma^2}$ for some $\sigma^2 > 0$ and $0 < \tau_1 \leq \cdots \leq \tau_n$. We fix any $R, L, M, \varepsilon > 0$ such that $\sqrt{L}R > c_1 \sqrt{\varepsilon} > 0$ and $M^2 R^2 > c_2 \varepsilon^2$. In the view Protocol 3, for any algorithm $A \in \mathcal{A}_{\text{zr}}^R$, there exists a function $f \in \mathcal{F}_{R,M,L}^{\text{conv}}$ and oracles and distributions $((O_1, \ldots, O_n), (\mathcal{D}_1, \ldots, \mathcal{D}_n)) \in \mathcal{O}_{\tau_1,\ldots,\tau_n}^{\text{conv},\sigma^2}(f)$ such that*

$$\mathbb{E}\left[ \inf_{k \in S_t} f(x^k) \right] - \inf_{x \in B_2(0,R)} f(x) > \varepsilon,$$

*where $S_t := \left\{ k \in \mathbb{N}_0 \big| t^k \leq t \right\}$, and*

$$t = c \times \min_{m \in [n]} \left[ \left( \frac{1}{m} \sum_{i=1}^{m} \frac{1}{\tau_i} \right)^{-1} \left( \min \left\{ \frac{\sqrt{L}R}{\sqrt{\varepsilon}}, \frac{M^2 R^2}{\varepsilon^2} \right\} + \frac{\sigma^2 R^2}{m\varepsilon^2} \right) \right].$$

*The quantities $c_1$, $c_2$ and $c$ are universal constants.*

### B.1.1   Discussion

We improve the lower bound obtained by (Woodworth et al., 2018) (see Table 3). Woodworth et al. (2018) try to reduce any optimization problem to an oracle graph. Then, they get a lower bound using the depth and the number of nodes in a graph. Our approach is different, as we directly estimate the required time and avoid the reduction to an oracle graph. One can think that our "oracle graph" is always linear in Protocol 3, but every node in an "oracle graph" is associated with a timestamp and an index. Unlike the oracle in (Woodworth et al., 2018), which always returns a stochastic gradient, our oracle (7) returns a stochastic gradient only if the conditions are satisfied. Also, Woodworth et al. (2018) construct different "worst case" functions and oracles for the "optimization" and "statistical" terms. While our construction consists only of one function and one oracle.

## B.2 Minimax optimal method

### B.2.1 Assumptions

Additionally to some assumptions from Section 7.1, we use the following assumptions in the convex case.

**Assumption B.5.** The function $f$ is convex and attains the minimum at some point $x^* \in \mathbb{R}^d$.

**Assumption B.6.** The function $f$ is $M$–Lipschitz, i.e.,

$$|f(x) - f(y)| \leq M \|x - y\|, \quad \forall x, y \in \mathbb{R}^d.$$

**Assumption B.7.** For all $x \in \mathbb{R}^d$, stochastic gradients $\widehat{\nabla} f(x; \xi)$ are unbiased and have $\sigma^2$-variance-bounded, i.e., $\mathbb{E}_{\xi \sim \mathcal{D}} \left[ \widehat{\nabla} f(x; \xi) \right] \in \partial f(x)$ and $\mathbb{E}_{\xi \sim \mathcal{D}} \left[ \left\| \widehat{\nabla} f(x; \xi) - \mathbb{E} \left[ \widehat{\nabla} f(x; \xi) \right] \right\|^2 \right] \leq \sigma^2$, where $\sigma^2 \geq 0$.

### B.2.2 Analysis of Rennala SGD and Accelerated Rennala SGD in convex case

**Theorem B.8.** *Assume that Assumptions B.5, B.6 and B.7 hold. Let us take the batch size* $S = \max \left\{ \lceil \sigma^2/M^2 \rceil, 1 \right\}$, *and* $\gamma = \frac{\varepsilon}{M^2 + \sigma^2/S} = \Theta(\varepsilon/M^2)$ *in Method 4, then after* $K \geq 2M^2R^2/\varepsilon^2$ *iterations the method guarantees that* $\mathbb{E}\left[ f(\widehat{x}^K) \right] - f(x^*) \leq \varepsilon$, *where* $\widehat{x}^K = \frac{1}{K} \sum_{k=0}^{K-1} x^k$ *and* $R = \|x^* - x^0\|$.

**Theorem B.9.** *Let us consider Theorem B.8. We assume that $i^{th}$ worker returns a stochastic gradient every $\tau_i$ seconds for all $i \in [n]$. Without loss of generality, we assume that $0 < \tau_1 \leq \cdots \leq \tau_n$. Then after*

$$8 \min_{m \in [n]} \left[ \left( \frac{1}{m} \sum_{i=1}^{m} \frac{1}{\tau_i} \right)^{-1} \left( \frac{M^2 R^2}{\varepsilon^2} + \frac{\sigma^2 R^2}{m \varepsilon^2} \right) \right] \tag{17}$$

*seconds Method 4 guarantees to find an $\varepsilon$-solution.*

Let us provide the theorems in the *smooth* convex case. We consider the accelerated version of Rennala SGD. In particular, we assume that instead of Line 13 in Method 4, we have

$$
\begin{aligned}
&\gamma_{k+1} = \gamma(k+1), \quad \alpha_{k+1} = 2/(k+2) \\
&y^{k+1} = (1 - \alpha_{k+1})x^k + \alpha_{k+1}u^k, \qquad (u^0 = x^0) \\
&u^{k+1} = u^k - \gamma_{k+1}g_k, \\
&x^{k+1} = (1 - \alpha_{k+1})x^k + \alpha_{k+1}u^{k+1}.
\end{aligned}
\tag{18}
$$

We refer to such method as Accelerated Method 4 or Accelerated Rennala SGD. The acceleration technique is based on (Lan, 2020).

**Theorem B.10.** *Assume that Assumptions B.5, 7.1 and 7.3 hold. Let us take the batch size* $S = \max \left\{ \left\lceil (\sigma^2 R)/(\varepsilon^{3/2}\sqrt{L}) \right\rceil, 1 \right\}$, *and* $\gamma = \min \left\{ \frac{1}{4L}, \left[ \frac{3R^2 S}{4\sigma^2(K+1)(K+2)^2} \right]^{1/2} \right\}$ *in Accelerated Method 4, then after* $K \geq \frac{8\sqrt{L}R}{\sqrt{\varepsilon}}$ *iterations the method guarantees that* $\mathbb{E}\left[ f(x^K) \right] - f(x^*) \leq \varepsilon$, *where* $R \geq \|x^* - x^0\|$.

**Theorem B.11.** *Let us consider Theorem B.10. We assume that $i^{th}$ worker returns a stochastic gradient every $\tau_i$ seconds for all $i \in [n]$. Without loss of generality, we assume that $0 < \tau_1 \leq \cdots \leq \tau_n$. Then after*

$$32 \min_{m \in [n]} \left[ \left( \frac{1}{m} \sum_{i=1}^{m} \frac{1}{\tau_i} \right)^{-1} \left( \frac{\sqrt{L}R}{\sqrt{\varepsilon}} + \frac{\sigma^2 R^2}{m \varepsilon^2} \right) \right]$$

*seconds Accelerated Method 4 guarantees to find an $\varepsilon$-solution.*

# C  Table of Notations

| Notation | Meaning |
|---|---|
| $g = \mathrm{O}(f)$ | Exist $C > 0$ such that $g(z) \le C \times f(z)$ for all $z \in \mathcal{Z}$ |
| $g = \Omega(f)$ | Exist $C' > 0$ such that $g(z) \ge C' \times f(z)$ for all $z \in \mathcal{Z}$ |
| $g = \Theta(f)$ | $g = \mathrm{O}(f)$ and $g = \Omega(f)$ |
| $\{a, \dots, b\}$ | Set $\{i \in \mathbb{Z} \mid a \le i \le b\}$ |
| $[n]$ | $\{1, \dots, n\}$ |

# D  Proofs for Homogeneous Regime

## D.1  The "worst case" function

In this section, we recall the "worst case" function that we use to prove our lower bounds. This is the standard function that is used in nonconvex optimization. Let us define

$$\mathrm{prog}(x) := \max\{i \ge 0 \mid x_i \ne 0\} \quad (x_0 \equiv 1).$$

In our proofs, we use the construction from (Carmon et al., 2020; Arjevani et al., 2022). For any $T \in \mathbb{N}$, the authors define

$$F_T(x) := -\Psi(1)\Phi(x_1) + \sum_{i=2}^{T} \left[\Psi(-x_{i-1})\Phi(-x_i) - \Psi(x_{i-1})\Phi(x_i)\right], \tag{19}$$

where

$$\Psi(x) = \begin{cases} 0, & x \le 1/2, \\ \exp\left(1 - \frac{1}{(2x-1)^2}\right), & x \ge 1/2, \end{cases} \quad \text{and} \quad \Phi(x) = \sqrt{e} \int_{-\infty}^{x} e^{-\frac{1}{2}t^2} dt.$$

The main property of the function $F_T(x)$ is that its gradients are large unless $\mathrm{prog}(x) \ge T$.

**Lemma D.1** (Carmon et al. (2020); Arjevani et al. (2022)). *The function $F_T$ satisfies:*

1. $F_T(0) - \inf_{x \in \mathbb{R}^T} F_T(x) \le \Delta^0 T$, *where* $\Delta^0 = 12$.

2. *The function $F_T$ is $l_1$–smooth, where $l_1 = 152$.*

3. *For all $x \in \mathbb{R}^T$, $\|\nabla F_T(x)\|_\infty \le \gamma_\infty$, where $\gamma_\infty = 23$.*

4. *For all $x \in \mathbb{R}^T$, $\mathrm{prog}(\nabla F_T(x)) \le \mathrm{prog}(x) + 1$.*

5. *For all $x \in \mathbb{R}^T$, if $\mathrm{prog}(x) < T$, then $\|\nabla F_T(x)\| > 1$.*

We use these properties in the proofs.

## D.2  Proof of Theorem 6.4

**Theorem 6.4.** *Let us consider the oracle class $\mathcal{O}_{\tau_1,\dots,\tau_n}^{\sigma^2}$ for some $\sigma^2 > 0$ and $0 < \tau_1 \le \dots \le \tau_n$. We fix any $L, \Delta > 0$ and $0 < \varepsilon \le c'L\Delta$. In view Protocol 3, for any algorithm $A \in \mathcal{A}_{zr}$, there exists a function $f \in \mathcal{F}_{\Delta,L}$ and oracles and distributions $((O_1,\dots,O_n),(\mathcal{D}_1,\dots,\mathcal{D}_n)) \in \mathcal{O}_{\tau_1,\dots,\tau_n}^{\sigma^2}(f)$ such that $\mathbb{E}\left[\inf_{k \in S_t} \|\nabla f(x^k)\|^2\right] > \varepsilon$, where $S_t := \{k \in \mathbb{N}_0 \mid t^k \le t\}$, and*

$$t = c \times \min_{m \in [n]} \left[\left(\frac{1}{m}\sum_{i=1}^{m}\frac{1}{\tau_i}\right)^{-1}\left(\frac{L\Delta}{\varepsilon} + \frac{\sigma^2 L\Delta}{m\varepsilon^2}\right)\right].$$

*The quantities $c'$ and $c$ are universal constants.*

Before we prove the theorem, let us briefly explain the idea. In Steps 1 and 2 of the proof, we construct the appropriate scaled function and stochastic oracles using the function (19). These steps are almost the same as in (Carmon et al., 2020; Arjevani et al., 2022).

In Step 3, we use the zero-chain property of the function (19) and the zero-respecting property of algorithms that would guarantee us that unless oracles send us a non-zero coordinate, an algorithm would not be able to progress to a new coordinate. The oracles send a non-zero coordinate with some probability $p$. We have $n$ parallel oracles that flip random coins *in parallel*. With a large probability, we show that will *not* get a new coordinate earlier than

$$\approx \min_{m \in [n]} \left[ \left( \sum_{i=1}^{m} \frac{1}{\tau_i} \right)^{-1} \left( \frac{1}{p} + m \right) \right]$$

seconds, where $\tau_i$ are the delays of the oracles. So, with a large probability, we will not be able to solve the optimization earlier than

$$\approx T \times \min_{m \in [n]} \left[ \left( \sum_{i=1}^{m} \frac{1}{\tau_i} \right)^{-1} \left( \frac{1}{p} + m \right) \right],$$

where $T$ is the dimension of the problem.

*Proof.* (**Step 1**: $f \in \mathcal{F}_{\Delta,L}$)

Let us fix $\lambda > 0$ and take a function $f(x) := L\lambda^2/l_1 F_T\left(\frac{x}{\lambda}\right)$, where the function $F_T$ is defined in Section D.1. Note that the function $f$ is $L$-smooth:

$$\|\nabla f(x) - \nabla f(y)\| = L\lambda/l_1 \left\| F_T\left(\frac{x}{\lambda}\right) - F_T\left(\frac{y}{\lambda}\right) \right\| \leq L\lambda \left\| \frac{x}{\lambda} - \frac{y}{\lambda} \right\| = L\|x - y\| \quad \forall x, y \in \mathbb{R}^d.$$

Let us take

$$T = \left\lfloor \frac{\Delta l_1}{L\lambda^2 \Delta^0} \right\rfloor,$$

then

$$f(0) - \inf_{x \in \mathbb{R}^T} f(x) = \frac{L\lambda^2}{l_1}(F_T(0) - \inf_{x \in \mathbb{R}^T} F_T(x)) \leq \frac{L\lambda^2 \Delta^0 T}{l_1} \leq \Delta.$$

We showed that the function $f \in \mathcal{F}_{\Delta,L}$.

(**Step 2**: Oracle Class)

In the oracles $O_i$, we have the freedom to choose a mapping $\widehat{\nabla} f(\cdot; \cdot)$ (see (7)). Let us take

$$[\widehat{\nabla} f(x; \xi)]_j := \nabla_j f(x) \left( 1 + \mathbb{1}[j > \text{prog}(x)] \left( \frac{\xi}{p} - 1 \right) \right) \quad \forall x \in \mathbb{R}^T,$$

and $\mathcal{D}_i = \text{Bernouilli}(p)$ for all $i \in [n]$, where $p \in (0, 1]$. We denote $[x]_j$ as the $j^{\text{th}}$ index of a vector $x \in \mathbb{R}^T$. It is left to show this mapping is unbiased and $\sigma^2$-variance-bounded. Indeed,

$$\mathbb{E}\left[ [\widehat{\nabla} f(x, \xi)]_i \right] = \nabla_i f(x) \left( 1 + \mathbb{1}[i > \text{prog}(x)] \left( \frac{\mathbb{E}[\xi]}{p} - 1 \right) \right) = \nabla_i f(x)$$

for all $i \in [T]$, and

$$\mathbb{E}\left[ \left\| \widehat{\nabla} f(x; \xi) - \nabla f(x) \right\|^2 \right] \leq \max_{j \in [T]} |\nabla_j f(x)|^2 \mathbb{E}\left[ \left( \frac{\xi}{p} - 1 \right)^2 \right]$$

because the difference is non-zero only in one coordinate. Thus

$$\mathbb{E}\left[ \left\| \widehat{\nabla} f(x, \xi) - \nabla f(x) \right\|^2 \right] \leq \frac{\|\nabla f(x)\|_\infty^2 (1 - p)}{p} = \frac{L^2 \lambda^2 \left\| F_T\left(\frac{x}{\lambda}\right) \right\|_\infty^2 (1 - p)}{l_1^2 p}$$

$$\leq \frac{L^2 \lambda^2 \gamma_\infty^2 (1 - p)}{l_1^2 p} \leq \sigma^2,$$

where we take

$$p = \min \left\{ \frac{L^2 \lambda^2 \gamma_\infty^2}{\sigma^2 l_1^2}, 1 \right\}.$$

(**Step 3**: Analysis of Protocol)

We choose

$$\lambda = \frac{\sqrt{2\varepsilon}l_1}{L}$$

to ensure that $\|\nabla f(x)\|^2 = \frac{L^2 \lambda^2}{l_1^2} \left\| \nabla F_T(\frac{x}{\lambda}) \right\|^2 > 2\varepsilon \mathbb{1}\left[\text{prog}(x) < T\right]$ for all $x \in \mathbb{R}^T$, where we use Lemma D.1. Thus

$$T = \left\lfloor \frac{\Delta L}{2\varepsilon l_1 \Delta^0} \right\rfloor$$

and

$$p = \min \left\{ \frac{2\varepsilon \gamma_\infty^2}{\sigma^2}, 1 \right\}.$$

Protocol 3 generates a sequence $\{x^k\}_{k=0}^\infty$. We have

$$\inf_{k \in S_t} \left\| \nabla f(x^k) \right\|^2 > 2\varepsilon \inf_{k \in S_t} \mathbb{1}\left[\text{prog}(x^k) < T\right]. \tag{20}$$

Using Lemma D.2 with $\delta = 1/2$ and (20), we obtain

$$\mathbb{E}\left[ \inf_{k \in S_t} \left\| \nabla f(x^k) \right\|^2 \right] \geq 2\varepsilon \mathbb{P}\left( \inf_{k \in S_t} \mathbb{1}\left[\text{prog}(x^k) < T\right] \geq 1 \right) > \varepsilon$$

for

$$t = \frac{1}{24} \min_{m \in [n]} \left[ \left( \sum_{i=1}^m \frac{1}{\tau_i} \right)^{-1} \left( \frac{\sigma^2}{2\varepsilon \gamma_\infty^2} + m \right) \right] \left( \frac{\Delta L}{2\varepsilon l_1 \Delta^0} - 2 \right).$$

$\square$

## D.3 Auxillary lemmas

### D.3.1 Proof of Lemma D.2

**Lemma D.2.** *Let us fix $T, T' \in \mathbb{N}$ such that $T \leq T'$, consider Protocol 3 with a differentiable function $f : \mathbb{R}^{T'} \to \mathbb{R}$ such that $\text{prog}(\nabla f(x)) \leq \text{prog}(x) + 1$ for all $x \in \text{domain}(f)$, delays $0 < \tau_1 \leq \cdots \leq \tau_n$, distributions $\mathcal{D}_i = \text{Bernouilli}(p)$ and oracles $O_i = O_{\tau_i}^{\widehat{\nabla}f}$ for all $i \in [n]$, mappings*

$$[\widehat{\nabla}f(x;\xi)]_j = \nabla_j f(x) \left( 1 + \mathbb{1}\left[ j > \text{prog}(x) \right] \left( \frac{\xi}{p} - 1 \right) \right) \quad \forall x \in \mathbb{R}^{T'}, \forall \xi \in \{0,1\}, \forall j \in [T], \tag{21}$$

*and an algorithm $A \in \mathcal{A}_{\text{zr}}$. With probability not less than $1 - \delta$,*

$$\inf_{k \in S_t} \mathbb{1}\left[\text{prog}(x^k) < T\right] \geq 1$$

*for*

$$t \leq \frac{1}{24} \min_{m \in [n]} \left[ \left( \sum_{i=1}^m \frac{1}{\tau_i} \right)^{-1} \left( \frac{1}{p} + m \right) \right] \left( \frac{T}{2} + \log \delta \right),$$

*where the iterates $x^k$ are defined in Protocol 3.*

*Proof.* **(Part 1):** *Comment: in this part, we formally show that if* $\inf_{k \in S_t} \mathbb{1}\left[\text{prog}(x^k) < T\right] < 1$ *holds, then we have the inequality* $\sum_{i=1}^{T} \widehat{t}_{\eta_i} \leq t$, *where* $\widehat{t}_{\eta_i}$ *are random variables with some known "good" distributions. If* $\inf_{k \in S_t} \mathbb{1}\left[\text{prog}(x^k) < T\right] < 1$, *then it means that exists* $k$ *such that* $\text{prog}(x^k) = T$. *Note that the algorithm is zero-respecting, so it can not progress to* $T^{th}$ *coordinate unless the oracles generate stochastic gradients with non-zero* $1^{st}$, $2^{nd}$, ..., $T^{th}$ *coordinates. The oracles flip coins in parallel, so the algorithm should wait for the moment when the oracles flip a success. At the same time, it takes time to generate a coin (calculate a stochastic gradient), and the oracles can not flip more than* $k$ *coins before some time* $\widehat{t}_k$. *So if the* $\eta_i$ *is an index of the first success to generate a non-zero* $i^{th}$ *coordinate, then the algorithm should wait at least* $\widehat{t}_{\eta_i}$ *seconds. Next, we give a formal proof.*

Let us fix $t \geq 0$ and define the smallest index $k(i)$ of the sequence when the progress $\text{prog}(x^{k(i)})$ equals $i$ :

$$k(i) := \inf\left\{k \in \mathbb{N}_0 \mid i = \text{prog}(x^k)\right\} \in \mathbb{N}_0 \cup \{\infty\}.$$

If $\inf_{k \in S_t} \mathbb{1}\left[\text{prog}(x^k) < T\right] < 1$ holds, then exists $k \in S_t$ such that $\text{prog}(x^k) = T$, thus, by the definition of $k(T)$, $t^{k(T)} \leq t^k \leq t$, and $k(T) < \infty$. Note that $t^{k(T)}$ is the smallest time when we make progress to the $T^{\text{th}}$ coordinate.

Since $x^0 = 0$ and $A$ is a zero-respecting algorithm, the algorithm can return a vector $x^k$ with the non-zero first coordinate only if some of returned by the oracles stochastic gradients have the first coordinate not equal to zero. The oracles $O_i$ are constructed in such a way (see (21) and (7)) that they zero out a coordinate based on i.i.d. Bernoulli trials.

**Definition D.3** (Sequence $k_j^{\xi}$). Let us consider a set

$$\left\{k \in \mathbb{N} \mid s_{i^k,q}^{k-1} = 1 \text{ and } t^k \geq s_{i^k,t}^{k-1} + \tau_{i^k}\right\}, \quad s_{i^k}^{k-1} \equiv (s_{i^k,t}^{k-1}, s_{i^k,q}^{k-1}, s_{i^k,x}^{k-1}).$$

We order this set and define the result sequence as $\{k_j^{\xi}\}_{j=1}^{m}$, where $m \in [0, \infty]$ is the size of the sequence. The sequence $k_i^{\xi}$ is a subsequence of iterations where the oracles use the generated Bernouilli random variables in the third output of (7). The sequence $s_{i^k}^{k-1}$ is defined in Protocol 3.

Let $k_{\text{success}}$ be the *first* iteration index when the oracles use a draw $\xi = 1$, i.e.,

$$k_{\text{success}} := \inf\{k \mid \xi^k = 1 \text{ and } k \in \{k_j^{\xi}\}_{j=1}^{m}\} \in \mathbb{N} \cup \{\infty\}.$$

Since the algorithm $A$ is a zero-respecting and the function $f$ is a zero-chain function, i.e., $\text{prog}(\nabla f(x)) \leq \text{prog}(x) + 1$ for all $x \in \text{domain}(f)$, then $\text{prog}(g^k) = \text{prog}(x^k) = 0$ for all $k < k_{\text{success}}$. If $\inf_{k \in S_t} \mathbb{1}\left[\text{prog}(x^k) < T\right] < 1$ holds, then $k_{\text{success}} < \infty$, and $t^{k_{\text{success}}} \leq t^{k(1)}$.

The oracles use the generated Bernoulli random variables $\{\xi^k \mid k \in \{k_j^{\xi}\}_{j=1}^{m}\}$. Let us denote the index of the first successful trial as $\eta_1$, i.e.,

$$\eta_1 := \inf\{i \mid \xi^{k_i^{\xi}} = 1 \text{ and } i \in [1, m]\} \in \mathbb{N} \cup \{\infty\}.$$

The $i^{\text{th}}$ worker can generate the first Bernoulli random variable not earlier than after $\tau_i$ seconds, the second Bernoulli random variable not earlier than after $2\tau_i$ seconds, and so forth.

**Definition D.4** (Sequence $\widehat{t}_k$). Let us consider a *multi-set* of times

$$\{j\tau_i \mid j \geq 1, i \in [n]\} \equiv \{\tau_1, 2\tau_1, \dots\} \uplus \cdots \uplus \{\tau_n, 2\tau_n, \dots\}.$$

We order this multi-set and define the result sequence as $\{\widehat{t}_k\}_{k=1}^{\infty}$, and $\widehat{t}_{\infty} := \lim_{k \to \infty} \widehat{t}_k = \infty$.

Then $\eta_1^{\text{th}}$ Bernoulli random variable can not be generated earlier than $\widehat{t}_{\eta_1}$ because $\widehat{t}_{\eta_1}$ is the earliest time when the oracles can generate $\eta_1$ random variables. Therefore, if $\inf_{k \in S_t} \mathbb{1}\left[\text{prog}(x^k) < T\right] < 1$ holds, then $\widehat{t}_{\eta_1} \leq t^{k_{\text{success}}} \leq t^{k(1)}$.

Using the same reasoning, $t^{k(j+1)} \geq t^{k(j)} + \widehat{t}_{\eta_{j+1}}$, where $\eta_{j+1}$ is the index of the first successful trial of Bernouilli random variables when $\text{prog}(\cdot) = j$ in the sequence $x^k$. More formally:

**Definition D.5** (Sequence $k_{j,i}^\xi$). Let us consider a set

$$\{k \in \mathbb{N} \,|\, s_{i^k,q}^{k-1} = 1 \text{ and } t^k \geq s_{i^k,t}^{k-1} + \tau_{i^k} \text{ and } \text{prog}(s_{i^k,x}^{k-1}) = j\}.$$

We order this set and define the result sequence as $\{k_{j,i}^\xi\}_{i=1}^{m_{j+1}}$, where $m_{j+1} \in [0, \infty]$ is the size of the sequence. The sequence $k_{j,i}^\xi$ is a subsequence of iterations where the oracles use the generated Bernouilli random variables in (7) when $\text{prog}(s_x) = j$.

Then

$$\eta_{j+1} := \inf\{i \,|\, \xi^{k_{j,i}^\xi} = 1 \text{ and } i \in [1, m_{j+1}]\} \in \mathbb{N} \cup \{\infty\} \quad \forall j \in \{0, \ldots, T-1\}. \tag{22}$$

By the definition of $k(j)$, $x^{k(j)}$ is the first vector of the sequence, that contains a non-zero $j$th coordinate. Thus the oracles will start returning stochastic gradients that potentially have a non-zero $j+1^{\text{th}}$ coordinate starting only from the iteration $k(j)$. Therefore,

$$t^{k(T)} \geq t^{k(T-1)} + \widehat{t}_{\eta_T} \geq \sum_{i=1}^{T} \widehat{t}_{\eta_i}.$$

Combining the observations, if $\inf_{k \in S_t} \mathbb{1}\left[\text{prog}(x^k) < T\right] < 1$ holds, then $\sum_{i=1}^{T} \widehat{t}_{\eta_i} \leq t^{k(T)} \leq t$. Thus

$$\mathbb{P}\left(\inf_{k \in S_t} \mathbb{1}\left[\text{prog}(x^k) < T\right] < 1\right) \leq \mathbb{P}\left(\sum_{i=1}^{T} \widehat{t}_{\eta_i} \leq t\right) \quad \forall t \geq 0.$$

In Section D.3.2, we prove the following inequality that we use in Part 2 of the proof.

**Lemma D.6.** *Let us take $l_{j+1} \in \mathbb{N}$. Then*

$$\mathbb{P}\left(\eta_{j+1} = l_{j+1} | \eta_j, \ldots, \eta_1\right) \leq (1-p)^{l_{j+1}-1} p$$

*for all $j \in \{0, \ldots, T-1\}$.*

**(Part 2):** *Comment: in this part, we use the standard technique to bound the large deviations of the sum $\sum_{i=1}^{T} \widehat{t}_{\eta_i}$.*

Let us fix $t' \geq 0$. Recall Definition D.4 of $\{\widehat{t}_k\}_{k=1}^\infty$. If the number of workers $n = 1$, then $\widehat{t}_k = k\tau_1$ for all $k \geq 1$. For $n > 1$, the sequence $\{\widehat{t}_k\}_{k=1}^\infty$ has more complicated structure and depends on the delays $\tau_1, \ldots, \tau_n$.

For any $k \geq 1$, if $\widehat{t}_k \leq t'$, then $k \leq \sum_{i=1}^{n} \lfloor \frac{t'}{\tau_i} \rfloor$. Indeed, let us assume that $k > \sum_{i=1}^{n} \lfloor \frac{t'}{\tau_i} \rfloor$. The sequence $\widehat{t}_k$ is constructed by the ordering the multi-set $\{j\tau_i \,|\, j \geq 1, i \in [n]\}$. The number of elements, which are less or equal to $t'$, equals $\sum_{i=1}^{n} \lfloor \frac{t'}{\tau_i} \rfloor$. Thus, we get a contradiction.

It means that

$$\mathbb{P}\left(\widehat{t}_{\eta_{j+1}} \leq t' | \eta_j, \ldots, \eta_1\right) \leq \mathbb{P}\left(\eta_{j+1} \leq \sum_{i=1}^{n} \left\lfloor \frac{t'}{\tau_i} \right\rfloor \bigg| \eta_j, \ldots, \eta_1\right).$$

Using Lemma D.6, we have

$$\mathbb{P}\left(\widehat{t}_{\eta_{j+1}} \leq t' | \eta_j, \ldots, \eta_1\right) \leq \sum_{j=1}^{\sum_{i=1}^{n} \left\lfloor \frac{t'}{\tau_i} \right\rfloor} (1-p)^{j-1} p.$$

If $0 \leq t' < \tau_1$, then $\sum_{i=1}^{n} \lfloor t'/\tau_i \rfloor = 0$, and

$$\mathbb{P}\left(\widehat{t}_{\eta_{j+1}} \leq t' | \eta_j, \ldots, \eta_1\right) = 0.$$

Otherwise, if $t' > \tau_1$, then $\sum_{i=1}^{n} \lfloor t'/\tau_i \rfloor \geq 1$, and

$$\mathbb{P}\left(\widehat{t}_{\eta_{j+1}} \leq t' \big| \eta_j, \ldots, \eta_1\right) \leq 1 - (1-p)^{\sum_{i=1}^{n} \lfloor \frac{t'}{\tau_i} \rfloor} \leq p \sum_{i=1}^{n} \left\lfloor \frac{t'}{\tau_i} \right\rfloor,$$

where we use the fact that $1 - (1-p)^m \leq pm$ for all $p \in [0,1]$ and $m \in \mathbb{N}$. For all $t' \geq 0$, we have

$$\mathbb{P}\left(\widehat{t}_{\eta_{j+1}} \leq t' \big| \eta_j, \ldots, \eta_1\right) \leq p \sum_{i=1}^{n} \left\lfloor \frac{t'}{\tau_i} \right\rfloor.$$

Let us define

$$p' := p \sum_{i=1}^{n} \left\lfloor \frac{t'}{\tau_i} \right\rfloor,$$

then

$$\mathbb{P}\left(\widehat{t}_{\eta_{j+1}} \leq t' \big| \eta_j, \ldots, \eta_1\right) \leq p'. \tag{23}$$

Let us fix $s \geq 0$ and $\widehat{t} \geq 0$. Using the Chernoff method, we have

$$\mathbb{P}\left(\sum_{i=1}^{T} \widehat{t}_{\eta_i} \leq \widehat{t}\right) = \mathbb{P}\left(-s\left(\sum_{i=1}^{T} \widehat{t}_{\eta_i}\right) \geq -s\widehat{t}\right) = \mathbb{P}\left(\exp\left(-s\sum_{i=1}^{T} \widehat{t}_{\eta_i}\right) \geq \exp\left(-s\widehat{t}\right)\right)$$

$$\leq e^{s\widehat{t}}\mathbb{E}\left[\exp\left(-s\sum_{i=1}^{T} \widehat{t}_{\eta_i}\right)\right].$$

Let us bound the expected value separately:

$$\mathbb{E}\left[\exp\left(-s\sum_{i=1}^{T} \widehat{t}_{\eta_i}\right)\right] = \mathbb{E}\left[\prod_{i=1}^{T} \mathbb{E}\left[e^{-s\widehat{t}_{\eta_i}} \big| \eta_{i-1}, \ldots, \eta_1\right]\right].$$

Since $\widehat{t}_{\eta_i} \geq 0$, we have

$$\mathbb{E}\left[e^{-s\widehat{t}_{\eta_i}} \big| \eta_{i-1}, \ldots, \eta_1\right] = \mathbb{E}\left[e^{-s\widehat{t}_{\eta_i}} \big| \widehat{t}_{\eta_i} \leq t', \eta_{i-1}, \ldots, \eta_1\right] \mathbb{P}\left(\widehat{t}_{\eta_i} \leq t' \big| \eta_{i-1}, \ldots, \eta_1\right)$$

$$+ \mathbb{E}\left[e^{-s\widehat{t}_{\eta_i}} \big| \widehat{t}_{\eta_i} > t', \eta_{i-1}, \ldots, \eta_1\right] \left(1 - \mathbb{P}\left(\widehat{t}_{\eta_i} \leq t' \big| \eta_{i-1}, \ldots, \eta_1\right)\right)$$

$$\leq \mathbb{P}\left(\widehat{t}_{\eta_i} \leq t' \big| \eta_{i-1}, \ldots, \eta_1\right) + e^{-st'}\left(1 - \mathbb{P}\left(\widehat{t}_{\eta_i} \leq t' \big| \eta_{i-1}, \ldots, \eta_1\right)\right)$$

$$\overset{(23)}{\leq} p' + e^{-st'}\left(1 - p'\right).$$

Thus

$$\mathbb{E}\left[\exp\left(-s\sum_{i=1}^{T} \widehat{t}_{\eta_i}\right)\right] \leq \left(p' + e^{-st'}\left(1 - p'\right)\right)^{T}$$

and

$$\mathbb{P}\left(\sum_{i=1}^{T} \widehat{t}_{\eta_i} \leq \widehat{t}\right) \leq e^{s\widehat{t}}\left(p' + e^{-st'}\left(1 - p'\right)\right)^{T} = e^{s\widehat{t}-st'T}\left(1 + \left(e^{st'} - 1\right)p'\right)^{T}.$$

Let us take $s = 1/t'$, and get

$$\mathbb{P}\left(\sum_{i=1}^{T} \widehat{t}_{\eta_i} \leq \widehat{t}\right) \leq e^{\widehat{t}/t'-T}\left(1 + (e-1)p'\right)^{T} \leq e^{\widehat{t}/t'-T+2p'T}. \tag{24}$$

Let us recall the definition of $p'$:

$$p' = p\left(\sum_{i=1}^{n} \left\lfloor \frac{t'}{\tau_i} \right\rfloor\right)$$

Now, we have to take the right $t'$. We will take it using a nonconstructive definition. Assume that $t' = \frac{1}{4p}\left(\sum_{i=1}^{j^*}\frac{1}{\tau_i}\right)^{-1}$, where

$$j^* = \inf\left\{m \in [n]\;\middle|\;\frac{1}{4p}\left(\sum_{i=1}^{m}\frac{1}{\tau_i}\right)^{-1} < \tau_{m+1}\right\}\quad(\tau_{n+1}\equiv\infty).$$

This set is not empty because $n$ belongs to it. Using the definition of $j^*$, we have

$$p' = p\left(\sum_{i=1}^{n}\left\lfloor\frac{t'}{\tau_i}\right\rfloor\right) = p\left(\sum_{i=1}^{j^*}\left\lfloor\frac{t'}{\tau_i}\right\rfloor\right) \le pt'\left(\sum_{i=1}^{j^*}\frac{1}{\tau_i}\right) = \frac{1}{4}.$$

Substituting this inequality to (24), we obtain

$$\mathbb{P}\left(\sum_{i=1}^{T}\widehat{t}_{\eta_i} \le \widehat{t}\right) \le e^{\widehat{t}/t' - \frac{T}{2}}.$$

For $\widehat{t} \le t'\left(\frac{T}{2} + \log\delta\right)$, we have

$$\mathbb{P}\left(\sum_{i=1}^{T}\widehat{t}_{\eta_i} \le \widehat{t}\right) \le \delta.$$

Recall that $t' = \frac{1}{4p}\left(\sum_{i=1}^{j^*}\frac{1}{\tau_i}\right)^{-1}$. Using Lemma D.7, we have

$$t' \ge \frac{1}{24}\min_{m\in[n]}\left[\left(\sum_{i=1}^{m}\frac{1}{\tau_i}\right)^{-1}\left(\frac{1}{p}+m\right)\right].$$

Finally, we obtain

$$\mathbb{P}\left(\inf_{k\in S_t}\mathbb{1}\left[\mathrm{prog}(x^k) < T\right] < 1\right) \le \mathbb{P}\left(\sum_{i=1}^{T}\widehat{t}_{\eta_i} \le t\right) \le \delta \tag{25}$$

for

$$t \le \frac{1}{24}\min_{m\in[n]}\left[\left(\sum_{i=1}^{m}\frac{1}{\tau_i}\right)^{-1}\left(\frac{1}{p}+m\right)\right]\left(\frac{T}{2}+\log\delta\right).$$

$\square$

### D.3.2 Proof of Lemma D.6

In the following lemma, we use notations from Part 1 of the proof of Lemma D.2.

**Lemma D.6.** *Let us take* $l_{j+1} \in \mathbb{N}$. *Then*

$$\mathbb{P}\left(\eta_{j+1} = l_{j+1}|\eta_j,\ldots,\eta_1\right) \le (1-p)^{l_{j+1}-1}p$$

*for all* $j \in \{0,\ldots,T-1\}$.

In this lemma, we want to bound the probability for the random variable $\eta_{j+1}$ from (22). $\eta_{j+1}$ is the index of the first successful trial of the sequence of Bernouilli random variables. At first sight, this is a trivial task since $\eta_{j+1}$ has a distribution similar to the geometric distribution. But the main problem here is that the sequence $k_{j,i}^\xi$ and the quantity $m_{j+1}$ are also random variables. Therefore, we must be careful with this.

*Proof.* Since the image of the random variables $\eta_j, \ldots, \eta_1$ is in $\mathbb{N} \cap \{\infty\}$. Let us take $l_1, \ldots, l_j \in \mathbb{N} \cup \{\infty\}$, and prove the theorem for a probability conditioned on an event $\bigcap_{i=1}^{j}\{\eta_i = l_i\}$ such that $\mathbb{P}\left(\bigcap_{i=1}^{j}\{\eta_i = l_i\}\right) > 0$. Therefore, it is enough to prove that

$$\mathbb{P}\left(\eta_{j+1} = l_{j+1} \middle| \bigcap_{i=1}^{j}\{\eta_i = l_i\}\right) \leq (1-p)^{l_{j+1}-1}p.$$

First, assume that exists $i \in [j]$ such that $l_i = \infty$. It means that, for all $k \geq 0$, $\text{prog}(x^k) < j$, thus

$$\mathbb{P}\left(\eta_{j+1} = l_{j+1} \middle| \bigcap_{i=1}^{j}\{\eta_i = l_i\}\right) = 0$$

for all $l_{j+1} \in \mathbb{N}$. Let us explain this step. if exists $i \in [j]$ such that $l_i = \infty$, then an algorithm never get a progress to $j^{\text{th}}$ coordinate, thus $\eta_{j+1} = \inf\{i \mid \xi^{k^{\xi}_{j,i}} = 1 \text{ and } i \in [1, m_{j+1}]\} = \infty$ a.s. because $m_{j+1} = 0$ and $k^{\xi}_{j,i}$ is an empty sequence.

Assume that $l_i < \infty$ for all $i \in [j]$. By the definition of $\eta_{j+1}$, we have $m_{j+1} \geq l_{j+1}$ and $\xi^{k^{\xi}_{j,1}} = \cdots = \xi^{k^{\xi}_{j,l_{j+1}-1}} = 0$ and $\xi^{k^{\xi}_{j,l_{j+1}}} = 1$. Thus

$$\mathbb{P}\left(\eta_{j+1} = l_{j+1} \middle| \bigcap_{i=1}^{j}\{\eta_i = l_i\}\right) \leq \mathbb{P}\left(\bigcap_{i=1}^{l_{j+1}-1}\{\xi^{k^{\xi}_{j,i}} = 0\}, \xi^{k^{\xi}_{j,l_{j+1}}} = 1, m_{j+1} \geq l_{j+1} \middle| \bigcap_{i=1}^{j}\{\eta_i = l_i\}\right).$$

Since $k^{\xi}_{j,s} < k^{\xi}_{j,i}$ a.s. for all $s < i \in [m_{j+1}]$, using the law of total probability, we have

$$\mathbb{P}\left(\eta_{j+1} = l_{j+1} \middle| \bigcap_{i=1}^{j}\{\eta_i = l_i\}\right)$$

$$\leq \sum_{k_1 < \cdots < k_{l_{j+1}}=1}^{\infty} \mathbb{P}\left(\bigcap_{i=1}^{l_{j+1}-1}\{\xi^{k^{\xi}_{j,i}} = 0\}, \xi^{k^{\xi}_{j,l_{j+1}}} = 1, m_{j+1} \geq l_{j+1}, \bigcap_{i=1}^{l_{j+1}}\{k^{\xi}_{j,i} = k_i\} \middle| \bigcap_{i=1}^{j}\{\eta_i = l_i\}\right)$$

$$= \sum_{k_1 < \cdots < k_{l_{j+1}}=1}^{\infty} \mathbb{P}\left(\bigcap_{i=1}^{l_{j+1}-1}\{\xi^{k_i} = 0\}, \xi^{k_{l_{j+1}}} = 1, m_{j+1} \geq l_{j+1}, \bigcap_{i=1}^{l_{j+1}}\{k^{\xi}_{j,i} = k_i\} \middle| \bigcap_{i=1}^{j}\{\eta_i = l_i\}\right),$$

where $\sum_{k_1 < \cdots < k_{l_{j+1}}=1}^{\infty}$ is a sum over a set $\{(k_1, \ldots, k_{l_{j+1}}) \in \mathbb{N}^{l_{j+1}} \mid \forall i < p \in [l_{j+1}] : k_i < k_p\}$. Next, if the event

$$\bigcap_{i=1}^{l_{j+1}}\{k^{\xi}_{j,i} = k_i\}\bigcap\{m_{j+1} \geq l_{j+1}\}$$

holds, then an event $\bigcap_{i=1}^{l_{j+1}} A_{k_i}$ holds, where

$$A_{k_i} := \{s^{k_i-1}_{i^{k_i},q} = 1 \text{ and } t^{k_i} \geq s^{k_i-1}_{i^{k_i},t} + \tau_{i^{k_i}} \text{ and } \text{prog}(s^{k_i-1}_{i^{k_i},x}) = j\}.$$

At the same time, if $\bigcap_{i=1}^{l_{j+1}} A_{k_i}$ holds, then $\{m_{j+1} \geq l_{j+1}\}$ holds. Therefore,

$$\bigcap_{i=1}^{l_{j+1}}\{k^{\xi}_{j,i} = k_i\}\bigcap\{m_{j+1} \geq l_{j+1}\} = \bigcap_{i=1}^{l_{j+1}}\left(\{k^{\xi}_{j,i} = k_i\}\bigcap A_{k_i}\right)$$

and

$$\mathbb{P}\left(\eta_{j+1} = l_{j+1} \middle| \bigcap_{i=1}^{j}\{\eta_i = l_i\}\right)$$

$$\leq \sum_{k_1 < \cdots < k_{l_{j+1}}=1}^{\infty} \mathbb{P}\left(\bigcap_{i=1}^{l_{j+1}-1}\{\xi^{k_i}=0\}, \xi^{k_{l_{j+1}}}=1, \bigcap_{i=1}^{l_{j+1}}\left(\{k_{j,i}^{\xi}=k_i\}\bigcap A_{k_i}\right)\middle| \bigcap_{i=1}^{j}\{\eta_i=l_i\}\right).$$

Let us define $\sigma(\xi^1,\ldots,\xi^{k_{l_{j+1}-1}})$ as a sigma-algebra generated by $\xi^1,\ldots,\xi^{k_{l_{j+1}-1}}$. Note that, for all $i \in [l_{j+1}-1]$, the event $\{\xi^{k_i}=0\} \in \sigma(\xi^1,\ldots,\xi^{k_{l_{j+1}-1}})$. Also, for all $i \in [l_{j+1}]$, the event $\{k_{j,i}^{\xi}=k_i\}\bigcap A_{k_i} \in \sigma(\xi^1,\ldots,\xi^{k_{l_{j+1}-1}})$. Finally, since $k_{i-1,l_i}^{\xi} < k_{j,l_{j+1}}^{\xi}$, the event

$$A_{k_{l_{j+1}}}\bigcap\{k_{j,l_{j+1}}^{\xi}=k_{l_{j+1}}\}\bigcap\{\eta_i=l_i\} \subseteq \sigma(\xi^1,\ldots,\xi^{k_{l_{j+1}-1}})$$

for all $i \in [j]$. Therefore, the event $\{\xi^{k_{l_{j+1}}}=1\}$ is independent of the event

$$\bigcap_{i=1}^{l_{j+1}-1}\{\xi^{k_i}=0\}\bigcap_{i=1}^{l_{j+1}}\left(\{k_{j,i}^{\xi}=k_i\}\bigcap A_{k_i}\right)\bigcap_{i=1}^{j}\{\eta_i=l_i\} \in \sigma(\xi^1,\ldots,\xi^{k_{l_{j+1}-1}})$$

because $\xi^k$ are i.i.d. random variables. Using the independence and the equality $\mathbb{P}\left(\xi^{k_{l_{j+1}}}=1\right)=p$, we have

$$\mathbb{P}\left(\eta_{j+1}=l_{j+1}\middle| \bigcap_{i=1}^{j}\{\eta_i=l_i\}\right)$$

$$\leq p \sum_{k_1 < \cdots < k_{l_{j+1}}=1}^{\infty} \mathbb{P}\left(\bigcap_{i=1}^{l_{j+1}-1}\{\xi^{k_i}=0\}, \bigcap_{i=1}^{l_{j+1}}\left(\{k_{j,i}^{\xi}=k_i\}\bigcap A_{k_i}\right)\middle| \bigcap_{i=1}^{j}\{\eta_i=l_i\}\right).$$

Since the events $\{k_{j,l_{j+1}}^{\xi}=k_{l_{j+1}}\}\bigcap A_{k_{l_{j+1}}}$ do not intersect, we can use the additivity of the probability. If $l_{j+1}=1$, we get

$$\mathbb{P}\left(\eta_{j+1}=l_{j+1}\middle| \bigcap_{i=1}^{j}\{\eta_i=l_i\}\right) \leq p\mathbb{P}\left(\bigcup_{i=1}^{\infty}\left(\{k_{j,l_{j+1}}^{\xi}=i\}\bigcap A_i\right)\middle| \bigcap_{i=1}^{j}\{\eta_i=l_i\}\right) \leq p,$$

and prove the lemma for $l_{j+1}=1$. Otherwise, if $l_{j+1}>1$, we obtain

$$\mathbb{P}\left(\eta_{j+1}=l_{j+1}\middle| \bigcap_{i=1}^{j}\{\eta_i=l_i\}\right)$$

$$\leq p \sum_{k_1 < \cdots < k_{l_{j+1}-1}=1}^{\infty} \mathbb{P}\left(\bigcap_{i=1}^{l_{j+1}-1}\{\xi^{k_i}=0\}, \bigcap_{i=1}^{l_{j+1}-1}\left(\{k_{j,i}^{\xi}=k_i\}\bigcap A_{k_i}\right),\right.$$

$$\left.\bigcup_{i=k_{l_{j+1}-1}+1}^{\infty}\left(\{k_{j,l_{j+1}}^{\xi}=i\}\bigcap A_i\right)\middle| \bigcap_{i=1}^{j}\{\eta_i=l_i\}\right).$$

For any events $A$ and $B$, we have $\mathbb{P}(A,B) \leq \mathbb{P}(A)$, thus

$$\mathbb{P}\left(\eta_{j+1}=l_{j+1}\middle| \bigcap_{i=1}^{j}\{\eta_i=l_i\}\right)$$

$$\leq p \sum_{k_1 < \cdots < k_{l_{j+1}-1}=1}^{\infty} \mathbb{P}\left(\bigcap_{i=1}^{l_{j+1}-1}\{\xi^{k_i}=0\}, \bigcap_{i=1}^{l_{j+1}-1}\left(\{k_{j,i}^{\xi}=k_i\}\bigcap A_{k_i}\right)\middle| \bigcap_{i=1}^{j}\{\eta_i=l_i\}\right).$$

Let us continue for $l_{j+1}>1$ and rewrite the last inequality:

$$\mathbb{P}\left(\eta_{j+1}=l_{j+1}\middle| \bigcap_{i=1}^{j}\{\eta_i=l_i\}\right)$$

$$\leq p \sum_{k_1 < \cdots < k_{l_{j+1}-1} = 1}^{\infty} \mathbb{P}\left( \bigcap_{i=1}^{l_{j+1}-2} \{\xi^{k_i} = 0\}, \xi^{k_{l_{j+1}-1}} = 0, \bigcap_{i=1}^{l_{j+1}-1} \left( \{k_{j,i}^{\xi} = k_i\} \bigcap A_{k_i} \right) \middle| \bigcap_{i=1}^{j} \{\eta_i = l_i\} \right).$$

Note that, for all $i \in [l_{j+1}-2]$, the event $\{\xi^{k_i} = 0\} \in \sigma(\xi^1, \ldots, \xi^{k_{l_{j+1}-2}})$. Also, for all $i \in [l_{j+1}-1]$, the event $\{k_{j,i}^{\xi} = k_i\} \bigcap A_{k_i} \in \sigma(\xi^1, \ldots, \xi^{k_{l_{j+1}-2}})$. Finally, since $k_{i-1,l_i}^{\xi} < k_{j,l_{j+1}-1}^{\xi}$, the event

$$A_{k_{l_{j+1}-1}} \bigcap \{k_{j,l_{j+1}-1}^{\xi} = k_{l_{j+1}-1}\} \bigcap \{\eta_i = l_i\} \subseteq \sigma(\xi^1, \ldots, \xi^{k_{l_{j+1}-2}})$$

for all $i \in [j]$. Therefore, the event $\{\xi^{k_{l_{j+1}-1}} = 0\}$ is independent of the event

$$\bigcap_{i=1}^{l_{j+1}-2} \{\xi^{k_i} = 0\} \bigcap_{i=1}^{l_{j+1}-1} \left( \{k_{j,i}^{\xi} = k_i\} \bigcap A_{k_i} \right) \bigcap_{i=1}^{j} \{\eta_i = l_i\}.$$

Thus, we have

$$\mathbb{P}\left( \eta_{j+1} = l_{j+1} \middle| \bigcap_{i=1}^{j} \{\eta_i = l_i\} \right)$$

$$\leq p(1-p) \sum_{k_1 < \cdots < k_{l_{j+1}-1} = 1}^{\infty} \mathbb{P}\left( \bigcap_{i=1}^{l_{j+1}-2} \{\xi^{k_i} = 0\}, \bigcap_{i=1}^{l_{j+1}-1} \left( \{k_{j,i}^{\xi} = k_i\} \bigcap A_{k_i} \right) \middle| \bigcap_{i=1}^{j} \{\eta_i = l_i\} \right).$$

Since the events $\{k_{j,l_{j+1}-1}^{\xi} = k_{l_{j+1}-1}\} \bigcap A_{k_{l_{j+1}-1}}$ do not intersect, we use the additivity of the probability. If $l_{j+1} = 2$, we get

$$\mathbb{P}\left( \eta_{j+1} = l_{j+1} \middle| \bigcap_{i=1}^{j} \{\eta_i = l_i\} \right) \leq p(1-p)\mathbb{P}\left( \bigcup_{i=1}^{\infty} \left( \{k_{j,l_{j+1}-1}^{\xi} = i\} \bigcap A_i \right) \middle| \bigcap_{i=1}^{j} \{\eta_i = l_i\} \right)$$

$$\leq p(1-p),$$

and prove the lemma for $l_{j+1} = 2$. Otherwise, if $l_{j+1} > 2$, we obtain

$$\mathbb{P}\left( \eta_{j+1} = l_{j+1} \middle| \bigcap_{i=1}^{j} \{\eta_i = l_i\} \right)$$

$$\leq p(1-p) \sum_{k_1 < \cdots < k_{l_{j+1}-2} = 1}^{\infty} \mathbb{P}\left( \bigcap_{i=1}^{l_{j+1}-2} \{\xi^{k_i} = 0\}, \bigcap_{i=1}^{l_{j+1}-2} \left( \{k_{j,i}^{\xi} = k_i\} \bigcap A_{k_i} \right), \right.$$

$$\left. \bigcup_{i=k_{l_{j+1}-2}+1}^{\infty} \left( \{k_{j,l_{j+1}-1}^{\xi} = i\} \bigcap A_i \right) \middle| \bigcap_{i=1}^{j} \{\eta_i = l_i\} \right)$$

$$\leq p(1-p) \sum_{k_1 < \cdots < k_{l_{j+1}-2} = 1}^{\infty} \mathbb{P}\left( \bigcap_{i=1}^{l_{j+1}-2} \{\xi^{k_i} = 0\}, \bigcap_{i=1}^{l_{j+1}-2} \left( \{k_{j,i}^{\xi} = k_i\} \bigcap A_{k_i} \right) \middle| \bigcap_{i=1}^{j} \{\eta_i = l_i\} \right),$$

where we use $\mathbb{P}(A, B) \leq \mathbb{P}(A)$ for any events $A$ and $B$. Using mathematical induction, we can continue and get that

$$\mathbb{P}\left( \eta_{j+1} = l_{j+1} \middle| \bigcap_{i=1}^{j} \{\eta_i = l_i\} \right) \leq p(1-p)^{l_{j+1}-1}.$$

$\square$

### D.3.3 Lemma D.7

This is a technical lemma that we use in the proof of Lemma D.2.

**Lemma D.7.** *Let us consider a sorted sequence* $0 < \tau_1 \leq \cdots \leq \tau_n \leq \tau_{n+1} = \infty$ *and a constant* $S \geq \frac{1}{4}$. *We define*

$$t_1 := S \left( \sum_{i=1}^{j_1^*} \frac{1}{\tau_i} \right)^{-1},$$

*where*

$$j_1^* = \inf \left\{ m \in [n] \,\middle|\, S \left( \sum_{i=1}^{m} \frac{1}{\tau_i} \right)^{-1} < \tau_{m+1} \right\},$$

*and*

$$t_2 := \min_{j \in [n]} \left[ \left( \sum_{i=1}^{j} \frac{1}{\tau_i} \right)^{-1} (S + j) \right].$$

*Then*

$$t_1 \leq t_2 \leq 6t_1.$$

*Proof.* Additionally, let us define

$$j_2^* = \arg\min_{j \in [n]} \left[ \left( \sum_{i=1}^{j} \frac{1}{\tau_i} \right)^{-1} (S + j) \right],$$

where $j_2^*$ is the smallest index. For $j_2^* = 1$, we have

$$t_2 = \tau_1 (S + 1) > \tau_1.$$

For $j_2^* > 1$, we have

$$\left( \sum_{i=1}^{j_2^*} \frac{1}{\tau_i} \right)^{-1} (S + j_2^*) < \left( \sum_{i=1}^{j_2^*-1} \frac{1}{\tau_i} \right)^{-1} (S + j_2^* - 1).$$

From this inequality, we get

$$\left( \sum_{i=1}^{j_2^*-1} \frac{1}{\tau_i} \right) (S + j_2^*) < \left( \sum_{i=1}^{j_2^*} \frac{1}{\tau_i} \right) (S + j_2^* - 1)$$

and

$$\left( \sum_{i=1}^{j_2^*} \frac{1}{\tau_i} \right) < \frac{1}{\tau_{j_2^*}} (S + j_2^*).$$

Thus $\tau_{j_2^*} < t_2$ for all $j_2^* \geq 1$.

Then either $j_2^* \leq j_1^*$ and

$$t_2 = \left( \sum_{i=1}^{j_2^*} \frac{1}{\tau_i} \right)^{-1} (S + j_2^*) \geq S \left( \sum_{i=1}^{j_2^*} \frac{1}{\tau_i} \right)^{-1} \geq S \left( \sum_{i=1}^{j_1^*} \frac{1}{\tau_i} \right)^{-1} = t_1,$$

or $j_2^* > j_1^*$ and

$$t_2 > \tau_{j_2^*} \geq \tau_{j_1^*+1} > S \left( \sum_{i=1}^{j_1^*} \frac{1}{\tau_i} \right)^{-1} = t_1,$$

where we used the definition of $j_1^*$. It concludes that $t_2 \geq t_1$.

Assume that $j_1^* > S + 1$. Since the harmonic mean of a sequence less or equal to the maximum, we have

$$S \left( \sum_{i=1}^{j_1^*-1} \frac{1}{\tau_i} \right)^{-1} < (j_1^* - 1) \left( \sum_{i=1}^{j_1^*-1} \frac{1}{\tau_i} \right)^{-1} \leq \tau_{j_1^*-1} \leq \tau_{j_1^*}.$$

This inequality contradicts the definition of $j_1^*$. It means that $j_1^* \leq S + 1$ and

$$t_2 \leq \left( \sum_{i=1}^{j_1^*} \frac{1}{\tau_i} \right)^{-1} (S + j_1^*) \leq \left( \sum_{i=1}^{j_1^*} \frac{1}{\tau_i} \right)^{-1} (2S + 1) \leq \left( \sum_{i=1}^{j_1^*} \frac{1}{\tau_i} \right)^{-1} (6S) \leq 6t_1.$$

$\square$

### D.4 Proof of Theorems 7.4 and 7.5

#### D.4.1 Proof of Theorems 7.4

**Theorem 7.4.** *Assume that Assumptions 7.1, 7.2 and 7.3 hold. Let us take the batch size $S = \max \left\{ \lceil \sigma^2/\varepsilon \rceil, 1 \right\}$, and $\gamma = \min \left\{ \frac{1}{L}, \frac{\varepsilon S}{2L\sigma^2} \right\} = \Theta \left( 1/L \right)$ in Method 4. Then after*

$$K \geq \frac{24 \Delta L}{\varepsilon}$$

*iterations, the method guarantees that $\frac{1}{K} \sum_{k=0}^{K-1} \mathbb{E} \left[ \left\| \nabla f(x^k) \right\|^2 \right] \leq \varepsilon.$*

*Proof.* Note that Method 4 is just the stochastic gradient method with the batch size $S$. Method 4 can be rewritten as $x^{k+1} = x^k - \gamma \frac{1}{S} \sum_{i=1}^{S} \widehat{\nabla} f(x^k; \xi_i)$, where the $\xi_i$ are independent random samples. It means that we can use the classical SGD result (see Theorem D.8). For a stepsize

$$\gamma = \min \left\{ \frac{1}{L}, \frac{\varepsilon S}{2L\sigma^2} \right\},$$

we have

$$\frac{1}{K} \sum_{k=0}^{K-1} \mathbb{E} \left[ \left\| \nabla f(x^k) \right\|^2 \right] \leq \varepsilon,$$

if

$$K \geq \frac{12 \Delta L}{\varepsilon} + \frac{12 \Delta L \sigma^2}{\varepsilon^2 S}.$$

Using the choice of $S$, we showed that Method 4 converges after

$$K \geq \frac{24 \Delta L}{\varepsilon}$$

steps with

$$\gamma = \min \left\{ \frac{1}{L}, \frac{\varepsilon S}{2L\sigma^2} \right\} \geq \frac{1}{2L}.$$

$\square$

#### D.4.2 The classical SGD theorem

We reprove the classical SGD result (Ghadimi and Lan, 2013; Khaled and Richtárik, 2020).

**Theorem D.8.** *Assume that Assumptions 7.1 and 7.2 hold. We consider the SGD method:*

$$x^{k+1} = x^k - \gamma g(x^k),$$

*where*

$$\gamma = \min \left\{ \frac{1}{L}, \frac{\varepsilon}{2L\sigma^2} \right\}$$

For a fixed $x \in \mathbb{R}^d$, $g(x)$ is a random vector such that $\mathbb{E}\left[g(x)\right] = \nabla f(x)$,

$$\mathbb{E}\left[\|g(x) - \nabla f(x)\|^2\right] \leq \sigma^2, \tag{26}$$

and $g(x^k)$ are independent vectors for all $k \geq 0$. Then

$$\frac{1}{K} \sum_{k=0}^{K-1} \mathbb{E}\left[\|\nabla f(x^k)\|^2\right] \leq \varepsilon$$

for

$$K \geq \frac{4\Delta L}{\varepsilon} + \frac{8\Delta L \sigma^2}{\varepsilon^2}.$$

*Proof.* From Assumption 7.1, we have

$$f(x^{k+1}) \leq f(x^k) + \left\langle \nabla f(x^k), x^{k+1} - x^k \right\rangle + \frac{L}{2}\left\|x^{k+1} - x^k\right\|^2$$

$$= f(x^k) - \gamma \left\langle \nabla f(x^k), g(x^k) \right\rangle + \frac{L\gamma^2}{2}\left\|g(x^k)\right\|^2.$$

We denote $\mathcal{G}^k$ as a sigma-algebra generated by $g(x^0), \ldots, g(x^{k-1})$. Using unbiasedness and (26), we obtain

$$\mathbb{E}\left[f(x^{k+1})\big|\,\mathcal{G}^k\right] \leq f(x^k) - \gamma\left(1 - \frac{L\gamma}{2}\right)\left\|\nabla f(x^k)\right\|^2 + \frac{L\gamma^2}{2}\mathbb{E}\left[\left\|g^k - \nabla f(x^k)\right\|^2\Big|\,\mathcal{G}^k\right]$$

$$\leq f(x^k) - \gamma\left(1 - \frac{L\gamma}{2}\right)\left\|\nabla f(x^k)\right\|^2 + \frac{L\gamma^2\sigma^2}{2}.$$

Since $\gamma \leq 1/L$, we get

$$\mathbb{E}\left[f(x^{k+1})\big|\,\mathcal{G}^k\right] \leq f(x^k) - \frac{\gamma}{2}\left\|\nabla f(x^k)\right\|^2 + \frac{L\gamma^2\sigma^2}{2}.$$

We subtract $f^*$ and take the full expectation to obtain

$$\mathbb{E}\left[f(x^{k+1}) - f^*\right] \leq \mathbb{E}\left[f(x^k) - f^*\right] - \frac{\gamma}{2}\mathbb{E}\left[\left\|\nabla f(x^k)\right\|^2\right] + \frac{L\gamma^2\sigma^2}{2}.$$

Next, we sum the inequality for $k \in \{0, \ldots, K-1\}$:

$$\mathbb{E}\left[f(x^K) - f^*\right] \leq f(x^0) - f^* - \sum_{k=0}^{K-1}\frac{\gamma}{2}\mathbb{E}\left[\left\|\nabla f(x^k)\right\|^2\right] + \frac{KL\gamma^2\sigma^2}{2}$$

$$= \Delta - \sum_{k=0}^{K-1}\frac{\gamma}{2}\mathbb{E}\left[\left\|\nabla f(x^k)\right\|^2\right] + \frac{KL\gamma^2\sigma^2}{2}.$$

Finally, we rearrange the terms and use that $\mathbb{E}\left[f(x^K) - f^*\right] \geq 0$:

$$\frac{1}{K}\sum_{k=0}^{K-1}\mathbb{E}\left[\left\|\nabla f(x^k)\right\|^2\right] \leq \frac{2\Delta}{\gamma K} + L\gamma\sigma^2.$$

The choice of $\gamma$ and $K$ ensures that

$$\frac{1}{K}\sum_{k=0}^{K-1}\mathbb{E}\left[\left\|\nabla f(x^k)\right\|^2\right] \leq \varepsilon.$$

$\square$

### D.4.3 Proof of Theorems 7.5

**Theorem 7.5.** *Consider Theorem 7.4. We assume that $i^{th}$ worker returns a stochastic gradient every $\tau_i$ seconds for all $i \in [n]$. Without loss of generality, we assume that $0 < \tau_1 \leq \cdots \leq \tau_n$. Then after*

$$96 \times \min_{m \in [n]} \left[ \left( \frac{1}{m} \sum_{i=1}^{m} \frac{1}{\tau_i} \right)^{-1} \left( \frac{L\Delta}{\varepsilon} + \frac{\sigma^2 L\Delta}{m\varepsilon^2} \right) \right] \tag{10}$$

*seconds, Method 4 guarantees to find an $\varepsilon$-stationary point.*

*Proof.* In this setup, the method converges after $K \times \{$*time required to collect a batch of the size $S\}$*. Without loss of generality, we assume that $\tau_1 \leq \cdots \leq \tau_n$.

Let us define time that is enough to collect a batch of the size $S$ as $t'$. Obviously, one can always take $t' = 3\tau_n \lceil \frac{S}{n} \rceil$ and guarantees that every worker calculates at least $\lceil \frac{S}{n} \rceil$ stochastic gradients, but we will provide a tighter $t'$.

We define $B_i$ as the number of received gradients with an iteration index equals to $k$.[7] For each worker, there are two options: either the $i^{th}$ worker does not send a gradient with an iteration index $k$ and $B_i = 0$, or it sends at least once and $B_i > 0$.

In the worst case, for $i^{th}$ worker, the time required to calculate $B_i$ gradients equals

$$t_i := \begin{cases} \tau_i (1 + B_i) & B_i > 0 \\ 0 & B_i = 0 \end{cases}$$

because either $B_i > 0$ and, in the worst case, a worker finishes the calculation of a gradient from the previous iteration (that we ignore) and only then starts the calculation of a gradient of the current iteration $k$, or $B_i = 0$ and the server does not receive any gradients from a worker.

Note that all workers work in parallel, so our goal is to find feasible points $t' \in \mathbb{R}$ and $B_1, \cdots, B_n \in \mathbb{N}_0$ such that

$$
\begin{aligned}
& t' \geq \max_{i \in [n]} t_i \\
& B_1, \cdots, B_n \in \mathbb{N}_0 \\
& \sum_{i=1}^{n} B_i \geq S
\end{aligned}
\tag{27}
$$

First, we relax an assumption that $B_i \in \mathbb{N}_0$ and assume that $B_i \in \mathbb{R}$ for all $i \in [n]$ :

$$
\begin{aligned}
& t' \geq \max_{i \in [n]} t_i \\
& B_1, \cdots, B_n \in \mathbb{R} \\
& B_1, \cdots, B_n \geq 0 \\
& \sum_{i=1}^{n} B_i \geq S
\end{aligned}
\tag{28}
$$

If $B_i \in \mathbb{R}$ are feasible points of (28), then

$$\max_{i \in [n]} t_i = \max_{B_i > 0} \tau_i (1 + B_i) \leq \max_{B_i > 0} \tau_i (1 + \lceil B_i \rceil)$$
$$\leq \max_{B_i > 0} \tau_i (2 + B_i) \leq 2 \max_{B_i > 0} \tau_i (1 + B_i) = 2 \max_{i \in [n]} t_i.$$

It means that if $t' \in \mathbb{R}$ and $B_1, \cdots, B_n$ are feasible points of (28), then $2t'$ and $\lceil B_1 \rceil, \cdots, \lceil B_n \rceil$ are feasible points of (27).

Let us define

$$t'(j) := \left( \sum_{i=1}^{j} \frac{1}{\tau_i} \right)^{-1} (S + j) \quad \forall j \in [n],$$

---

[7]Note that a worker may send a gradient from the previous iterations that we ignore in the method.

and take $j^* = \arg\min_{j \in [n]} t'(j)$, $j^*$ is the smallest index from all minimizers of $t'(j)$. Let us show that $t'(j^*)$ and

$$B_i = \begin{cases} \frac{t'(j^*)}{\tau_i} - 1, & i \le j^* \\ 0, & i > j^* \end{cases}$$

are feasible points of (28). First, we have

$$\sum_{i=1}^{n} B_i = \sum_{i=1}^{j^*} \left( \frac{t'(j^*)}{\tau_i} - 1 \right) = \left( \sum_{i=1}^{j^*} \frac{1}{\tau_i} \right)^{-1} (S + j^*) \left( \sum_{i=1}^{j^*} \frac{1}{\tau_i} \right) - j^* = S.$$

Next, we show that $B_i > 0$ for all $i \le j^*$. If $j^* = 1$, then $t'(1) = \tau_1(S + 1)$, thus $B_1 = S > 0$. If $j^* > 1$, then, by its definition, we have $t'(j^*) < t'(j^* - 1)$, thus

$$\left( \sum_{i=1}^{j^*} \frac{1}{\tau_i} \right)^{-1} (S + j^*) < \left( \sum_{i=1}^{j^*-1} \frac{1}{\tau_i} \right)^{-1} (S + j^* - 1).$$

From this inequality, we get

$$\left( \sum_{i=1}^{j^*-1} \frac{1}{\tau_i} \right) (S + j^*) < \left( \sum_{i=1}^{j^*} \frac{1}{\tau_i} \right) (S + j^* - 1)$$

and

$$\left( \sum_{i=1}^{j^*} \frac{1}{\tau_i} \right) < \frac{1}{\tau_{j^*}} (S + j^*).$$

From the last inequality, we get that $\tau_{j^*} < t'(j^*)$, thus $B_i \ge B_{j^*} > 0$ for all $i \le j^*$. It is left to show that

$$\max_{i \in [n]} t_i = \max_{i \le j^*} \tau_i (B_i + 1) = t'(j^*).$$

Finally, we can conclude that Method 4 returns a solution after

$$K \times 2t'(j^*) = \frac{48 \Delta L}{\varepsilon} \min_{j \in [n]} \left[ \left( \sum_{i=1}^{j} \frac{1}{\tau_i} \right)^{-1} (S + j) \right]$$

seconds. $\qquad\square$

# E    Proofs for Heterogeneous Regime

## E.1    Proof of Theorem A.2

**Theorem A.2.** *Let us consider the oracle class $\mathcal{O}_{\tau_1,\ldots,\tau_n}^{\sigma^2, heterog}$ for some $\sigma^2 > 0$ and $0 < \tau_1 \le \cdots \le \tau_n$. We fix any $L, \Delta > 0$ and $0 < \varepsilon \le c'L\Delta$. In the view Protocol 3, for any algorithm $A \in \mathcal{A}_{zr}$, there exists a function $f = \frac{1}{n} \sum_{i=1}^{n} f_i \in \mathcal{F}_{\Delta, L}$ and oracles and distributions $((O_1, \ldots, O_n), (\mathcal{D}_1, \ldots, \mathcal{D}_n)) \in \mathcal{O}_{\tau_1,\ldots,\tau_n}^{\sigma^2, heterog}(f_1, \ldots, f_n)$ such that $\mathbb{E}\left[ \inf_{k \in S_t} \left\| \nabla f(x^k) \right\|^2 \right] > \varepsilon$, where $S_t := \left\{ k \in \mathbb{N}_0 \big| t^k \le t \right\}$, and*

$$t = c \times \left( \tau_n \frac{L\Delta}{\varepsilon} + \left( \frac{1}{n} \sum_{i=1}^{n} \tau_i \right) \frac{\sigma^2 L \Delta}{n \varepsilon^2} \right). \tag{14}$$

*The quantity $c'$ and $c$ are universal constants.*

The structure of the following proof is similar to the proof of Theorem 6.4. In the heterogeneous regime, the main difference is that we have more freedom to choose the functions $f_i$.

*Proof.* In (14), we have the sum of two terms. We split the proof in two parts for each of the terms.
(**Part 1**)
(**Step 1**: $f \in \mathcal{F}_{\Delta, L}$)
Let us fix $\lambda > 0$. We consider the following functions $f_i$ :

$$f_i(x) := \begin{cases} 0, & i < n, \\ \frac{nL\lambda^2}{l_1} F_T\left(\frac{x}{\lambda}\right), & i = n. \end{cases}$$

Let us show that the function $f$ is $L$-smooth:

$$\|\nabla f(x) - \nabla f(y)\| = \frac{1}{n} \left\| \sum_{i=1}^{n} (\nabla f_i(x) - \nabla f_i(y)) \right\| = \frac{L\lambda}{l_1} \left\| \nabla F_T\left(\frac{x}{\lambda}\right) - \nabla F_T\left(\frac{y}{\lambda}\right) \right\| \leq L\|x - y\|.$$

Let us take

$$T = \left\lfloor \frac{\Delta l_1}{L\lambda^2 \Delta^0} \right\rfloor,$$

then

$$f(0) - \inf_{x \in \mathbb{R}^T} f(x) = \frac{1}{n} \frac{nL\lambda^2}{l_1} (F_T(0) - \inf_{x \in \mathbb{R}^T} F_T(x)) \leq \frac{L\lambda^2 \Delta^0 T}{l_1} \leq \Delta.$$

We showed that the function $f \in \mathcal{F}_{\Delta, L}$.
(**Step 2**: Oracle Class)

In the oracles $O_i$, we have the freedom to choose a mapping $\widehat{\nabla} f_i(\cdot; \cdot)$ (see (7)). In this part of the proof, we simply take non-stochastic mappings $\widehat{\nabla} f_i(x; \xi) := \nabla f_i(x)$ that are, obviously, unbiased and $\sigma^2$-variance-bounded. We can take an arbitrary distribution, for instance, let us take $\mathcal{D}_i = \text{Bernouilli}(1)$ for all $i \in [n]$.

(**Step 3**: Analysis of Protocol)

We take

$$\lambda = \frac{l_1 \sqrt{\varepsilon}}{L}$$

to ensure that

$$\|\nabla f(x)\|^2 = \frac{1}{n^2} \|\nabla f_n(x)\|^2 = \frac{L^2 \lambda^2}{l_1^2} \left\| \nabla F_T\left(\frac{x}{\lambda}\right) \right\|^2 > \frac{L^2 \lambda^2}{l_1^2} = \varepsilon$$

for all $x \in \mathbb{R}^T$ such that $\text{prog}(x) < T$. Thus

$$T = \left\lfloor \frac{\Delta L}{l_1 \varepsilon \Delta^0} \right\rfloor.$$

Only the $n^{\text{th}}$ worker contains a nonzero function and can provide a gradient every $\tau_n$ seconds. Since $A$ is a zero-respecting algorithm and the function $f_n$ is a zero-chain function, for all $k \geq 0$ such that

$$t^k < \tau_n T,$$

we have

$$\|\nabla f(x^k)\| > \varepsilon$$

because we need at least $T$ oracle calls to obtain $\text{prog}(x^k) \geq T$. It means that

$$\inf_{k \in S_t} \|\nabla f(x^k)\|^2 > \varepsilon$$

for

$$t = \tau_n \left( \frac{\Delta L}{l_1 \varepsilon \Delta^0} - 1 \right).$$

We now prove the second part of the lower bound.
(**Part 2**)
(**Step 1**: $f \in \mathcal{F}_{\Delta, L}$)

Let us fix $\lambda > 0$. We assume that $x = [x_1, \ldots, x_n] \in \mathbb{R}^{nT}$. We define $x_i \in \mathbb{R}^T$ as the $i^{\text{th}}$ block of a vector $x = [x_1, \ldots, x_n] \in \mathbb{R}^{nT}$. We consider the following functions $f_i$ :

$$f_i(x) := \frac{nL\lambda_i^2}{l_1} F_T\left(\frac{x_i}{\lambda_i}\right).$$

The function $f_i$ depends only on a subset of variables $x_i$ from $x$. Let us show that the function $f$ is $L$-smooth. Indeed, we have

$$\|\nabla f_i(x) - \nabla f_i(y)\| = \frac{nL\lambda_i}{l_1}\left\|\nabla F_T\left(\frac{x_i}{\lambda_i}\right) - \nabla F_T\left(\frac{y_i}{\lambda_i}\right)\right\| \leq nL\|x_i - y_i\| \quad \forall i \in [n],$$

and

$$\|\nabla f(x) - \nabla f(y)\|^2 = \frac{1}{n^2}\left\|\sum_{i=1}^n (\nabla f_i(x) - \nabla f_i(y))\right\|^2 = \frac{1}{n^2}\sum_{i=1}^n \|\nabla f_i(x) - \nabla f_i(y)\|^2$$

$$\leq \frac{1}{n^2}\sum_{i=1}^n n^2 L^2 \|x_i - y_i\|^2 = L^2 \|x - y\|^2.$$

Let us take

$$T = \left\lfloor \frac{\Delta l_1}{L\sum_{j=1}^n \lambda_j^2 \Delta^0} \right\rfloor \quad,$$

then

$$f(0) - \inf_{x \in \mathbb{R}^T} f(x) = \frac{1}{n}\sum_{i=1}^n \frac{nL\lambda_i^2}{l_1}\left(F_T(0) - \inf_{x \in \mathbb{R}^T} F_T(x)\right) \leq \sum_{i=1}^n \frac{L\lambda_i^2 \Delta^0 T}{l_1} \leq \Delta.$$

We showed that the function $f \in \mathcal{F}_{\Delta,L}$.

(**Step 2**: Oracle Class)

In the oracles $O_i$, we have the freedom to choose a mapping $\widehat{\nabla} f_i(\cdot; \cdot)$ (see (7)). Let us take

$$[\widehat{\nabla} f_i(x; \xi)]_j := \nabla_j f_i(x)\left(1 + \mathbb{1}\left[j > (i-1)T + \text{prog}(x_i)\right]\left(\frac{\xi}{p_i} - 1\right)\right) \quad \forall x \in \mathbb{R}^{nT},$$

$\mathcal{D}_i = \text{Bernouilli}(p_i)$, and $p_i \in (0, 1]$ for all $i \in [n]$. Let us show it is unbiased and $\sigma^2$-variance-bounded:

$$\mathbb{E}\left[[\widehat{\nabla} f_i(x; \xi)]_j\right] = \nabla_j f_i(x)\left(1 + \mathbb{1}\left[j > (i-1)T + \text{prog}(x_i)\right]\left(\frac{\mathbb{E}[\xi]}{p_i} - 1\right)\right) = \nabla_j f_i(x)$$

for all $j \in nT$, and

$$\mathbb{E}\left[\left\|\widehat{\nabla} f_i(x; \xi) - \nabla f_i(x)\right\|^2\right] \leq \|\nabla f_i(x)\|_\infty^2 \, \mathbb{E}\left[\left(\frac{\mathbb{E}[\xi]}{p_i} - 1\right)^2\right]$$

because the difference is non-zero only in one coordinate. Thus

$$\mathbb{E}\left[\left\|\widehat{\nabla} f_i(x; \xi) - \nabla f_i(x)\right\|^2\right] \leq \frac{\|\nabla f_i(x)\|_\infty^2 (1 - p_i)}{p_i} = \frac{n^2 L^2 \lambda_i^2 \left\|F_T\left(\frac{x_i}{\lambda_i}\right)\right\|_\infty^2 (1 - p_i)}{l_1^2 p_i}$$

$$\leq \frac{n^2 L^2 \lambda_i^2 \gamma_\infty^2 (1 - p_i)}{l_1^2 p_i} \leq \sigma^2,$$

where we take

$$p_i = \min\left\{\frac{n^2 L^2 \lambda_i^2 \gamma_\infty^2}{\sigma^2 l_1^2}, 1\right\}.$$

(**Step 3**: Analysis of Protocol)

We fix $\eta > 0$ and choose

$$\lambda_i = \frac{l_1 \sqrt{\eta \varepsilon \tau_i}}{L \sqrt{\sum_{i=1}^{n} \tau_i}}$$

to ensure that

$$\|\nabla f(x)\|^2 = \frac{1}{n^2} \sum_{i=1}^{n} \|\nabla f_i(x)\|^2 = \sum_{i=1}^{n} \frac{L^2 \lambda_i^2}{l_1^2} \left\|\nabla F_{T_i}\left(\frac{x_i}{\lambda_i}\right)\right\|^2$$

$$= \sum_{i=1}^{n} \frac{\eta \varepsilon \tau_i}{\sum_{i=1}^{n} \tau_i} \left\|\nabla F_{T_i}\left(\frac{x_i}{\lambda_i}\right)\right\|^2 > \sum_{i=1}^{n} \frac{\eta \varepsilon \tau_i}{\sum_{i=1}^{n} \tau_i} \mathbb{1}[\mathrm{prog}(x_i) < T] \quad (29)$$

for all $x = [x_1, \ldots, x_n] \in \mathbb{R}^T$. Thus

$$T = \left\lfloor \frac{\Delta L}{\eta \varepsilon l_1 \Delta^0} \right\rfloor$$

and

$$p_i = \min\left\{ \frac{n^2 \gamma_\infty^2 \eta \varepsilon \tau_i}{\sigma^2 \sum_{i=1}^{n} \tau_i}, 1 \right\} \quad \forall i \in [n]. \quad (30)$$

Protocol 3 generates the sequence $\{x^k\}_{k=0}^{\infty} \equiv \{[x_1^k, \ldots, x_n^k]\}_{k=0}^{\infty}$. From (29), we have

$$\inf_{k \in S_t} \left\|\nabla f(x^k)\right\|^2 > \inf_{k \in S_t} \sum_{i=1}^{n} \frac{\eta \varepsilon \tau_i}{\sum_{i=1}^{n} \tau_i} \mathbb{1}[\mathrm{prog}(x_i^k) < T] \geq \sum_{i=1}^{n} \frac{\eta \varepsilon \tau_i}{\sum_{i=1}^{n} \tau_i} \inf_{k \in S_t} \mathbb{1}[\mathrm{prog}(x_i^k) < T]. \quad (31)$$

Further, we require the following auxillary lemma. See the proof in Section E.2.

**Lemma E.1.** *For $\eta = 4$, with probability not less than $1/2$,*

$$\sum_{i=1}^{n} \frac{\eta \varepsilon \tau_i}{\sum_{i=1}^{n} \tau_i} \inf_{k \in S_t} \mathbb{1}[\mathrm{prog}(x_i^k) < T] > 2\varepsilon$$

*for*

$$t \leq \frac{1}{24} \left(\frac{\sigma^2 \sum_{i=1}^{n} \tau_i}{n^2 \gamma_\infty^2 \eta \varepsilon}\right) \left(\frac{T}{2} - 1\right).$$

Using Lemma E.1 and (31), we have

$$\mathbb{E}\left[\inf_{k \in S_t} \left\|\nabla f(x^k)\right\|^2\right] > \varepsilon$$

for

$$t = \frac{1}{24} \left(\frac{\sigma^2 \sum_{i=1}^{n} \tau_i}{4n^2 \gamma_\infty^2 \varepsilon}\right) \left(\frac{\Delta L}{8\varepsilon l_1 \Delta^0} - 2\right).$$

This finishes the proof of Part 2.

$\square$

### E.2 Proof of Lemma E.1

In the following lemma, we use notations from the proof of Theorem A.2.

**Lemma E.1.** *For $\eta = 4$, with probability not less than $1/2$,*

$$\sum_{i=1}^{n} \frac{\eta \varepsilon \tau_i}{\sum_{i=1}^{n} \tau_i} \inf_{k \in S_t} \mathbb{1}[\mathrm{prog}(x_i^k) < T] > 2\varepsilon$$

*for*

$$t \leq \frac{1}{24} \left(\frac{\sigma^2 \sum_{i=1}^{n} \tau_i}{n^2 \gamma_\infty^2 \eta \varepsilon}\right) \left(\frac{T}{2} - 1\right).$$

*Proof.* Let us fix $t \geq 0$. Our goal is to show that the probability of an inequality

$$\sum_{i=1}^{n} \frac{\eta \varepsilon \tau_i}{\sum_{i=1}^{n} \tau_i} \inf_{k \in S_t} \mathbb{1}[\text{prog}(x_i^k) < T] \leq 2\varepsilon \tag{32}$$

is small.

We now use the same reasoning as in Lemma D.2. We use a notation $\{x\}_i$ is $i^{\text{th}}$ block of the vector $x$. Let us fix a worker's index $i \in [n]$.

**Definition E.2** (Sequence $k_{i,j,l}^{\xi}$). Let us consider a set

$$\left\{ k \in \mathbb{N} \, | \, s_{i^k,q}^{k-1} = 1, t^k \geq s_{i^k,t}^{k-1} + \tau_{i^k}, \text{prog}\left(\left\{s_{i^k,x}^{k-1}\right\}_i\right) = j, i^k = i \right\}, \quad s_{i^k}^{k-1} \equiv (s_{i^k,t}^{k-1}, s_{i^k,q}^{k-1}, s_{i^k,x}^{k-1}).$$

We order this set and define the result sequence as $\{k_{i,j,l}^{\xi}\}_{i=1}^{m_{i,j+1}}$, where $m_{i,j+1} \in [0, \infty]$ is the size of the sequence. The sequence $k_{i,j,l}^{\xi}$ is a subsequence of iterations where the $i^{\text{th}}$ oracle use the generated Bernouilli random variables in (7) when $\text{prog}(\{s_x\}_i) = j$. The sequence $s_{i^k}^{k-1}$ is defined in Protocol 3.

Then

$$\eta_{i,j+1} := \inf\{l \, | \, \xi^{k_{i,j,l}^{\xi}} = 1 \text{ and } l \in [1, m_{i,j+1}]\} \in \mathbb{N} \cup \{\infty\}.$$

The quantity $\eta_{i,j+1}$ is the index of the first successful trial, when $\text{prog}(\cdot) = j$ in the $i^{\text{th}}$ block of the sequence $x^k$. Since the algorithm $A$ is a zero-respecting algorithm, for all $k < k_{i,j,\eta_{i,j+1}}^{\xi}$, the progress $\text{prog}(x_i^k) < j + 1$.

As in Lemma D.2 (we skip the proof since the idea is the same. It is only required to use the different notations: $x^k \to x_i^k, \hat{t}_{\eta_j} \to \hat{t}_{i,\eta_{i,j}}$), for all $i \in [n]$, one can show that if $\inf_{k \in S_t} \mathbb{1}[\text{prog}(x_i^k) < T] < 1$ holds, then $\sum_{j=1}^{T} \hat{t}_{i,\eta_{i,j}} \leq t$, where $\hat{t}_{i,k} := k\tau_i$ for all $k \geq 1$. The time $\hat{t}_{i,k}$ is the smallest possible time when the $i^{\text{th}}$ oracle can return the $k^{\text{th}}$ stochastic gradient. Thus

$$\mathbb{P}\left(\sum_{i=1}^{n} \frac{\eta \varepsilon \tau_i}{\sum_{i=1}^{n} \tau_i} \inf_{k \in S_t} \mathbb{1}[\text{prog}(x_i^k) < T] \leq 2\varepsilon\right) \leq \mathbb{P}\left(\sum_{i=1}^{n} \frac{\eta \varepsilon \tau_i}{\sum_{i=1}^{n} \tau_i} \mathbb{1}\left[\sum_{j=1}^{T} \hat{t}_{i,\eta_{i,j}} \leq t\right] \leq 2\varepsilon\right).$$

Using (25) with $n = 1$ (and the different notations $p \to p_i, \tau_1 \to \tau_i$, and $\hat{t}_{\eta_j} \to \hat{t}_{i,\eta_{i,j}}$), we have

$$\mathbb{P}\left(\sum_{j=1}^{T} \hat{t}_{i,\eta_{i,j}} \leq t\right) \leq \delta \quad \forall t \leq \frac{\tau_i}{24 p_i}\left(\frac{T}{2} + \log \delta\right). \tag{33}$$

We now rearrange the terms and use Markov's inequality to obtain

$$\mathbb{P}\left(\sum_{i=1}^{n} \frac{\eta \varepsilon \tau_i}{\sum_{i=1}^{n} \tau_i} \mathbb{1}\left[\sum_{j=1}^{T} \hat{t}_{i,\eta_{i,j}} > t\right] \leq 2\varepsilon\right)$$

$$= \mathbb{P}\left(\sum_{i=1}^{n} \tau_i \mathbb{1}\left[\sum_{j=1}^{T} \hat{t}_{i,\eta_{i,j}} \leq t\right] \geq \left(1 - \frac{2}{\eta}\right)\sum_{i=1}^{n} \tau_i\right)$$

$$\leq \left(1 - \frac{2}{\eta}\right)^{-1}\left(\sum_{i=1}^{n} \tau_i\right)^{-1} \mathbb{E}\left[\sum_{i=1}^{n} \tau_i \mathbb{1}\left[\sum_{j=1}^{T} \hat{t}_{i,\eta_{i,j}} \leq t\right]\right]$$

$$= \left(1 - \frac{2}{\eta}\right)^{-1}\left(\sum_{i=1}^{n} \tau_i\right)^{-1} \sum_{i=1}^{n} \tau_i \mathbb{P}\left(\sum_{j=1}^{T} \hat{t}_{i,\eta_{i,j}} \leq t\right),$$

for $\eta > 2$. Using the choice of $p_i$ in (30), we have

$$\frac{\tau_i}{p_i} \geq \frac{\sigma^2 \sum_{i=1}^{n} \tau_i}{n^2 \gamma_\infty^2 \eta \varepsilon}$$

for all $i \in [n]$. The last term does not depend on $i$. Therefore, we can use (33) with

$$t \leq \frac{1}{24} \left( \frac{\sigma^2 \sum_{i=1}^n \tau_i}{n^2 \gamma_\infty^2 \eta \varepsilon} \right) \left( \frac{T}{2} + \log \delta \right)$$

to get

$$\mathbb{P} \left( \sum_{i=1}^n \frac{\eta \varepsilon \tau_i}{\sum_{i=1}^n \tau_i} \mathbb{1} \left[ \sum_{j=1}^T \widehat{t}_{i,\eta_{i,j}} > t \right] \leq 2\varepsilon \right) \leq \left( 1 - \frac{2}{\eta} \right)^{-1} \left( \sum_{i=1}^n \tau_i \right)^{-1} \left( \sum_{i=1}^n \tau_i \right) \delta = \left( 1 - \frac{2}{\eta} \right)^{-1} \delta.$$

Finally, for $\eta = 4$ and $\delta = 1/4$, we have

$$\mathbb{P} \left( \sum_{i=1}^n \frac{\eta \varepsilon \tau_i}{\sum_{i=1}^n \tau_i} \inf_{k \in S_t} \mathbb{1}[\text{prog}(x_i^k) < T] \leq 2\varepsilon \right) \leq \frac{1}{2}.$$

$\square$

### E.3   Proof of Theorem A.3

**Theorem A.3.** *Assume that Assumptions 7.1 and 7.2 hold for the function $f$. Assumption 7.3 holds for the function $f_i$ for all $i \in [n]$. Let us take the parameter $S = \max \left\{ \lceil \sigma^2 / \varepsilon \rceil, n \right\}$, and $\gamma = \min \left\{ \frac{1}{L}, \frac{\varepsilon S}{2L\sigma^2} \right\} = \Theta(1/L)$ in Method 6, then after $K \geq 24\Delta L / \varepsilon$ iterations the method guarantees that $\frac{1}{K} \sum_{k=0}^{K-1} \mathbb{E} \left[ \|\nabla f(x^k)\|^2 \right] \leq \varepsilon.$*

*Proof.* Note that Method 6 can be rewritten as $x^{k+1} = x^k - \gamma \frac{1}{n} \sum_{i=1}^n \frac{1}{B_i} \sum_{j=1}^{B_i} \widehat{\nabla} f_i(x^k; \xi_{i,j})$, where the $\xi_{i,j}$ are independent random samples. The variance of the gradient estimator equals

$$\mathbb{E} \left[ \left\| \frac{1}{n} \sum_{i=1}^n \frac{1}{B_i} \sum_{j=1}^{B_i} \widehat{\nabla} f_i(x^k; \xi_{i,j}) - \nabla f(x^k) \right\|^2 \right]$$

$$= \frac{1}{n^2} \sum_{i=1}^n \mathbb{E} \left[ \left\| \frac{1}{B_i} \sum_{j=1}^{B_i} \widehat{\nabla} f_i(x^k; \xi_{i,j}) - \nabla f_i(x^k) \right\|^2 \right]$$

$$= \frac{1}{n^2} \sum_{i=1}^n \frac{1}{B_i^2} \sum_{j=1}^{B_i} \mathbb{E} \left[ \left\| \widehat{\nabla} f_i(x^k; \xi_{i,j}) - \nabla f_i(x^k) \right\|^2 \right] \leq \frac{1}{n^2} \sum_{i=1}^n \frac{\sigma^2}{B_i} = \left( \frac{1}{n} \sum_{i=1}^n \frac{1}{B_i} \right) \frac{\sigma^2}{n} \leq \frac{\sigma^2}{S},$$

where we use the inequality $\left( \frac{1}{n} \sum_{i=1}^n \frac{1}{B_i} \right)^{-1} \geq \frac{S}{n}$, We can use the classical SGD result (see Theorem D.8). For a stepsize

$$\gamma = \min \left\{ \frac{1}{L}, \frac{\varepsilon S}{2L\sigma^2} \right\},$$

we have

$$\frac{1}{K} \sum_{k=0}^{K-1} \mathbb{E} \left[ \|\nabla f(x^k)\|^2 \right] \leq \varepsilon,$$

if

$$K \geq \frac{12\Delta L}{\varepsilon} + \frac{12\Delta L \sigma^2}{\varepsilon^2 S}.$$

Using the choice of $S$, we obtain that Method 6 converges after

$$K \geq \frac{24\Delta L}{\varepsilon}$$

steps.

$\square$

## E.4 Proof of Theorem A.4

**Theorem A.4.** *Let us consider Theorem A.3. We assume that $i^{th}$ worker returns a stochastic gradient every $\tau_i$ seconds for all $i \in [n]$. Without loss of generality, we assume that $0 < \tau_1 \leq \cdots \leq \tau_n$. Then after*

$$96 \left( \tau_n \frac{L\Delta}{\varepsilon} + \left( \frac{1}{n} \sum_{i=1}^{n} \tau_i \right) \frac{\sigma^2 L\Delta}{n\varepsilon^2} \right) \tag{15}$$

*seconds, Method 6 guarantees to find an $\varepsilon$-stationary point.*

*Proof.* The method converges after $K \times \{$*time required to collect batches with the sizes $B_i$ such that* $\left( \frac{1}{n} \sum_{i=1}^{n} 1/B_i \right)^{-1} \geq \frac{S}{n}$ *holds*$\}$.

In the worst case, for $i^{\text{th}}$ worker, the time required to calculate $B_i$ gradients equals

$$t_i := \tau_i \left( 1 + B_i \right)$$

because it is possible that a worker finishes the calculation of a gradient from the previous iteration (that we ignore) and only then starts the calculation of a gradient of the current iteration.

Our goal is to find feasible points $t' \in \mathbb{R}$ and $B_1, \cdots, B_n \in \mathbb{N}$ such that

$$t' \geq \max_{i \in [n]} t_i,$$

$$B_1, \cdots, B_n \in \mathbb{N}, \tag{34}$$

$$\left( \frac{1}{n} \sum_{i=1}^{n} \frac{1}{B_i} \right)^{-1} \geq \frac{S}{n}.$$

Using the same reasoning as in Theorem 7.5, we relax the assumption that $B_i \in \mathbb{N}$ and assume that $B_i \in \mathbb{R}$ for all $i \in [n]$ :

$$t' \geq \max_{i \in [n]} t_i$$

$$B_1, \cdots, B_n \in \mathbb{R} \tag{35}$$

$$B_1, \cdots, B_n > 0$$

$$\left( \frac{1}{n} \sum_{i=1}^{n} \frac{1}{B_i} \right)^{-1} \geq \frac{S}{n},$$

If $t' \in \mathbb{R}$ and $B_1, \cdots, B_n$ are feasible points of (35), then $2t'$ and $\lceil B_1 \rceil, \cdots, \lceil B_n \rceil$ are feasible points of (34). Let us show that $t' = 2 \left( \tau_n + \left( \frac{1}{n} \sum_{i=1}^{n} \tau_i \right) \frac{S}{n} \right)$ and $B_i = \frac{t'}{\tau_i} - 1$ are feasible points. Indeed, for all $i \in [n]$,

$$B_i = \frac{t'}{\tau_i} - 1 \geq \frac{2\tau_n}{\tau_i} - 1 \geq 1.$$

Next, we have

$$\max_{i \in [n]} t_i = \max_{i \in [n]} \tau_i \left( B_i + 1 \right) = t'$$

and

$$\left( \frac{1}{n} \sum_{i=1}^{n} \frac{1}{B_i} \right)^{-1} = \left( \frac{1}{n} \sum_{i=1}^{n} \frac{\tau_i}{t' - \tau_i} \right)^{-1} \geq \left( \frac{1}{n} \sum_{i=1}^{n} \frac{2\tau_i}{t'} \right)^{-1} = \frac{t'}{2} \left( \frac{1}{n} \sum_{i=1}^{n} \tau_i \right)^{-1} \geq \frac{S}{n},$$

where we use $t' - \tau_i \geq t'/2 + \tau_i - \tau_i = t'/2$ for all $i \in [n]$. Finally, it means that Method 6 returns a solution after

$$K \times 2t' = \frac{96\Delta L}{\varepsilon} \left( \tau_n + \left( \frac{1}{n} \sum_{i=1}^{n} \tau_i \right) \frac{S}{n} \right)$$

seconds. $\qquad \square$

# F   Interrupt Oracle Calculations

Let us define a protocol and an oracle where an algorithm can stop the oracle anytime. If an algorithm stops the oracle, its current calculations are canceled and discarded.

---

**Protocol 8** Time Multiple Oracles Protocol With Control

---

1: **Input:** functions $f \in \mathcal{F}$, oracles and distributions $((O_1, \ldots, O_n), (\mathcal{D}_1, \ldots, \mathcal{D}_n)) \in \mathcal{O}(f)$, algorithm $A \in \mathcal{A}$
2: $s_i^0 = 0$ for all $i \in [n]$
3: **for** $k = 0, \ldots, \infty$ **do**
4: $\quad (t^{k+1}, i^{k+1}, c^k, x^k) = A^k(g^1, \ldots, g^k),$ $\qquad\qquad\qquad \rhd t^{k+1} \geq t^k$
5: $\quad (s_{i^{k+1}}^{k+1}, g^{k+1}) = O_{i^{k+1}}(t^{k+1}, x^k, c^k, s_{i^{k+1}}^k, \xi^{k+1}), \quad \xi^{k+1} \sim \mathcal{D} \quad \rhd s_j^{k+1} = s_j^k \quad \forall j \neq i^{k+1}$
6: **end for**

---

In Protocol 8, we allow algorithms to output the control variables $c^k$ that can be used in the following oracle.

We take an oracle

$$O_\tau^{\widehat{\nabla} f} : \mathbb{R}_{\geq 0} \times \mathbb{R}^d \times \underbrace{(\mathbb{R}_{\geq 0} \times \mathbb{R}^d \times \{0,1\})}_{\text{input state}} \times \underbrace{\{0,1\}}_{\text{control}} \times \mathbb{S}_\xi \to \underbrace{(\mathbb{R}_{\geq 0} \times \mathbb{R}^d \times \{0,1\})}_{\text{output state}} \times \mathbb{R}^d$$

such that

$$O_\tau^{\widehat{\nabla} f}(t, x, (s_t, s_x, s_q), c, \xi) = \begin{cases} ((t, x, 1), & 0), & c = 0 \text{ and } s_q = 0, \\ ((s_t, s_x, 1), & 0), & c = 0 \text{ and } s_q = 1 \text{ and } t < s_t + \tau, \\ ((0,0,0), & \widehat{\nabla} f(s_x; \xi)), & c = 0 \text{ and } s_q = 1 \text{ and } t \geq s_t + \tau, \\ ((0,0,0), & 0), & c = 1, \end{cases}$$
(36)

and $\widehat{\nabla} f$ is a mapping such that

$$\widehat{\nabla} f : \mathbb{R}^d \times \mathbb{S}_\xi \to \mathbb{R}^d.$$

The oracle (36) generalizes the oracle (7) since an algorithm can send a signal $c$ to the oracle (36) and interrupt the calculations. Note that if $c = 1$, then (36) has the same behavior as (7). But, if $c = 0$, then the oracle (36) discards all previous information in the state, and changes $s_q$ to 0.

Let us define an oracle class:

**Definition F.1** (Oracle Class $\mathcal{O}_{\tau_1, \ldots, \tau_n}^{\sigma^2, \text{stop}}$).
Let us consider an oracle class such that, for any $f \in \mathcal{F}_{\Delta, L}$, it returns oracles $O_i = O_{\tau_i}^{\widehat{\nabla} f}$ and distributions $\mathcal{D}_i$ for all $i \in [n]$, where $\widehat{\nabla} f$ is an unbiased $\sigma^2$-variance-bounded mapping (see Assumption 7.3). The oracles $O_{\tau_i}^{\widehat{\nabla} f}$ are defined in (36). We define such oracle class as $\mathcal{O}_{\tau_1, \ldots, \tau_n}^{\sigma^2, \text{stop}}$. Without loss of generality, we assume that $0 < \tau_1 \leq \cdots \leq \tau_n$.

For the oracle class $\mathcal{O}_{\tau_1, \ldots, \tau_n}^{\sigma^2, \text{stop}}$, we state that

$$\mathfrak{m}_{\text{time}}(\mathcal{A}_{\text{zr}}, \mathcal{F}_{\Delta, L}) = \Omega \left( \min_{m \in [n]} \left[ \left( \frac{1}{m} \sum_{i=1}^m \frac{1}{\tau_i} \right)^{-1} \left( \frac{L\Delta}{\varepsilon} + \frac{\sigma^2 L \Delta}{m \varepsilon^2} \right) \right] \right).$$

The lower bound is the same as for the oracle class from Definition 6.3. We do not provide a formal proof, but a close investigation can reveal that the proof is the same as in Theorem 6.4.

Indeed, in Part 1 of the proof of Lemma D.2, we reduce the the inequality $\inf_{k \in S_t} \mathbb{1}\left[\text{prog}(x^k) < T\right] < 1$ to the inequality $\sum_{i=1}^T \widehat{t}_{\eta_i} \leq t$, where $\widehat{t}_{\eta_i}$ is the shortest time when the oracles can draw a successful Bernouilli random variable. The fact that an algorithm can interrupt the oracles can not change the quantity $\widehat{t}_{\eta_i}$.

# G Time Complexity with Synchronized Start

In this section, we continue and fill up the discussion in Section 8.

Let us design an oracle for the synchronized start setting. We take an oracle

$$O_{\tau_1,\dots,\tau_n}^{\widehat{\nabla}f} \; : \; \mathbb{R}_{\geq 0} \times \mathbb{R}^d \times \underbrace{(\mathbb{R}_{\geq 0} \times \mathbb{R}^d \times \{0,1\})}_{\text{input state}} \times \underbrace{(\mathbb{S}_\xi \times \cdots \times \mathbb{S}_\xi)}_{n \text{ times}} \to \underbrace{(\mathbb{R}_{\geq 0} \times \mathbb{R}^d \times \{0,1\})}_{\text{output state}} \times \mathbb{R}^d$$

such that

$$
O_{\tau_1,\dots,\tau_n}^{\widehat{\nabla}f}(t, x, (s_t, s_x, s_q), (\xi_1, \dots, \xi_n)) =
\begin{cases}
((t, x, 1), & 0), & s_q = 0, \\
((0,0,0), & 0), & s_q = 1 \text{ and } t \in [0, s_t + \tau_1), \\
((0,0,0), & \widehat{\nabla}f(s_x; \xi_1)), & s_q = 1 \text{ and } t \in [s_t + \tau_1, s_t + \tau_2), \\
((0,0,0), & \displaystyle\sum_{i=1}^{2} \widehat{\nabla}f(s_x; \xi_i)), & s_q = 1 \text{ and } t \in [s_t + \tau_2, s_t + \tau_3), \\
\dots \\
((0,0,0), & \displaystyle\sum_{i=1}^{n} \widehat{\nabla}f(s_x; \xi_i)), & s_q = 1 \text{ and } t \in [s_t + \tau_n, \infty),
\end{cases}
\tag{37}
$$

and $\widehat{\nabla}f$ is a mapping such that

$$\widehat{\nabla}f \; : \; \mathbb{R}^d \times \mathbb{S}_\xi \to \mathbb{R}^d.$$

We assume that $\xi^{k+1}$ is a tuple in Protocol 2: $\xi^{k+1} \equiv (\xi_1^{k+1}, \dots, \xi_n^{k+1}) \sim \mathcal{D}$.

The oracle (37) with Protocol 2 emulates the behavior of a setting where we broadcast an iterate $x$ to all workers, and they start calculations simultaneously. The workers have different time delays, hence some finish earlier than others. An algorithm can stop the procedure earlier and get calculated stochastic gradients, but other non-calculated ones will be discarded.

For this oracle, we define an oracle class:

**Definition G.1** (Oracle Class $\mathcal{O}_{\tau_1,\dots,\tau_n}^{\sigma^2,\text{sync}}$)**.**
Let us consider an oracle class such that, for any $f \in \mathcal{F}_{\Delta,L}$, it return an oracle $O = \mathcal{O}_{\tau_1,\dots,\tau_n}^{\widehat{\nabla}f}$ and a distribution $\mathcal{D}$, where $\widehat{\nabla}f$ is an unbiased $\sigma^2$-variance-bounded mapping (see Assumption 7.3). The oracle $O_{\tau_1,\dots,\tau_n}^{\widehat{\nabla}f}$ is defined in (37). We define such oracle class as $\mathcal{O}_{\tau_1,\dots,\tau_n}^{\sigma^2,\text{sync}}$. Without loss of generality, we assume that $0 < \tau_1 \leq \cdots \leq \tau_n$.

We now provide the lower bound for Protocol 2 and the oracle class $\mathcal{O}_{\tau_1,\dots,\tau_n}^{\sigma^2,\text{sync}}$.

**Theorem G.2.** *Let us consider the oracle class $\mathcal{O}_{\tau_1,\dots,\tau_n}^{\sigma^2,\text{sync}}$ for some $\sigma^2 > 0$ and $0 < \tau_1 \leq \cdots \leq \tau_n$. We fix any $L, \Delta > 0$ and $0 < \varepsilon \leq c'L\Delta$. In the view Protocol 2, for any algorithm $A \in \mathcal{A}_{\text{zr}}$, there exists a function $f \in \mathcal{F}_{\Delta,L}$ and an oracle and a distribution $(O, \mathcal{D}) \in \mathcal{O}_{\tau_1,\dots,\tau_n}^{\sigma^2,\text{sync}}(f)$ such that $\mathbb{E}\left[\inf_{k \in S_t} \left\|\nabla f(x^k)\right\|^2\right] > \varepsilon$, where $S_t := \left\{k \in \mathbb{N}_0 \middle| t^k \leq t\right\}$, and*

$$t = c \times \min_{m \in [n]}\left[\tau_m\left(\frac{L\Delta}{\varepsilon} + \frac{\sigma^2 L\Delta}{m\varepsilon^2}\right)\right].$$

*The quantity $c'$ and $c$ are universal constants.*

## G.1 Minimax optimal method

In this section, we analyze the $m$–Minibatch SGD method (see Method 9). This method generalizes the Minibatch SGD method from Section 1.2. Unlike Minibatch SGD, the $m$–Minibatch SGD method only asks for stochastic gradients from the first $m \in [n]$ (fastest) workers. Later, we show that optimal

---

**Method 9** $m$–Minibatch SGD

---

1: **Input:** starting point $x^0$, stepsize $\gamma$, number of workers $m \in [n]$
2: **for** $k = 0, 1, \ldots, K - 1$ **do**
3:     Send the point $x^k$ to the first $m$ workers
4:     Receive i.i.d. stochastic gradients $\widehat{\nabla} f(x^k, \xi_1), \ldots, \widehat{\nabla} f(x^k, \xi_m)$ from the workers
5:     $x^{k+1} = x^k - \gamma \frac{1}{m} \sum_{i=1}^{m} \widehat{\nabla} f(x^k, \xi_i)$
6: **end for**

---

$m$ is determined by (38). And with this parameter, $m$–Minibatch SGD method is minimax optimal under the setting from Sections 8 and G. Note that $m$–Minibatch SGD is Minibatch SGD if $m = n$.

We now provide the convergence rate and the time complexity.

**Theorem G.3.** *Assume that Assumptions 7.1, 7.2 and 7.3 hold. Let us take the step size*

$$\gamma = \min \left\{ \frac{1}{L}, \frac{\varepsilon m}{2L\sigma^2} \right\}$$

*in Method 9, then after*

$$K \geq \frac{12\Delta L}{\varepsilon} + \frac{12\Delta L\sigma^2}{\varepsilon^2 m}$$

*iterations the method guarantees that $\frac{1}{K} \sum_{k=0}^{K-1} \mathbb{E}\left[ \left\| \nabla f(x^k) \right\|^2 \right] \leq \varepsilon$.*

**Theorem G.4.** *Let us consider Theorem G.3. We assume that $i^{th}$ worker returns a stochastic gradient every $\tau_i$ seconds for all $i \in [n]$. Without loss of generality, we assume that $0 < \tau_1 \leq \cdots \leq \tau_n$. Let us take*

$$m = \underset{m' \in [n]}{\arg\min} \, \tau_{m'} \left( 1 + \frac{\sigma^2}{m'\varepsilon} \right). \tag{38}$$

*Then after*

$$12 \min_{m \in [n]} \tau_m \left( \frac{L\Delta}{\varepsilon} + \frac{\sigma^2 L\Delta}{m\varepsilon^2} \right) \tag{39}$$

*seconds Method 9 guarantees to find an $\varepsilon$-stationary point.*

Despite the triviality of the $m$–Minibatch SGD and it analysis, we provide it to show that the lower bound in Theorem G.2 is tight.

## G.2    Proof of Theorem G.2

**Theorem G.2.** *Let us consider the oracle class $\mathcal{O}^{\sigma^2, \text{sync}}_{\tau_1, \ldots, \tau_n}$ for some $\sigma^2 > 0$ and $0 < \tau_1 \leq \cdots \leq \tau_n$. We fix any $L, \Delta > 0$ and $0 < \varepsilon \leq c'L\Delta$. In the view Protocol 2, for any algorithm $A \in \mathcal{A}_{\text{zr}}$, there exists a function $f \in \mathcal{F}_{\Delta, L}$ and an oracle and a distribution $(O, D) \in \mathcal{O}^{\sigma^2, \text{sync}}_{\tau_1, \ldots, \tau_n}(f)$ such that $\mathbb{E}\left[ \inf_{k \in S_t} \left\| \nabla f(x^k) \right\|^2 \right] > \varepsilon$, where $S_t := \left\{ k \in \mathbb{N}_0 \middle| t^k \leq t \right\}$, and*

$$t = c \times \min_{m \in [n]} \left[ \tau_m \left( \frac{L\Delta}{\varepsilon} + \frac{\sigma^2 L\Delta}{m\varepsilon^2} \right) \right].$$

*The quantity $c'$ and $c$ are universal constants.*

Step 1 and Step 2 mirrors the corresponding steps from the proof of Theorem G.2.

*Proof.* (**Step 1**: $f \in \mathcal{F}_{\Delta, L}$)
Let us fix $\lambda > 0$. We take the same function $f \in \mathcal{F}_{\Delta, L}$ as in the proof of Theorem 6.4. We define

$$f(x) := \frac{L\lambda^2}{l_1} F_T \left( \frac{x}{\lambda} \right)$$

with

$$T = \left\lfloor \frac{\Delta l_1}{L\lambda^2 \Delta^0} \right\rfloor.$$

(**Step 2**: Oracle Class)

Following the proof of Theorem 6.4, in the oracle $O$, we take the following stochastic estimator

$$[\widehat{\nabla} f(x; \xi)]_j := \nabla_j f(x) \left( 1 + \mathbb{1}\left[ j > \text{prog}(x) \right] \left( \frac{\xi}{p} - 1 \right) \right) \quad \forall x \in \mathbb{R}^T, \tag{40}$$

and $\mathcal{D} = \underbrace{(\text{Bernouilli}(p), \dots, \text{Bernouilli}(p))}_{n \text{ times}}$, where $p \in (0, 1]$. The stochastic gradient is unbiased and $\sigma^2$-variance-bounded if

$$p = \min \left\{ \frac{L^2 \lambda^2 \gamma_\infty^2}{\sigma^2 l_1^2}, 1 \right\}.$$

(**Step 3**: Analysis of Protocol)

We choose

$$\lambda = \frac{\sqrt{2\varepsilon} l_1}{L}$$

to ensure that $\|\nabla f(x)\|^2 = \frac{L^2 \lambda^2}{l_1^2} \left\| \nabla F_T(\frac{x}{\lambda}) \right\|^2 > 2\varepsilon \mathbb{1}\left[ \text{prog}(x) < T \right]$ for all $x \in \mathbb{R}^T$, where we use Lemma D.1. Thus

$$T = \left\lfloor \frac{\Delta L}{2\varepsilon l_1 \Delta^0} \right\rfloor$$

and

$$p = \min \left\{ \frac{2\varepsilon \gamma_\infty^2}{\sigma^2}, 1 \right\}.$$

Protocol 2 generates a sequence $\{x^k\}_{k=0}^\infty$. We have

$$\inf_{k \in S_t} \left\| \nabla f(x^k) \right\|^2 > 2\varepsilon \inf_{k \in S_t} \mathbb{1}\left[ \text{prog}(x^k) < T \right]. \tag{41}$$

Further, we require the following auxillary lemma. See the proof in Section D.3.

**Lemma G.5.** *With probability not less than $1 - \delta$,*

$$\inf_{k \in S_t} \mathbb{1}\left[ \text{prog}(x^k) < T \right] \geq 1$$

*for*

$$t \leq \frac{1}{2} \min_{m \in [n]} \tau_m \left( 1 + \frac{1}{4pm} \right) \left( \frac{T}{2} + \log \delta \right).$$

Using Lemma G.5 with $\delta = 1/2$ and (41), we obtain

$$\mathbb{E}\left[ \inf_{k \in S_t} \left\| \nabla f(x^k) \right\|^2 \right] \geq 2\varepsilon \mathbb{P}\left( \inf_{k \in S_t} \mathbb{1}\left[ \text{prog}(x^k) < T \right] \right) > \varepsilon$$

for

$$t = \frac{1}{2} \min_{m \in [n]} \tau_m \left( 1 + \frac{\sigma^2}{8\gamma_\infty^2 m\varepsilon} \right) \left( \frac{\Delta L}{2\varepsilon l_1 \Delta^0} - 2 \right).$$

$\square$

### G.2.1 Proof of Lemma G.5

**Lemma G.5.** *With probability not less than $1 - \delta$,*

$$\inf_{k \in S_t} \mathbb{1} \left[ \text{prog}(x^k) < T \right] \geq 1$$

*for*

$$t \leq \frac{1}{2} \min_{m \in [n]} \tau_m \left( 1 + \frac{1}{4pm} \right) \left( \frac{T}{2} + \log \delta \right).$$

*Proof.* **(Part 1):** *Comment: in this part, we mirror the proof of Lemma D.2. We also show that if $\inf_{k \in S_t} \mathbb{1} \left[ \text{prog}(x^k) < T \right] < 1$ holds, then we have the inequality $\sum_{i=1}^{T} \widehat{t}_{\eta_i} \leq t$, where $\widehat{t}_{\eta_i}$ are random variables with some known "good" distributions. However, in this lemma the quantities $\widehat{t}_{\eta_i}$ are different.*

In Protocol 2, the algorithm $A$ consequently calls the oracle $O$. If $\inf_{k \in S_t} \mathbb{1} \left[ \text{prog}(x^k) < T \right] < 1$ holds, then exists $k \in S_t$ such that $\text{prog}(x^k) = T$. Since the algorithm $A$ is zero-respecting, the mappings $A^k$ will not output a non-zero vector (a vector with a non-zero first coordinate) unless the oracle returns a non-zero vector.

Let us define the smallest index $k(i)$ of the sequence when the progress $\text{prog}(x^{k(i)})$ equals $i$ :

$$k(i) := \inf \left\{ k \in \mathbb{N}_0 \,|\, i = \text{prog}(x^k) \right\} \in \mathbb{N}_0 \cup \{\infty\}.$$

**Definition G.6** (Sequence $(k_p^\xi, i_p^\xi)$). Let us consider a set

$$\{(k, i) \in \mathbb{N} \times [n] \,|\, s_q^{k-1} = 1 \text{ and } t^k \geq s_t^{k-1} + \tau_i\}, \quad s^{k-1} \equiv (s_t^{k-1}, s_q^{k-1}, s_x^{k-1}).$$

We order this set lexicographically and define the result sequence as $\{(k_p^\xi, i_p^\xi)\}_{p=1}^m$, where $m \in [0, \infty]$ is the size of the sequence. The sequence $s^{k-1}$ is defined in Protocol 2.

Note that the algorithm $A$ is only depends on the random samples $\left\{ \xi_{i_p^\xi}^{k_p^\xi} \right\}_{p=1}^m$ since the sequence $\{(k_p^\xi, i_p^\xi)\}_{p=1}^m$ are the indices of the random samples that are used in the oracle (37).

Let us denote the index of the first successful trial as $\eta_1$, i.e.,

$$\eta_1 := \inf \left\{ i \,\middle|\, \xi_{i_p^\xi}^{k_p^\xi} = 1 \text{ and } p \in [1, m] \right\} \in \mathbb{N} \cup \{\infty\}.$$

If $\inf_{k \in S_t} \mathbb{1} \left[ \text{prog}(x^k) < T \right] < 1$ holds, then $\eta_1 < \infty$. Using the times $t^k$, the algorithm $A$ consequently requests the gradient estimators from the oracle (37).

In each round of the oracle's calculation, the algorithm can get the vector $\widehat{\nabla} f(s_x; \xi_1)$ in not less than $\tau_1$ seconds, the vector $\sum_{i=1}^{2} \widehat{\nabla} f(s_x; \xi_i)$ in less than $\tau_2$ seconds, and so forth (see (37)). The algorithm can repeat any number of these rounds sequentially.

The algorithm $A$ one by one requests $m_1, \ldots, m_i, \cdots \in \{0, \ldots, n\}$ gradient estimators from the oracle (37). It takes at least $\tau_{m_i}$ seconds ($\tau_0 \equiv 0$) to get a vector $\sum_{i=1}^{m_i} \widehat{\nabla} f(s_x; \xi_i)$. Let us consider that $k \in \mathbb{N}$ is the *first* index of gradient estimators such that is depends on some $\xi_i = 1$. Necessarily, we have $\sum_{j=1}^{k} m_j \geq \eta_1$. Also, since the $A$ is zero-respecting, we have $\sum_{j=1}^{k} \tau_{m_j} \leq t^{k(1)}$. Note that $k$, and $m_1, \ldots, m_k$. depend on the algorithm's strategy. Let us find "the best possible" quantities $k$, and $m_1, \ldots, m_k$ that are independent of an algorithm.

Let us assume that $k^*$, and $m_1^*, \ldots m_k^*$ minimize the quantity

$$\min_{k, m_1, \ldots, m_k} \sum_{j=1}^{k} \tau_{m_j},$$

$$\text{s.t.} \quad k \in \mathbb{N},$$

$$m_1, \ldots m_k \in \{0, \ldots, n\},$$

$$\sum_{j=1}^{k} m_j \geq \eta_1. \tag{42}$$

Then, we have

$$t^{k(1)} \geq \sum_{j=1}^{k^*} \tau_{m_j^*}.$$

Note that if exists $j \in [k^*]$ such that $m_j^* > \eta_1$, then $k^*$, and $m_1^*, \ldots, m_{j-1}^*, \eta_1, m_{j+1}^* \ldots, m_k^*$ are also minimizers of (42) since the sequence $\tau_k$ is not decreasing. Therefore, (42) is equivalent to

$$\min_{k, m_1, \ldots, m_k} \sum_{j=1}^{k} \tau_{m_j},$$

$$\text{s.t.} \quad k \in \mathbb{N},$$

$$m_1, \ldots m_k \in \{0, \ldots, \eta_1\}, \tag{43}$$

$$\sum_{j=1}^{k} m_j \geq \eta_1.$$

Then, using the simple algebra, we have

$$t^{k(1)} \geq \sum_{j=1}^{k^*} \tau_{m_j^*} = \sum_{j \,:\, m_j^* \neq 0} \tau_{m_j^*} = \sum_{j \,:\, m_j^* \neq 0} m_j^* \frac{\tau_{m_j^*}}{m_j^*} \geq \sum_{j \,:\, m_j^* \neq 0} m_j^* \min_{m \in [\eta_1]} \frac{\tau_m}{m}$$

$$= \sum_{j=1}^{k^*} m_j^* \min_{m \in [\eta_1]} \frac{\tau_m}{m} \geq \eta_1 \min_{m \in [\eta_1]} \frac{\tau_m}{m}.$$

In the first inequality, we use that $m_j^* \in \{0, \ldots, \eta_1\}$ for all $j \in [k^*]$. Next, using Lemma G.9, we get

$$t^{k(1)} \geq \frac{1}{2} \min_{m \in [n]} \tau_m \left(1 + \frac{\eta_1}{m}\right).$$

Using the same reasoning, for $j \in \{0, \ldots, T-1\}$,

$$t^{k(j+1)} \geq t^{k(j)} + \frac{1}{2} \min_{m \in [n]} \tau_m \left(1 + \frac{\eta_{j+1}}{m}\right),$$

where $\eta_{j+1}$ is the index of the first successful trial of Bernouilli random variables when $\mathrm{prog}(\cdot) = j$. More formally, for all $j \in \{0, \ldots, T-1\}$:

**Definition G.7** (Sequence $(k_{j,p}^\xi, i_{j,p}^\xi)$)**.** Let us consider a set

$$\{(k, i) \in \mathbb{N} \times [n] \mid s_q^{k-1} = 1 \text{ and } t^k \geq s_t^{k-1} + \tau_i \text{ and } \mathrm{prog}(s_x^{k-1}) = j\}, \quad s^{k-1} \equiv (s_t^{k-1}, s_q^{k-1}, s_x^{k-1}).$$

We order this set lexicographically and define the result sequence as $\{(k_{j,p}^\xi, i_{j,p}^\xi)\}_{p=1}^{m_{j+1}}$, where $m_{j+1} \in [0, \infty]$ is the size of the sequence. The sequence $s^{k-1}$ is defined in Protocol 2.

Then,

$$\eta_{j+1} := \inf \left\{ i \,\middle|\, \xi_{i_{j,p}^\xi}^{k_{j,p}^\xi} = 1 \text{ and } p \in [1, m_{j+1}] \right\} \in \mathbb{N} \cup \{\infty\}.$$

By the definition of $k(j)$, $x^{k(j)}$ is the first iterate such that $\mathrm{prog}(\cdot) = j$. Therefore, the oracle can potentially start returning gradient estimators with the non-zero $j + 1^{\text{th}}$ coordinate from the $k(j)^{\text{th}}$ iteration.

Thus, if $\inf_{k \in S_t} \mathbb{1}\left[\mathrm{prog}(x^k) < T\right] < 1$ holds, then

$$\frac{1}{2} \sum_{i=1}^{T} \min_{m \in [n]} \tau_m \left(1 + \frac{\eta_i}{m}\right) \leq t^{k(T)} \leq t.$$

Finally, we can conclude that

$$\mathbb{P}\left(\inf_{k \in S_t} \mathbb{1}\left[\mathrm{prog}(x^k) < T\right] < 1\right) \leq \mathbb{P}\left(\frac{1}{2} \sum_{i=1}^{T} \min_{m \in [n]} \tau_m \left(1 + \frac{\eta_i}{m}\right) \leq t\right) \quad \forall t \geq 0. \tag{44}$$

As in Lemma D.6, we show that

**Lemma G.8.** *Let us take $l_{j+1} \in \mathbb{N}$. Then*

$$\mathbb{P}\left(\eta_{j+1} = l_{j+1}|\eta_j, \ldots, \eta_1\right) \leq (1-p)^{l_{j+1}-1}p \tag{45}$$

*for all $j \in \{0, \ldots, T-1\}$.*

See the proof in Section G.2.3. Intuitively, the algorithm $A$ can not increase the probability of getting a successful Bernouilli random variable earlier with its decisions.

Let us temporally define

$$\widehat{t}_{\eta_i} := \frac{1}{2} \min_{m \in [n]} \tau_m \left(1 + \frac{\eta_i}{m}\right)$$

for all $i \in [T]$.

**(Part 2):** *Comment: in this part, we use the standard technique to bound the large deviations of the sum $\sum_{i=1}^T \widehat{t}_{\eta_i}$.*

Note that a function $g : \mathbb{R} \to \mathbb{R}$ such that $g(x) := \frac{1}{2} \min_{m \in [n]} \tau_m \left(1 + \frac{x}{m}\right)$ is continuous, strongly-monotone and invertible. For $t' \geq 0$, we have

$$\mathbb{P}\left(\widehat{t}_{\eta_{j+1}} \leq t'|\eta_j, \ldots, \eta_1\right) = \mathbb{P}\left(g(\eta_{j+1}) \leq t'|\eta_j, \ldots, \eta_1\right) = \mathbb{P}\left(\eta_{j+1} \leq g^{-1}(t')|\eta_j, \ldots, \eta_1\right).$$

Using (45), we obtain

$$\mathbb{P}\left(\widehat{t}_{\eta_{j+1}} \leq t'|\eta_j, \ldots, \eta_1\right) \leq \sum_{j=1}^{\lfloor g^{-1}(t')\rfloor} (1-p)^{j-1}p \leq p\left\lfloor g^{-1}(t')\right\rfloor.$$

Let us define

$$p' := p\left\lfloor g^{-1}(t')\right\rfloor,$$

then

$$\mathbb{P}\left(\widehat{t}_{\eta_{j+1}} \leq t'|\eta_j, \ldots, \eta_1\right) \leq p'.$$

Using the Chernoff method, as in Lemma D.2, one can get

$$\mathbb{P}\left(\sum_{i=1}^T \widehat{t}_{\eta_i} \leq \widehat{t}\right) \leq e^{\widehat{t}/t' - T + 2p'T}.$$

Let us take

$$t' = g\left(\frac{1}{4p}\right) = \frac{1}{2} \min_{m \in [n]} \tau_m \left(1 + \frac{1}{4pm}\right),$$

then

$$p' = p\left\lfloor g^{-1}\left(g\left(\frac{1}{4p}\right)\right)\right\rfloor = p\left\lfloor \frac{1}{4p}\right\rfloor \leq \frac{1}{4}.$$

Therefore,

$$\mathbb{P}\left(\sum_{i=1}^T \widehat{t}_{\eta_i} \leq \widehat{t}\right) \leq e^{\widehat{t}/t' - \frac{T}{2}}.$$

Using (44), for

$$t \leq \frac{1}{2} \min_{m \in [n]} \tau_m \left(1 + \frac{1}{4pm}\right)\left(\frac{T}{2} + \log \delta\right),$$

we have

$$\mathbb{P}\left(\inf_{k \in S_t} \mathbb{1}\left[\text{prog}(x^k) < T\right] < 1\right) \leq \mathbb{P}\left(\sum_{i=1}^T \widehat{t}_{\eta_i} \leq t\right) \leq \delta.$$

The last inequality concludes the proof.

$$\square$$

### G.2.2   Lemma G.9

**Lemma G.9.** *Let us consider a sorted sequence $0 < \tau_1 \leq \cdots \leq \tau_n$ and a constant $\eta \in \mathbb{N}$. We define*

$$t_1 := \eta \min_{m \in [\eta]} \frac{\tau_m}{m},$$

*and*

$$t_2 := \min_{m \in [n]} \tau_m \left( 1 + \frac{\eta}{m} \right).$$

*Then*

$$t_1 \leq t_2 \leq 2t_1.$$

*Proof.* Additionally, let us define

$$m_1 := \arg\min_{m \in [\eta]} \frac{\tau_m}{m}$$

and

$$m_2 := \arg\min_{m \in [n]} \tau_m \left( 1 + \frac{\eta}{m} \right).$$

Then, using $m_1 \leq \eta$, we have

$$t_2 = \min_{m \in [n]} \tau_m \left( 1 + \frac{\eta}{m} \right) \leq \tau_{m_1} \left( 1 + \frac{\eta}{m_1} \right) \leq 2\tau_{m_1} \frac{\eta}{m_1} = 2t_1.$$

If $m_2 \leq \eta$, then

$$t_1 = \eta \min_{m \in [\eta]} \frac{\tau_m}{m} \leq \eta \frac{\tau_{m_2}}{m_2} \leq \tau_{m_2} \left( 1 + \frac{\eta}{m_2} \right) = t_2.$$

Otherwise, if $m_2 > \eta$,

$$t_1 = \eta \min_{m \in [\eta]} \frac{\tau_m}{m} \leq \eta \frac{\tau_\eta}{\eta} = \tau_\eta \leq \tau_{m_2} \leq \tau_{m_2} \left( 1 + \frac{\eta}{m_2} \right) = t_2.$$

$\square$

### G.2.3   Proof of Lemma G.8

In the following lemma, we use notations from Part 1 of the proof of Lemma G.5.

**Lemma G.8.** *Let us take $l_{j+1} \in \mathbb{N}$. Then*

$$\mathbb{P}\left( \eta_{j+1} = l_{j+1} | \eta_j, \ldots, \eta_1 \right) \leq (1-p)^{l_{j+1}-1} p \tag{45}$$

*for all $j \in \{0, \ldots, T-1\}$.*

The idea of the following proof repeats the proof of Lemma G.8. But Protocol 2 with the oracle (37) differ, so we present the proof for completeness.

*Proof.* We prove that

$$\mathbb{P}\left( \eta_{j+1} = l_{j+1} \middle| \bigcap_{i=1}^{j} \{\eta_i = l_i\} \right) \leq (1-p)^{l_{j+1}-1} p.$$

for all $l_1, \ldots, l_j \in \mathbb{N} \cup \{\infty\}$ such that $\mathbb{P}\left( \bigcap_{i=1}^{j} \{\eta_i = l_i\} \right) > 0$.

If exists $i \in [j]$ such that $l_i = \infty$, then

$$\mathbb{P}\left( \eta_{j+1} = l_{j+1} \middle| \bigcap_{i=1}^{j} \{\eta_i = l_i\} \right) = 0$$

for all $l_{j+1} \in \mathbb{N}$ (see details in the proof of Lemma D.6).

Assume that $l_i < \infty$ for all $i \in [j]$. By the definition of $\eta_{j+1}$, we have $m_{j+1} \geq l_{j+1}$ and $\xi^{k^\xi_{j,1}}_{i^\xi_{j,1}} = \cdots = \xi^{k^\xi_{j,l_{j+1}-1}}_{i^\xi_{j,l_{j+1}-1}} = 0$ and $\xi^{k^\xi_{j,l_{j+1}}}_{i^\xi_{j,l_{j+1}}} = 1$. Thus

$$\mathbb{P}\left(\eta_{j+1} = l_{j+1} \middle| \bigcap_{i=1}^{j}\{\eta_i = l_i\}\right) \leq \mathbb{P}\left(\bigcap_{p=1}^{l_{j+1}-1}\{\xi^{k^\xi_{j,p}}_{i^\xi_{j,p}} = 0\}, \xi^{k^\xi_{j,l_{j+1}}}_{i^\xi_{j,l_{j+1}}} = 1, m_{j+1} \geq l_{j+1} \middle| \bigcap_{i=1}^{j}\{\eta_i = l_i\}\right).$$

Let us define a set

$$S_{l_{j+1}} := \{((k_1, i_1), \ldots, (k_{l_{j+1}}, i_{l_{j+1}})) \in (\mathbb{N} \times [n])^{l_{j+1}} \mid \forall p < j \in [l_{j+1}] : (k_p, i_p) < (k_j, i_j)\} \quad \forall l_{j+1} \geq 1.$$

Using the law of total probability, we have

$$\mathbb{P}\left(\eta_{j+1} = l_{j+1} \middle| \bigcap_{i=1}^{j}\{\eta_i = l_i\}\right)$$

$$\leq \sum_{S_{l_{j+1}}} \mathbb{P}\left(\bigcap_{p=1}^{l_{j+1}-1}\{\xi^{k^\xi_{j,p}}_{i^\xi_{j,p}} = 0\}, \xi^{k^\xi_{j,l_{j+1}}}_{i^\xi_{j,l_{j+1}}} = 1, m_{j+1} \geq l_{j+1}, \bigcap_{p=1}^{l_{j+1}}\{(k^\xi_{j,p}, i^\xi_{j,p}) = (k_p, i_p)\} \middle| \bigcap_{i=1}^{j}\{\eta_i = l_i\}\right)$$

$$= \sum_{S_{l_{j+1}}} \mathbb{P}\left(\bigcap_{p=1}^{l_{j+1}-1}\{\xi^{k_p}_{i_p} = 0\}, \xi^{k_{l_{j+1}}}_{i_{l_{j+1}}} = 1, m_{j+1} \geq l_{j+1}, \bigcap_{p=1}^{l_{j+1}}\{(k^\xi_{j,p}, i^\xi_{j,p}) = (k_p, i_p)\} \middle| \bigcap_{i=1}^{j}\{\eta_i = l_i\}\right).$$

where we take the sum over all $((k_1, i_1), \ldots, (k_{l_{j+1}}, i_{l_{j+1}})) \in S_{l_{j+1}}$. If the event

$$\bigcap_{p=1}^{l_{j+1}}\{(k^\xi_{j,p}, i^\xi_{j,p}) = (k_p, i_p)\} \bigcap \{m_{j+1} \geq l_{j+1}\}$$

holds, then an event $\bigcap_{p=1}^{l_{j+1}} A_{(k_p, i_p)}$ holds, where

$$A_{(k_p, i_p)} := \{s^{k_p-1}_q = 1 \text{ and } t^{k_p} \geq s^{k_p-1}_t + \tau_{i_p} \text{ and } \text{prog}(s^{k_p-1}_x) = j\}.$$

At the same time, if $\bigcap_{p=1}^{l_{j+1}} A_{(k_p, i_p)}$ holds, then $\{m_{j+1} \geq l_{j+1}\}$ holds. Therefore,

$$\bigcap_{p=1}^{l_{j+1}}\{(k^\xi_{j,p}, i^\xi_{j,p}) = (k_p, i_p)\} \bigcap \{m_{j+1} \geq l_{j+1}\} = \bigcap_{p=1}^{l_{j+1}}\left(\{(k^\xi_{j,p}, i^\xi_{j,p}) = (k_p, i_p)\} \bigcap A_{(k_p, i_p)}\right)$$

and

$$\mathbb{P}\left(\eta_{j+1} = l_{j+1} \middle| \bigcap_{i=1}^{j}\{\eta_i = l_i\}\right)$$

$$\leq \sum_{S_{l_{j+1}}} \mathbb{P}\left(\bigcap_{p=1}^{l_{j+1}-1}\{\xi^{k_p}_{i_p} = 0\}, \xi^{k_{l_{j+1}}}_{i_{l_{j+1}}} = 1, \bigcap_{p=1}^{l_{j+1}}\left(\{(k^\xi_{j,p}, i^\xi_{j,p}) = (k_p, i_p)\} \bigcap A_{(k_p, i_p)}\right) \middle| \bigcap_{i=1}^{j}\{\eta_i = l_i\}\right).$$

Let us define $\sigma^i_j$ as a sigma-algebra generated by $(\xi^1_1, \ldots, \xi^1_n), (\xi^2_1, \ldots, \xi^2_n), \ldots, (\xi^i_1, \ldots, \xi^i_j)$ for all $i \geq 0$ and $j \in \{0, \ldots, n\}$. Then, we have

$$\bigcap_{p=1}^{l_{j+1}-1}\{\xi^{k_p}_{i_p} = 0\} \in \sigma^{k_{l_{j+1}-1}}_{i_{l_{j+1}-1}},$$

and

$$\bigcap_{p=1}^{l_{j+1}}\left(\{(k^\xi_{j,p}, i^\xi_{j,p}) = (k_p, i_p)\} \bigcap A_{(k_p, i_p)}\right) \in \sigma^{k_{l_{j+1}}-1}_n$$

since, for all $p \in [l_{j+1}]$, the event $A_{(k_p, i_p)}$ is only determined by $s^{k_p-1}$ and $t^{k_p}$ that *do not* depend on $\{\xi^j\}_{j=k_{l_{j+1}}}^{\infty}$. And, for all $p \in [l_{j+1}]$, the fact that $(k_{j,p}^{\xi}, i_{j,p}^{\xi}) = (k_p, i_p)$ *does not* depend on $\{\xi^j\}_{j=k_{l_{j+1}}}^{\infty}$. Note that $k_{i-1,l_i}^{\xi} < k_{j,l_{j+1}}^{\xi}$ (a.s.) for all $i \in [j]$. Thus

$$A_{(k_{l_{j+1}}, i_{l_{j+1}})} \bigcap \{(k_{j,l_{j+1}}^{\xi}, i_{j,l_{j+1}}^{\xi}) = (k_{l_{j+1}}, i_{l_{j+1}})\} \bigcap_{i=1}^{j} \{\eta_i = l_i\} \in \sigma_n^{k_{l_{j+1}}-1}$$

since, for all $i \in [j]$, this event implies that $k_{i-1,l_i}^{\xi} < k_{l_{j+1}}$. All in all, we have that

$$\bigcap_{p=1}^{l_{j+1}-1} \{\xi_{i_p}^{k_p} = 0\} \bigcap_{p=1}^{l_{j+1}} \left(\{(k_{j,p}^{\xi}, i_{j,p}^{\xi}) = (k_p, i_p)\} \bigcap A_{(k_p, i_p)}\right) \bigcap_{i=1}^{j} \{\eta_i = l_i\} \in \sigma_n^{k_{l_{j+1}}-1} \bigcup \sigma_{i_{l_{j+1}-1}}^{k_{l_{j+1}}-1}.$$

Since $\{\xi_{i_{l_{j+1}}}^{k_{l_{j+1}}} = 1\}$ is independent of $\sigma_n^{k_{l_{j+1}}-1} \bigcup \sigma_{i_{l_{j+1}-1}}^{k_{l_{j+1}}-1}$,[8] we get

$$\mathbb{P}\left(\eta_{j+1} = l_{j+1} \middle| \bigcap_{i=1}^{j} \{\eta_i = l_i\}\right)$$

$$\leq p \sum_{S_{l_{j+1}}} \mathbb{P}\left(\bigcap_{p=1}^{l_{j+1}-1} \{\xi_{i_p}^{k_p} = 0\}, \bigcap_{p=1}^{l_{j+1}} \left(\{(k_{j,p}^{\xi}, i_{j,p}^{\xi}) = (k_p, i_p)\} \bigcap A_{(k_p, i_p)}\right) \middle| \bigcap_{i=1}^{j} \{\eta_i = l_i\}\right).$$

If $l_{j+1} = 1$, observe that the events $\{(k_{j,l_{j+1}}^{\xi}, i_{j,l_{j+1}}^{\xi}) = (k_{l_{j+1}}, i_{l_{j+1}})\} \bigcap A_{(k_{l_{j+1}}, i_{l_{j+1}})}$ do not intersect for all $(k_{l_{j+1}}, i_{l_{j+1}}) \in \mathbb{N} \times [n]$. Thus, we can use the additivity of the probability, and obtain

$$\mathbb{P}\left(\eta_{j+1} = l_{j+1} \middle| \bigcap_{i=1}^{j} \{\eta_i = l_i\}\right)$$

$$\leq p\mathbb{P}\left(\bigcup_{(k,i) \in \mathbb{N} \times [n]} \left(\{(k_{j,l_{j+1}}^{\xi}, i_{j,l_{j+1}}^{\xi}) = (k, i)\} \bigcap A_{(k,i)}\right) \middle| \bigcap_{i=1}^{j} \{\eta_i = l_i\}\right) \leq p.$$

Otherwise, if $l_{j+1} > 1$, we also use the fact that the events do not intersect and get

$$\mathbb{P}\left(\eta_{j+1} = l_{j+1} \middle| \bigcap_{i=1}^{j} \{\eta_i = l_i\}\right)$$

$$\leq p \sum_{S_{l_{j+1}-1}} \mathbb{P}\left(\bigcap_{p=1}^{l_{j+1}-1} \{\xi_{i_p}^{k_p} = 0\}, \bigcap_{p=1}^{l_{j+1}-1} \left(\{(k_{j,p}^{\xi}, i_{j,p}^{\xi}) = (k_p, i_p)\} \bigcap A_{(k_p, i_p)}\right),\right.$$

$$\left. \bigcup_{(k,i) > (k_{l_{j+1}-1}, i_{l_{j+1}-1})} \left(\{(k_{j,l_{j+1}}^{\xi}, i_{j,l_{j+1}}^{\xi}) = (k, i)\} \bigcap A_{(k,i)}\right) \middle| \bigcap_{i=1}^{j} \{\eta_i = l_i\}\right)$$

$$\leq p \sum_{S_{l_{j+1}-1}} \mathbb{P}\left(\bigcap_{p=1}^{l_{j+1}-1} \{\xi_{i_p}^{k_p} = 0\}, \bigcap_{p=1}^{l_{j+1}-1} \left(\{(k_{j,p}^{\xi}, i_{j,p}^{\xi}) = (k_p, i_p)\} \bigcap A_{(k_p, i_p)}\right) \middle| \bigcap_{i=1}^{j} \{\eta_i = l_i\}\right),$$

where we used an inequality $\mathbb{P}(A, B) \leq \mathbb{P}(A)$ for any events $A$ and $B$. We take the sum over all $((k_1, i_1), \ldots, (k_{l_{j+1}-1}, i_{l_{j+1}-1})) \in S_{l_{j+1}-1}$, where

$$S_{l_{j+1}-1} = \{((k_1, i_1), \ldots, (k_{l_{j+1}-1}, i_{l_{j+1}-1})) \in (\mathbb{N} \times [n])^{l_{j+1}-1} \mid \forall p < j \in [l_{j+1}-1] : (k_p, i_p) < (k_j, i_j)\}.$$

Using the same reasoning, one can continue and get that

$$\mathbb{P}\left(\eta_{j+1} = l_{j+1} \middle| \bigcap_{i=1}^{j} \{\eta_i = l_i\}\right) \leq p(1-p)^{l_{j+1}-1}.$$

$\square$

---

[8]For all $i, j \geq 0$ and $l, p \in \{0, \ldots, n\}$, the union of $\sigma_l^i$ and $\sigma_p^j$ is a sigma-algebra since either $\sigma_l^i \subseteq \sigma_p^j$, or $\sigma_p^j \subseteq \sigma_l^i$.

## G.3 Proof of Theorem G.3

**Theorem G.3.** *Assume that Assumptions 7.1, 7.2 and 7.3 hold. Let us take the step size*

$$\gamma = \min\left\{\frac{1}{L}, \frac{\varepsilon m}{2L\sigma^2}\right\}$$

*in Method 9, then after*

$$K \geq \frac{12\Delta L}{\varepsilon} + \frac{12\Delta L\sigma^2}{\varepsilon^2 m}$$

*iterations the method guarantees that $\frac{1}{K}\sum_{k=0}^{K-1} \mathbb{E}\left[\left\|\nabla f(x^k)\right\|^2\right] \leq \varepsilon$.*

*Proof.* Note that

$$x^{k+1} = x^k - \gamma\frac{1}{m}\sum_{i=1}^{m} \widehat{\nabla} f(x^k, \xi_i),$$

where the stochatsic gradients are i.i.d. Therefore, we can use the classical SGD result (see Theorem D.8). For a stepsize

$$\gamma = \min\left\{\frac{1}{L}, \frac{\varepsilon m}{2L\sigma^2}\right\},$$

we have

$$\frac{1}{K}\sum_{k=0}^{K-1} \mathbb{E}\left[\left\|\nabla f(x^k)\right\|^2\right] \leq \varepsilon,$$

if

$$K \geq \frac{12\Delta L}{\varepsilon} + \frac{12\Delta L\sigma^2}{\varepsilon^2 m}.$$

$\square$

## G.4 Proof of Theorem G.4

**Theorem G.4.** *Let us consider Theorem G.3. We assume that $i^{th}$ worker returns a stochastic gradient every $\tau_i$ seconds for all $i \in [n]$. Without loss of generality, we assume that $0 < \tau_1 \leq \cdots \leq \tau_n$. Let us take*

$$m = \underset{m' \in [n]}{\arg\min} \tau_{m'}\left(1 + \frac{\sigma^2}{m'\varepsilon}\right). \tag{38}$$

*Then after*

$$12 \min_{m \in [n]} \tau_m\left(\frac{L\Delta}{\varepsilon} + \frac{\sigma^2 L\Delta}{m\varepsilon^2}\right) \tag{39}$$

*seconds Method 9 guarantees to find an $\varepsilon$-stationary point.*

*Proof.* In this setup, the method converges after $K \times \tau_m$ seconds because the delay of each iteration is determined by the slowest worker. Thus, the time complexity equals

$$\tau_m\left(\frac{12\Delta L}{\varepsilon} + \frac{12\Delta L\sigma^2}{\varepsilon^2 m}\right) = \frac{12\Delta L}{\varepsilon}\tau_m\left(1 + \frac{\sigma^2}{\varepsilon m}\right) = \frac{12\Delta L}{\varepsilon}\min_{m' \in [n]}\tau_m\left(1 + \frac{\sigma^2}{m'\varepsilon}\right),$$

where we use the choice of the number of workers $m$. $\square$

# H Proofs for Convex Case

## H.1 The "worst case" function in convex case

In this proof, we use the construction from (Woodworth et al., 2018). Let us define $B_2(0, R) := \left\{ x \in \mathbb{R}^{T+1} : \|x\| \le R \right\}$.

Let us take functions $f_{l,\eta} : \mathbb{R}^{T+1} \to \mathbb{R}$ and $\widetilde{f}_{l,\eta} : \mathbb{R}^{T+1} \to \mathbb{R}$ such that

$$f_{l,\eta}(x) := \min_{y \in \mathbb{R}^{T+1}} \left\{ \widetilde{f}_{l,\eta}(y) + \frac{\eta}{2} \|y - x\|^2 \right\}$$

and

$$\widetilde{f}_{l,\eta}(x) := \max_{1 \le r \le T+1} \left( lx_r - \frac{5l^2(r-1)}{\eta} \right),$$

where $l, \eta > 0$ are free parameters. Let us define

$$y(x) := \arg\min_{y \in \mathbb{R}^{T+1}} \left\{ \widetilde{f}_{l,\eta}(y) + \frac{\eta}{2} \|y - x\|^2 \right\}.$$

For ths function $f_{l,\eta}$, we have the following properties:

**Lemma H.1** (Woodworth et al. (2018))**.** *The function $f_{l,\eta}$ satisfies:*

1. *(Lemma 4) The function $f_{l,\eta}$ is convex, $l$–Lipschitz, and $\eta$–smooth.*

2. *(eq. 75)*

$$\min_{x \in B_2(0,1)} f_{l,\eta}(x) \le -\frac{l}{\sqrt{T+1}}.$$

3. *(Lemma 6) For all $x \in B_2(0,1)$, $\mathrm{prog}(\nabla f_{l,\eta}(x)) \le \mathrm{prog}(x) + 1$ and $\mathrm{prog}(y(x)) \le \mathrm{prog}(x) + 1$.*

## H.2 Proof of Theorem B.4

**Theorem B.4.** *Let us consider the oracle class $\mathcal{O}_{\tau_1,\dots,\tau_n}^{\mathrm{conv},\sigma^2}$ for some $\sigma^2 > 0$ and $0 < \tau_1 \le \cdots \le \tau_n$. We fix any $R, L, M, \varepsilon > 0$ such that $\sqrt{L}R > c_1\sqrt{\varepsilon} > 0$ and $M^2R^2 > c_2\varepsilon^2$. In the view Protocol 3, for any algorithm $A \in \mathcal{A}_{\mathrm{zr}}^R$, there exists a function $f \in \mathcal{F}_{R,M,L}^{\mathrm{conv}}$ and oracles and distributions $((O_1,\dots,O_n),(\mathcal{D}_1,\dots,\mathcal{D}_n)) \in \mathcal{O}_{\tau_1,\dots,\tau_n}^{\mathrm{conv},\sigma^2}(f)$ such that*

$$\mathbb{E}\left[ \inf_{k \in S_t} f(x^k) \right] - \inf_{x \in B_2(0,R)} f(x) > \varepsilon,$$

*where $S_t := \left\{ k \in \mathbb{N}_0 \big| t^k \le t \right\}$, and*

$$t = c \times \min_{m \in [n]} \left[ \left( \frac{1}{m} \sum_{i=1}^m \frac{1}{\tau_i} \right)^{-1} \left( \min \left\{ \frac{\sqrt{L}R}{\sqrt{\varepsilon}}, \frac{M^2R^2}{\varepsilon^2} \right\} + \frac{\sigma^2 R^2}{m\varepsilon^2} \right) \right].$$

*The quantities $c_1$, $c_2$ and $c$ are universal constants.*

In Step 1, we almost repeat the proof from Woodworth et al. (2018). Steps 2 and 3 are very close to Steps 2 and 3 of the proofs for the nonconvex case.

*Proof.* (**Step 1**: $f \in \mathcal{F}_{R,M,L}^{\mathrm{conv}}$) Following Woodworth et al. (2018), we assume that $R = 1$. Otherwise, one can rescale the parameters of the construction. Let us take the function $f_{l,\eta}$ from Section H.1 with parameters

$$l = \min \left\{ M, \frac{L}{10(T+1)^{3/2}} \right\} \text{ and } \eta = 10(T+1)^{3/2}l.$$

Using Lemma H.1, one can see that $f_{l,\eta}$ is convex, $M$–Lipschitz, and $L$–smooth. Therefore, $f_{l,\eta} \in \mathcal{F}_{R,M,L}^{\text{conv}}$. Further, we use the notation $f := f_{l,\eta}$.

(**Step 2**: Oracle Class)
Let us take

$$[\widehat{\nabla} f(x; \xi)]_j := \nabla_j f(x) \left( 1 + \mathbb{1}\left[ j > \text{prog}(x) \right] \left( \frac{\xi}{p} - 1 \right) \right) \quad \forall x \in \mathbb{R}^T,$$

and $\mathcal{D}_i = \text{Bernouilli}(p)$ for all $i \in [n]$, where $p \in (0, 1]$. We denote $[x]_j$ as the $j^{\text{th}}$ index of a vector $x \in \mathbb{R}^{T+1}$. It is left to show this mapping is unbiased and $\sigma^2$-variance-bounded. Indeed,

$$\mathbb{E}\left[ [\widehat{\nabla} f(x, \xi)]_i \right] = \nabla_i f(x) \left( 1 + \mathbb{1}\left[ i > \text{prog}(x) \right] \left( \frac{\mathbb{E}[\xi]}{p} - 1 \right) \right) = \nabla_i f(x)$$

for all $i \in [T + 1]$, and

$$\mathbb{E}\left[ \left\| \widehat{\nabla} f(x; \xi) - \nabla f(x) \right\|^2 \right] \leq \max_{j \in [T+1]} |\nabla_j f(x)|^2 \, \mathbb{E}\left[ \left( \frac{\xi}{p} - 1 \right)^2 \right]$$

because the difference is non-zero only in one coordinate. Thus

$$\mathbb{E}\left[ \left\| \widehat{\nabla} f(x, \xi) - \nabla f(x) \right\|^2 \right] \leq \frac{\|\nabla f(x)\|_\infty^2 (1 - p)}{p} \leq \frac{\|\nabla f(x)\|^2 (1 - p)}{p}$$

$$\leq \frac{l^2(1 - p)}{p} \leq \sigma^2 \quad \forall x \in B_2(0, 1).$$

where we take

$$p = \min\left\{ \frac{l^2}{\sigma^2}, 1 \right\}.$$

(**Step 3**: Analysis of Protocol)
Protocol 3 generates the sequence $\{x^k\}_{k=0}^\infty$. Then, we have

$$f(x^k) = \max_{1 \leq r \leq T+1} \left( l[y(x^k)]_r - \frac{5l^2(r - 1)}{\eta} \right) + \frac{\eta}{2} \left\| y(x^k) - x^k \right\|^2 \geq l[y(x^k)]_{T+1} - \frac{5l^2 T}{\eta}.$$

Assume that $\text{prog}(x^k) < T$. Using Lemma H.1, if $\text{prog}(x^k) < T$, then $\text{prog}(y(x^k)) \leq T$ and $[y(x^k)]_{T+1} = 0$. Therefore, $f(x^k) \geq -\frac{5l^2 T}{\eta} \geq -\frac{5l^2(T+1)}{\eta}$ and

$$f(x^k) - \min_{x \in B_2(0,1)} f(x) \geq \frac{l}{\sqrt{T+1}} - \frac{5l^2(T+1)}{\eta} = \frac{l}{2\sqrt{T+1}} = \min\left\{ \frac{M}{2\sqrt{T+1}}, \frac{L}{20(T+1)^2} \right\}$$

if $\text{prog}(x^k) < T$. Let us take

$$T = \min\left\{ \left\lfloor \frac{M^2}{64\varepsilon^2} - 1 \right\rfloor, \left\lfloor \frac{\sqrt{L}}{\sqrt{80}\sqrt{\varepsilon}} - 1 \right\rfloor \right\}$$

to ensure that

$$f(x^k) - \min_{x \in B_2(0,1)} f(x) \geq \min\left\{ \frac{M}{2\sqrt{T+1}}, \frac{L}{20(T+1)^2} \right\} \geq 4\varepsilon > 2\varepsilon$$

if $\text{prog}(x^k) < T$. It is left to use Lemma D.2 with $\delta = 1/2$:

$$\mathbb{E}\left[ \inf_{k \in S_t} f(x^k) \right] - \min_{x \in B_2(0,1)} f(x) > 2\varepsilon \mathbb{P}\left( \inf_{k \in S_t} \mathbb{1}\left[ \text{prog}(x^k) < T \right] \geq 1 \right) > \varepsilon \qquad (46)$$

for

$$t \leq \frac{1}{24} \min_{m \in [n]} \left[ \left( \sum_{i=1}^m \frac{1}{\tau_i} \right)^{-1} \left( \frac{\sigma^2}{l^2} + m \right) \right] \left( \frac{T}{2} - 1 \right)$$

$$= \frac{1}{24} \min_{m \in [n]} \left[ \left( \frac{1}{m} \sum_{i=1}^{m} \frac{1}{\tau_i} \right)^{-1} \left( \frac{\sigma^2}{ml^2} + 1 \right) \right] \left( \frac{T}{2} - 1 \right).$$

We can take universal constants $c_1$ and $c_2$ equal to $8 \cdot 64$ to ensure that $T \geq 6$. Therefore, $16\varepsilon \geq \frac{l}{2\sqrt{T+1}}$ and (46) holds for

$$t \leq \frac{1}{96} \min_{m \in [n]} \left[ \left( \frac{1}{m} \sum_{i=1}^{m} \frac{1}{\tau_i} \right)^{-1} \left( \frac{\sigma^2}{ml^2}(T+1) + (T+2) \right) \right],$$

for

$$t \leq \frac{1}{96} \min_{m \in [n]} \left[ \left( \frac{1}{m} \sum_{i=1}^{m} \frac{1}{\tau_i} \right)^{-1} \left( \frac{\sigma^2}{ml^2} \frac{l^2}{32^2 \varepsilon^2} + (T+2) \right) \right],$$

and for

$$t \leq \frac{1}{96} \min_{m \in [n]} \left[ \left( \frac{1}{m} \sum_{i=1}^{m} \frac{1}{\tau_i} \right)^{-1} \left( \frac{\sigma^2}{32^2 m \varepsilon^2} + \min \left\{ \frac{M^2}{64\varepsilon^2}, \frac{\sqrt{L}}{\sqrt{80}\sqrt{\varepsilon}} \right\} \right) \right].$$

$\square$

## H.3 Proof of Theorem B.8

**Theorem B.8.** *Assume that Assumptions B.5, B.6 and B.7 hold. Let us take the batch size $S = \max \left\{ \lceil \sigma^2/M^2 \rceil, 1 \right\}$, and $\gamma = \frac{\varepsilon}{M^2 + \sigma^2/S} = \Theta(\varepsilon/M^2)$ in Method 4, then after $K \geq 2M^2R^2/\varepsilon^2$ iterations the method guarantees that $\mathbb{E}\left[ f(\widehat{x}^K) \right] - f(x^*) \leq \varepsilon$, where $\widehat{x}^K = \frac{1}{K} \sum_{k=0}^{K-1} x^k$ and $R = \left\| x^* - x^0 \right\|$.*

*Proof.* Using the same reasoning as in the proof of Theorem 7.4, one can see that Method 4 is just the stochastic gradient method with the batch size $S$. Method 4 can be rewritten as $x^{k+1} = x^k - \gamma \frac{1}{S} \sum_{i=1}^{S} \widehat{\nabla} f(x^k; \xi_i)$, where the $\xi_i$ are independent random samples. It means that we can use the classical SGD result (Theorem H.2). For a stepsize

$$\gamma = \frac{\varepsilon}{M^2 + \frac{\sigma^2}{S}},$$

we have

$$\mathbb{E}\left[ f(\widehat{x}^K) \right] - f(x^*) \leq \varepsilon$$

if

$$K \geq \frac{2M^2 \left\| x^* - x^0 \right\|^2}{\varepsilon^2} \geq \frac{(M^2 + \frac{\sigma^2}{S}) \left\| x^* - x^0 \right\|^2}{\varepsilon^2}.$$

$\square$

### H.3.1 The classical SGD theorem in convex optimization

We reprove the classical SGD result (see, for instance, (Lan, 2020)) for convex functions.

**Theorem H.2.** *Assume that Assumptions B.5 and B.6 hold. We consider the SGD method:*

$$x^{k+1} = x^k - \gamma g(x^k),$$

*where*

$$\gamma = \frac{\varepsilon}{M^2 + \sigma^2}$$

*For a fixed $x \in \mathbb{R}^d$, $g(x)$ is a random vector such that $\mathbb{E}\left[ g(x) \right] \in \partial f(x)$ ($\partial f(x)$ is the subdifferential of the function $f$ at the point $x$),*

$$\mathbb{E}\left[ \left\| g(x) - \mathbb{E}\left[ g(x) \right] \right\|^2 \right] \leq \sigma^2,$$

*and $g(x^k)$ are independent vectors for all $k \geq 0$. Then*

$$\mathbb{E}\left[f\left(\frac{1}{K}\sum_{k=0}^{K-1}x^k\right)\right] - f(x^*) \leq \varepsilon \tag{47}$$

*for*

$$K \geq \frac{(M^2 + \sigma^2)\left\|x^* - x^0\right\|^2}{\varepsilon^2}.$$

*Proof.* We denote $\mathcal{G}^k$ as a sigma-algebra generated by $g(x^0),\ldots,g(x^{k-1})$. Using the convexity, for all $x \in \mathbb{R}^d$, we have

$$f(x) \geq f(x^k) + \left\langle \mathbb{E}\left[g(x^k)\big|\mathcal{G}^k\right], x - x^k\right\rangle = f(x^k) + \mathbb{E}\left[\left\langle g(x^k), x - x^k\right\rangle\big|\mathcal{G}^k\right].$$

Note that

$$\begin{aligned}
\left\langle g(x^k), x - x^k\right\rangle &= \left\langle g(x^k), x^{k+1} - x^k\right\rangle + \left\langle g(x^k), x - x^{k+1}\right\rangle \\
&= -\gamma\left\|g(x^k)\right\|^2 + \frac{1}{\gamma}\left\langle x^k - x^{k+1}, x - x^{k+1}\right\rangle \\
&= -\gamma\left\|g(x^k)\right\|^2 + \frac{1}{2\gamma}\left\|x^k - x^{k+1}\right\|^2 + \frac{1}{2\gamma}\left\|x - x^{k+1}\right\|^2 - \frac{1}{2\gamma}\left\|x - x^k\right\|^2 \\
&= -\frac{\gamma}{2}\left\|g(x^k)\right\|^2 + \frac{1}{2\gamma}\left\|x - x^{k+1}\right\|^2 - \frac{1}{2\gamma}\left\|x - x^k\right\|^2
\end{aligned}$$

and

$$\mathbb{E}\left[\left\|g(x^k)\right\|^2\Big|\mathcal{G}^k\right] = \mathbb{E}\left[\left\|g(x^k) - \mathbb{E}\left[g(x^k)\big|\mathcal{G}^k\right]\right\|^2\Big|\mathcal{G}^k\right] + \left\|\mathbb{E}\left[g(x^k)\big|\mathcal{G}^k\right]\right\|^2 \leq \sigma^2 + M^2.$$

Therefore, we get

$$\begin{aligned}
f(x^k) &\leq f(x) + \mathbb{E}\left[\left\langle g(x^k), x^k - x\right\rangle\big|\mathcal{G}^k\right] \\
&= f(x) + \frac{\gamma}{2}\mathbb{E}\left[\left\|g(x^k)\right\|^2\Big|\mathcal{G}^k\right] + \frac{1}{2\gamma}\left\|x - x^k\right\|^2 - \frac{1}{2\gamma}\mathbb{E}\left[\left\|x - x^{k+1}\right\|^2\Big|\mathcal{G}^k\right] \\
&\leq f(x) + \frac{\gamma}{2}\left(M^2 + \sigma^2\right) + \frac{1}{2\gamma}\left\|x - x^k\right\|^2 - \frac{1}{2\gamma}\mathbb{E}\left[\left\|x - x^{k+1}\right\|^2\Big|\mathcal{G}^k\right].
\end{aligned}$$

By taking the full expectation and summing the last inequality for $t$ from 0 to $K - 1$, we obtain

$$\begin{aligned}
\mathbb{E}\left[\sum_{k=0}^{K-1}f(x^k)\right] &\leq Kf(x) + \frac{K\gamma}{2}\left(M^2 + \sigma^2\right) + \frac{1}{2\gamma}\left\|x - x^0\right\|^2 - \frac{1}{2\gamma}\mathbb{E}\left[\left\|x - x^K\right\|^2\right] \\
&\leq Kf(x) + \frac{K\gamma}{2}\left(M^2 + \sigma^2\right) + \frac{1}{2\gamma}\left\|x - x^0\right\|^2.
\end{aligned}$$

Let divide the last inequality by $K$, take $x = x^*$, and use the convexity:

$$\mathbb{E}\left[f\left(\frac{1}{K}\sum_{k=0}^{K-1}x^k\right)\right] - f(x^*) \leq \frac{\gamma}{2}\left(M^2 + \sigma^2\right) + \frac{1}{2\gamma K}\left\|x^* - x^0\right\|^2.$$

The choices of $\gamma$ and $K$ ensure that (47) holds. $\qquad\square$

## H.4  Proof of Theorem B.9

**Theorem B.9.** *Let us consider Theorem B.8. We assume that $i^{th}$ worker returns a stochastic gradient every $\tau_i$ seconds for all $i \in [n]$. Without loss of generality, we assume that $0 < \tau_1 \leq \cdots \leq \tau_n$. Then after*

$$8\min_{m\in[n]}\left[\left(\frac{1}{m}\sum_{i=1}^{m}\frac{1}{\tau_i}\right)^{-1}\left(\frac{M^2R^2}{\varepsilon^2} + \frac{\sigma^2R^2}{m\varepsilon^2}\right)\right] \tag{17}$$

*seconds Method 4 guarantees to find an $\varepsilon$-solution.*

*Proof.* The proof is the same as in Theorem 7.5. It is only required to estimate the time that is required to collect a batch of size $S$. Method 4 returns a solution after

$$K \times 2t'(j^*) = \frac{4M^2 \left\| x^* - x^0 \right\|^2}{\varepsilon^2} \min_{j \in [n]} \left[ \left( \sum_{i=1}^{j} \frac{1}{\tau_i} \right)^{-1} (S + j) \right]$$

seconds. □

## H.5 Proof of Theorem B.10

**Theorem B.10.** *Assume that Assumptions B.5, 7.1 and 7.3 hold. Let us take the batch size* $S = \max \left\{ \left\lceil (\sigma^2 R)/(\varepsilon^{3/2} \sqrt{L}) \right\rceil, 1 \right\}$, *and* $\gamma = \min \left\{ \frac{1}{4L}, \left[ \frac{3R^2 S}{4\sigma^2 (K+1)(K+2)^2} \right]^{1/2} \right\}$ *in Accelerated Method 4, then after* $K \geq \frac{8\sqrt{L}R}{\sqrt{\varepsilon}}$ *iterations the method guarantees that* $\mathbb{E} \left[ f(x^K) \right] - f(x^*) \leq \varepsilon$, *where* $R \geq \left\| x^* - x^0 \right\|$.

*Proof.* One can see that Accelerated Method 4 is just the accelerated stochastic gradient method with the batch size $S$. It means that we can use the classical SGD result (Proposition 4.4 in Lan (2020)). For a stepsize

$$\gamma = \min \left\{ \frac{1}{4L}, \left[ \frac{3R^2 S}{4\sigma^2 (K+1)(K+2)^2} \right]^{1/2} \right\},$$

we have

$$\mathbb{E} \left[ f(x^K) \right] - f(x^*) \leq \frac{4LR^2}{K^2} + \frac{4\sqrt{\sigma^2 R^2}}{\sqrt{SK}}.$$

Therefore,

$$\mathbb{E} \left[ f(x^K) \right] - f(x^*) \leq \varepsilon$$

if

$$K \geq \frac{8\sqrt{L}R}{\sqrt{\varepsilon}} \geq 8 \max \left\{ \frac{\sqrt{L}R}{\sqrt{\varepsilon}}, \frac{\sigma^2 R^2}{\varepsilon^2 S} \right\},$$

where we use the choice of $S$. □

## H.6 Proof of Theorem B.11

**Theorem B.11.** *Let us consider Theorem B.10. We assume that* $i^{th}$ *worker returns a stochastic gradient every* $\tau_i$ *seconds for all* $i \in [n]$. *Without loss of generality, we assume that* $0 < \tau_1 \leq \cdots \leq \tau_n$. *Then after*

$$32 \min_{m \in [n]} \left[ \left( \frac{1}{m} \sum_{i=1}^{m} \frac{1}{\tau_i} \right)^{-1} \left( \frac{\sqrt{L}R}{\sqrt{\varepsilon}} + \frac{\sigma^2 R^2}{m\varepsilon^2} \right) \right]$$

*seconds Accelerated Method 4 guarantees to find an* $\varepsilon$*-solution.*

*Proof.* The proof is the same as in Theorem 7.5. It is only required to estimate the time that is required to collect a batch of size $S$. Accelerated Method 4 returns a solution after

$$K \times 2t'(j^*) = \frac{16\sqrt{L}R}{\sqrt{\varepsilon}} \min_{j \in [n]} \left[ \left( \sum_{i=1}^{j} \frac{1}{\tau_i} \right)^{-1} (S + j) \right]$$

seconds. □

# I  Construction of Algorithm for Rennala SGD

In this section, we provide the formal construction of the algorithm from Definition 4.1 for Rennala SGD. We consider the fixed computation model, where $i^{\text{th}}$ worker requires $\tau_i$ seconds to calculate stochastic gradients. We now define the corresponding sequence $\{A^k\}_{k=0}^{\infty}$. Let us fix a starting point $x^0 \in \mathbb{R}^d$, a stepsize $\gamma \geq 0$, and a batch size $S \in \mathbb{N}$.

First, let us consider the sequence $\{\widehat{t}_k\}_{k=1}^{\infty}$ from Definition D.4 that represents the times when the workers would be ready to provide stochastic gradients. Additionally, let us define $\widehat{t}_0 := 0$ and a sequence of the workers' indices $\{\widehat{i}_k\}_{k=1}^{\infty}$ that are corresponding to the times $\{\widehat{t}_k\}_{k=1}^{\infty}$. We can define

$$
A^k(g^1, \dots, g^k) = \begin{cases} \left( \widehat{t}_{(\lfloor k/2 \rfloor)}, \widehat{i}_{(\lfloor k/2 \rfloor + 1)}, x^0 - \dfrac{\gamma}{S} \displaystyle\sum_{j=1}^{2S \times \lfloor k/(2S) \rfloor} g^j \right), & k \ (\mathrm{mod}\ 2) = 0, \\[4mm] \left( \widehat{t}_{(\lfloor k/2 \rfloor + 1)}, \widehat{i}_{(\lfloor k/2 \rfloor + 1)}, 0 \right), & k \ (\mathrm{mod}\ 2) = 1, \end{cases} \tag{48}
$$

for all $k \geq 1$, and $A^0 = (\widehat{t}_0, \widehat{i}_1, x^0)$. For the fixed computation model, one can use this algorithm in Protocol 3 with the oracle (7) to get an equivalent procedure to Method 4.

## J Experiments

In this section, we compare Rennala SGD with Asynchronous SGD and Minibatch SGD on quadratic optimization tasks with stochastic gradients. The experiments were implemented in Python 3.7.9. The distributed environment was emulated on machines with Intel(R) Xeon(R) Gold 6248 CPU @ 2.50GHz.

### J.1 Setup

We consider the homogeneous optimization problem (1) with the convex quadratic function

$$f(x) = \frac{1}{2}x^\top \mathbf{A}x - b^\top x \quad \forall x \in \mathbb{R}^d.$$

We take $d = 1000$,

$$\mathbf{A} = \frac{1}{4} \begin{pmatrix} 2 & -1 & & 0 \\ -1 & \ddots & \ddots & \\ & \ddots & \ddots & -1 \\ 0 & & -1 & 2 \end{pmatrix} \in \mathbb{R}^{d \times d} \quad \text{and} \quad b = \frac{1}{4} \begin{bmatrix} -1 \\ 0 \\ \vdots \\ 0 \end{bmatrix} \in \mathbb{R}^d.$$

Assume that all $n$ workers has access to the following unbiased stochastic gradients:

$$[\widehat{\nabla} f(x; \xi)]_j := \nabla_j f(x) \left( 1 + \mathbb{1}\left[j > \text{prog}(x)\right] \left( \frac{\xi}{p} - 1 \right) \right) \quad \forall x \in \mathbb{R}^d,$$

where $\xi \sim \mathcal{D}_i = \text{Bernouilli}(p)$ for all $i \in [n]$, where $p \in (0, 1]$. We denote $[x]_j$ as the $j^{\text{th}}$ index of a vector $x \in \mathbb{R}^d$. In our experiments, we take $p = 0.01$ and the starting point $x^0 = [\sqrt{d}, 0, \ldots, 0]^\top$. We emulate our setup by considering that the $i^{\text{th}}$ worker requires $\sqrt{i}$ seconds to calculate a stochastic gradient. In all methods, we fine-tune step sizes from a set $\{2^i \,|\, i \in [-20, 20]\}$. In Rennala SGD, we fine-tune the batch size $S \in \{1, 5, 10, 20, 40, 80, 100, 200, 500, 1000\}$.

### J.2 Results

In Figures 1, 2, and 3, we present experiments with different number of workers $n \in \{100, 1000, 10000\}$. When the number of workers $n = 100$, Rennala SGD with Asynchronous SGD converge to the minimum at almost the same rate. However, when we start increasing the number of workers $n$, one can see that Asynchronous SGD[9] starts converging slower. This is an expected behavior since the maximum theoretical step size in Asynchronous SGD decreases as the number of workers $n$ increases (Koloskova et al., 2022; Mishchenko et al., 2022).

## K Experiment with Small-Scale Machine Learning Task

We also consider the methods in a more practical scenario. We solve a logistic regression problem with the *MNIST* dataset (LeCun et al., 2010). We take $n = 1000$ workers that hold the *same* subset of *MNIST* of the size 3000. Each worker samples stochastic gradients of size 4. In Figures 4 and 5, we provide convergence rates and a histogram of the time delays from the experiment. As in (Mishchenko et al., 2018), we can observe that asynchronous methods converge faster than Minibatch SGD. Unlike Section J where Rennala SGD converges faster than Asynchronous SGD, these methods have almost the same performance in this particular experiment.

---

[9]We implemented Asynchronous SGD with delay-adaptive stepsizes from (Koloskova et al., 2022)

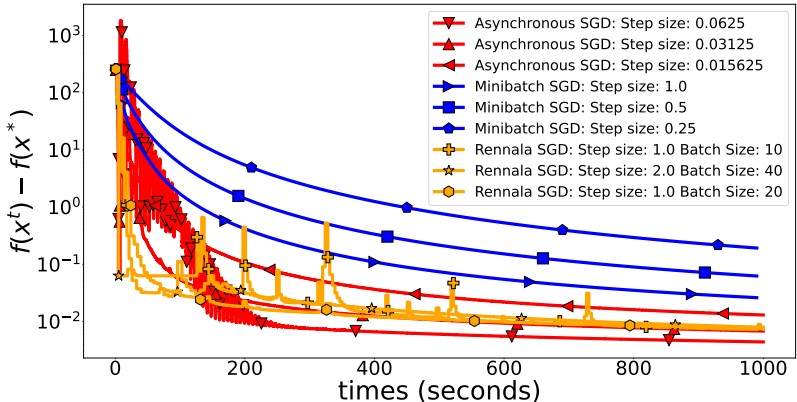

Figure 1: # of workers $n = 100$.

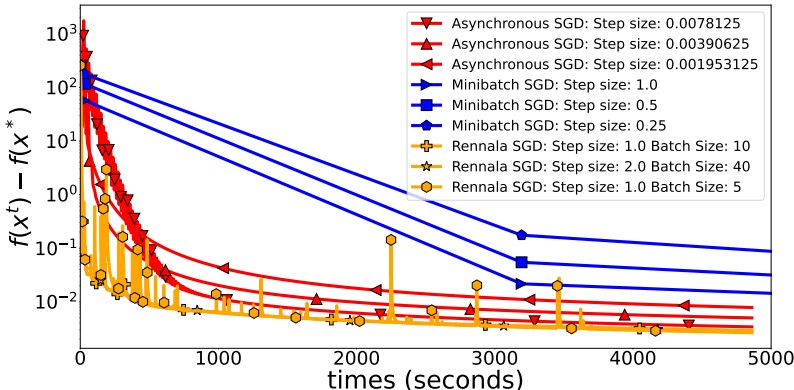

Figure 2: # of workers $n = 1000$.

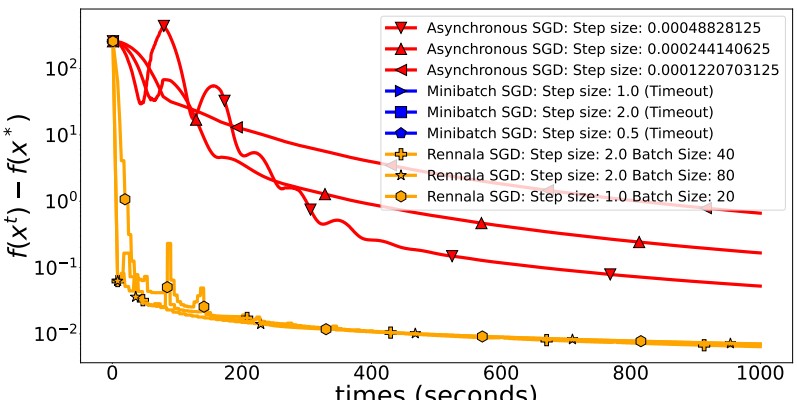

Figure 3: # of workers $n = 10000$.

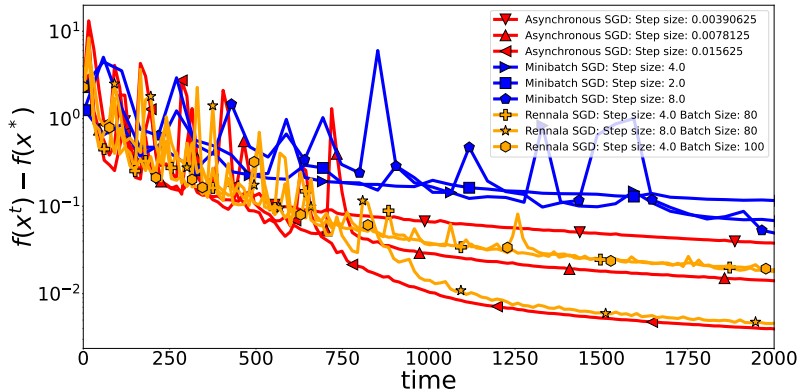

Figure 4: Logistic regression experiment

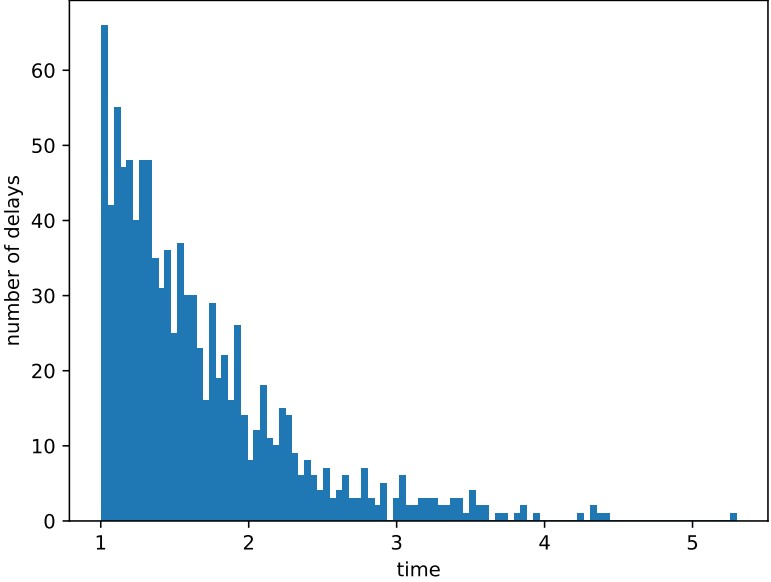

Figure 5: Histogram of time delays

## L    Time Complexity of Asynchronous SGD

The works (Mishchenko et al., 2022; Koloskova et al., 2022) state that Asynchronous SGD convereges after

$$O\left(\frac{nL\Delta}{\varepsilon} + \frac{\sigma^2 L\Delta}{\varepsilon^2}\right)$$

iterations. This result directly does not reveal the time complexity of Asynchronous SGD. Let us provide the time complexity for the case when the workers require exactly $\tau_i$ seconds to compute stochastic gradients. Let us fix a time $t \geq 0$. Then the workers will calculate at most

$$\sum_{i=1}^{n} \left\lfloor \frac{t}{\tau_i} \right\rfloor$$

stochastic gradients. To get $\varepsilon$-stationary point, we have to find the minimal $t$ such that

$$c \times \left(\frac{nL\Delta}{\varepsilon} + \frac{\sigma^2 L\Delta}{\varepsilon^2}\right) \leq \sum_{i=1}^{n} \left\lfloor \frac{t}{\tau_i} \right\rfloor, \tag{49}$$

where the quantity $c$ is a numerical constant, since the number of iterations can not be larger than the number of calculated gradients. Note that

$$\sum_{i=1}^{n} \left\lfloor \frac{t}{\tau_i} \right\rfloor \leq \sum_{i=1}^{n} \frac{t}{\tau_i}$$

The time $t'$ that satisfies

$$c \left( \frac{nL\Delta}{\varepsilon} + \frac{\sigma^2 L\Delta}{\varepsilon^2} \right) = \sum_{i=1}^{n} \frac{t'}{\tau_i}$$

is

$$t' = c \left( \frac{1}{n} \sum_{i=1}^{n} \frac{1}{\tau_i} \right)^{-1} \left( \frac{L\Delta}{\varepsilon} + \frac{\sigma^2 L\Delta}{n\varepsilon^2} \right). \tag{50}$$

Therefore, the minimal time $t$ that satisfies (49) is *greater or equal* to (50).

## M    Analysis of Fixed-Computation Model Using Graph Oracle Models

In this section, we analyze the fixed computation model, where $i^{\text{th}}$ worker requires $\tau_i$ seconds to calculate stochastic gradients. Without loss of generality, let us assume that $0 < \tau_1 \leq \cdots \leq \tau_n$. We use the graph oracle framework by Woodworth et al. (2018). Let us fix some $t > \tau_n$. Then, in the fixed computation model, the depth of the graph oracle $D$ is at least $\Omega\left( t/\tau_1 \right)$. This is the number of gradients that the first node can calculate after $t$ seconds. The size of the graph $N$ equals $\Theta\left( \sum_{i=1}^{n} \frac{t}{\tau_i} \right)$ since all workers calculate in parallel. Applying these estimates to Theorem 1 of (Woodworth et al., 2018), one can see that, for convex, $L$–smooth, $M$-Lipschitz problems with unbiased and $\sigma^2$-variance-bounded stochastic gradients on the ball $B_2(0, R)$, the lower bound is at most equals

$$O\left( \min\left\{ \frac{MR}{\sqrt{t/\tau_1}}, \frac{LR^2}{(t/\tau_1)^2} \right\} + \frac{\sigma R}{\sqrt{\sum_{i=1}^{n} \frac{t}{\tau_i}}} \right).$$

From this estimate, we can conclude that in order to get an $\varepsilon$–solution, it is required

$$t = O\left( \tau_1 \min\left\{ \frac{M^2 R^2}{\varepsilon^2}, \frac{\sqrt{L}R}{\sqrt{\varepsilon}} \right\} + \left( \frac{1}{n} \sum_{i=1}^{n} \frac{1}{\tau_i} \right)^{-1} \frac{\sigma^2 R^2}{n\varepsilon^2} \right)$$

seconds.

### M.1    Example when the lower bound from (Woodworth et al., 2018) is not tight

Let us provide an example when the lower bound from Theorem B.4 is strictly higher. Let us take $\tau_i = \sqrt{i}$ for all $i \in [n]$. Then, it is sufficient to compare

$$\min_{m \in [n]} \left[ \left( \frac{1}{m} \sum_{i=1}^{m} \frac{1}{\tau_i} \right)^{-1} \left( \min\left\{ \frac{\sqrt{L}R}{\sqrt{\varepsilon}}, \frac{M^2 R^2}{\varepsilon^2} \right\} + \frac{\sigma^2 R^2}{m\varepsilon^2} \right) \right]$$

and

$$\tau_1 \min\left\{ \frac{\sqrt{L}R}{\sqrt{\varepsilon}}, \frac{M^2 R^2}{\varepsilon^2} \right\} + \left( \frac{1}{n} \sum_{i=1}^{n} \frac{1}{\tau_i} \right)^{-1} \frac{\sigma^2 R^2}{n\varepsilon^2}.$$

Let us divide both formulas by the term with minimum and obtain

$$\min_{m \in [n]} \left[ \left( \frac{1}{m} \sum_{i=1}^{m} \frac{1}{\tau_i} \right)^{-1} \left( 1 + \frac{a}{m} \right) \right]$$

and

$$\tau_1 + \left(\frac{1}{n}\sum_{i=1}^{n}\frac{1}{\tau_i}\right)^{-1}\frac{a}{n}$$

for some $a \geq 0$. Since $\sum_{i=1}^{m}\frac{1}{\tau_i} = \sum_{i=1}^{m}\frac{1}{\sqrt{i}} = \Theta\left(\sqrt{m}\right)$, then we get

$$\min_{m\in[n]}\left[\left(\frac{1}{m}\sum_{i=1}^{m}\frac{1}{\tau_i}\right)^{-1}\left(1+\frac{a}{m}\right)\right] = \Theta\left(\min_{m\in[n]}\left[\sqrt{m}\left(1+\frac{a}{m}\right)\right]\right) = \Theta\left(\sqrt{a}\right)$$

and

$$\tau_1 + \left(\frac{1}{n}\sum_{i=1}^{n}\frac{1}{\tau_i}\right)^{-1}\frac{a}{n} = \Theta\left(1+\frac{a}{\sqrt{n}}\right),$$

if $a \leq n$. For instance, let us additionally assume that $a = \sqrt{n}$, then the first term equals $\Theta\left(n^{1/4}\right)$, while the second equals $\Theta(1)$. Theorefore, the lower bound from Theorem B.4 is tighter.

