# OpenReview forum: "Optimal Time Complexities of Parallel Stochastic Optimization Methods Under a Fixed Computation Model"
_NeurIPS.cc/2023/Conference — NeurIPS 2023 poster_

### Official Review · Reviewer_AK4H · 2023-07-06

**Soundness:** 4 excellent
**Presentation:** 4 excellent
**Contribution:** 3 good
**Rating:** 7
**Confidence:** 4

**Summary:**

Consider the classical statistical risk minimization framework with stochastic gradients, and propose new lower bounds on the optimal time complexity. Also propose a new method Rennala SGD that attains the lower bound. This work makes the case for asynchronous methods.

In particular, this work proposes a new oracle-based lower bound on the optimal convergence rate in time, rather than iterations, and develops an algorithm (essentially straggler-based SGD) that achieves this bound in the shared-system/parameter-server setting.

**Strengths:**

**Significance**
This work takes a step towards developing more practical asynchronous optimization methods, and providing convergence rates and complexity bounds that more closely reflect practical considerations of convergence time, rather than iterations.

**Originality**
The proposed method is not entirely novel (e.g., see Dutta et al., 2018 or related works on straggler-based SGD methods that ignore computation from stragglers), however the new lower bounds, problem formulation, and analysis are of sufficient novelty, to the best of my knowledge.

**Clarity**
The work is well written and easy to follow.

**Weaknesses:**

Lacking empirical evaluations, the only such experiments are on quadratics, buried deep inside the appendix. While these plots are compelling, I would encourage the authors to consider small-scale numerical results on other machine learning workloads.

Another weakness of this work is on assuming a fixed computation time for each worker. Such models are highly restrictive and impractical. I would encourage the authors to consider ways for extending their framework to arbitrary, but bounded worker delays. I would appreciate a discussion on how such lower-bounds (or the optimal method), may change if these bounds were overly conservative and long-tailed.

**Questions:**

Please see my questions provided in-line above, in the weaknesses section.

**Limitations:**

Limitations of the work are not adequately addressed. There are for instance, many assumptions that are taken for granted in the analysis that may limit the applicability of the findings to practical settings... namely the fixed delay model per worker, and the relationship between the batch-size S and the number of workers n. I am curious how findings would change in settings where S >> n or S << n.

---

> ### Author Rebuttal · Authors · 2023-08-06
>
> Thank you for the interesting comments!
>
> > Lacking empirical evaluations, the only such experiments are on quadratics, buried deep inside the appendix. While these plots are compelling, I would encourage the authors to consider small-scale numerical results on other machine learning workloads.
>
> Our work is theoretical. Thus we seek to understand the theoretical properties of asynchronous methods and their limits. One can see that even on simple quadratic optimization tasks, Asynchronous SGD can have bad convergence properties. We expect such behavior on other machine learning tasks. Having said that, we will follow your advice and include a few more small-scale experiments with different tasks. We expect the results to be similar.
>
> > Another weakness of this work is on assuming a fixed computation time for each worker. Such models are highly restrictive and impractical. I would encourage the authors to consider ways for extending their framework to arbitrary, but bounded worker delays. I would appreciate a discussion on how such lower-bounds (or the optimal method), may change if these bounds were overly conservative and long-tailed.
>
> Please note that unlike much of existing asynchronous optimization/SGD literature, which works with the formalism of **(iteration) delays** (which does not refer to time but to the difference in iteration counters between when a model was read and an update by a "delayed" worker written/applied), we directly work with the notion of **time** it takes for each worker to perform its computation. This is a different paradigm. The classical iteration delays are not first-class citizens in our formalism, i.e., they are not assumed. Instead, they are the result of the algorithm and the problem (e.g., larger noise $\sigma^2$ and/or smaller error tolerance $\varepsilon$ lead to larger batch size in Thm 7.4, and in classical asynchronous SGD, this would lead to larger iteration delays since such an approach counts every stochastic gradient as an iteration), and an indirect function of the speeds $\tau_i$, which are the first-class citizens in our work. We believe our approach is more reasonable - we assign processing times to all workers, and the  delays are a function of the algorithm. Not assumed, but observed.
>
> We say this to make it more clear that our constants $\tau_i$ should not be mistaken with the iteration delays from the majority of  previous works on asynchronous SGD (we are not saying the reviewer made such a mistake - we just want to make this very clear since the reviewer refers to delays in his/her question). These prior works sometimes indeed work with non-constant and even unbounded delays. Due to the different nature of iteration delays and compute times $\tau_i$, the fact that they do so, and that we work with fixed times $\tau_i$ should therefore not be seen as a relative shortcoming of our approach.
>
> We therefore interpret the question as follows: " I would encourage the authors to consider ways for extending their framework to arbitrary, but bounded worker computation times $\tau_i$."
>
> Having said that, it absolutely does make sense to extend our setup to non-constant processing times $\tau_i$, e.g.,
> - processing times belonging to some interval,
> - or random processing times following some distribution (with mean $\tau_i$ an variance $\sigma_i^2$,
> - or processing times which depend on the stochastic gradient sampled).
>
> We believe these are all good ideas for future study, which should be simpler given our foundational work in the constant $\tau_i$ setting. We believe that more complicated computation models deserve a separate research effort and analysis. We will add a future work section where we will comment on this question.
>
> We remark that Reviewer 563Q also asked us a similar question, too.
>
> > Limitations of the work are not adequately addressed. There are for instance, many assumptions that are taken for granted in the analysis that may limit the applicability of the findings to practical settings... namely the fixed delay model per worker, and the relationship between the batch-size S and the number of workers n. I am curious how findings would change in settings where S >> n or S << n.
>
> The assumption "the fixed delay model per worker" (here delays mean computation/processing times $\tau_i$, and not iteration delays) was never hidden. It is immediately considered in the introduction. Even in the title of the paper, we say that we get theoretical complexities "Under a Fixed Computation Model."
>
> All our findings will hold in both regimes $S \gg n$ or $S \ll n$. Let us clarify it. If $S \gg n$, then $\frac{\sigma^2}{\varepsilon} \gg n.$ It is a "high noise/small # of workers" regime. Then one can show that (11) is minimized when $m \approx n.$  In the "high noise/small # of workers" regime, we need the contribution of all workers. However, when $S \ll n,$ then $\frac{\sigma^2}{\varepsilon} \ll n.$ It is a "low noise/large # of workers" regime. In this case, the optimal $m$ can be much smaller than $n$ (see Lines 199-205). We will be happy to add this clarification to the camera-ready paper.

---

> > ### Comment · Reviewer_AK4H · 2023-08-21
> > **Re: Rebuttal**
> >
> > I acknowledge having read the author response, which addressed my minor comments. I have no major issues with this work, and will maintain my recommendation.

---

### Official Review · Reviewer_DBuG · 2023-07-09

**Soundness:** 2 fair
**Presentation:** 2 fair
**Contribution:** 2 fair
**Rating:** 4
**Confidence:** 2

**Summary:**

This paper proposes a protocol that generalizes the classical oracle framework approach. Using this protocol, it establishes minimax complexities for parallel optimization methods that have access to an unbiased stochastic gradient oracle with bounded variance.

**Strengths:**

- The motivation of this paper is clear.
- The theoretical analysis seems sound.

**Weaknesses:**

I am not an expert on this topic of this paper.

1. The challenge of theoretical analysis should be made more clear, which highly relates to the contribution of this paper.
2. The empirical study lacks.
3. The organization of this paper is strange. E.g., related work is located in 6.1, and conclusion section lacks, etc.

**Questions:**

Please cf. weaknesses part.

---

> ### Author Rebuttal · Authors · 2023-08-06
>
> Thank you for the review!
>
> > The challenge of theoretical analysis should be made more clear, which highly relates to the contribution of this paper.
>
> Let us briefly explain the main challenges of the paper:
> 1. Even before we started proving the lower bounds, the main challenge was to understand how to analyze parallel optimization methods. In Section 3, we explain that previous papers were counting the number of oracle calls. We proposed a completely different paradigm, where instead of counting *oracle calls*, we count *time*. We believe that this is the first challenge.
> 2. The next challenge comes from analyzing the lower bound for the complexity (6) (lines 113-144 in the paper). Nobody before us analyzed this complexity. The analysis required us to develop new non-trivial proof steps. For example, we discuss these non-trivialities in Section D (e.g., lines 571-573, 596-605, 670-674). There are many technical details that were not considered before.
> 3. The next challenge was to develop the new optimal methods, Rennala SGD and Malenia SGD. These are new methods that improve the complexity of the celebrated Asynchronous SGD! For a long time, the community *believed* that Asynchronous SGD is the best asynchronous method. Our work shows that Asynchronous SGD is suboptimal. Instead of it, we designed new optimal methods. We believe that these are non-trivial solved challenges.
>
> **Summing up, our goal was to understand the theoretical limits of the celebrated stochastic gradient descent (SGD) method in the asynchronous setting with $n$ parallel workers. We believe that we took a nontrivial step towards a better understanding of parallel optimization methods. Further, we believe our work is a fundamental contribution to an important subfield of machine learning, resolving a long-standing open problem. As such, we think that a borderline/weak score is not appropriate. We hope we can convince the reviewer about this.**
>
> > The empirical study lacks.
>
> Note that in Section J, we consider experiments. There we show that even on simple quadratic optimization tasks, Asynchronous SGD can have bad convergence rates. Our new method significantly outperforms the previous baseline. However, our work is theoretical and we would want it to be judged as such.
>
> > The organization of this paper is strange. E.g., related work is located in 6.1, and conclusion section lacks, etc.
>
> We understand that related work sections are typically written at the beginning of papers. In our paper, we put related work later because it makes the explanation of the Time Oracle Protocol easier. We can add the future work or conclusion section. We followed an organization that best fits our paper and its contents rather than the other way around: fit the paper into a pre-conceived/static organization structure. We believe the paper benefitted from this.
>
> We hope that our comments have addressed the reviewer's concerns. If the reviewer has other questions, we will be happy to continue the discussion.

---

> ### Author Response · Authors · 2023-08-16
> **A question**
>
> Dear reviewer,
>
> Please can you let us know if the other reviews and our rebuttal changed your mind about our paper? Thanks in advance for taking the time to reply during Summer/vacation season.
>
> Cheers,
>
> authors

---

### Official Review · Reviewer_Vdkt · 2023-07-14

**Soundness:** 3 good
**Presentation:** 3 good
**Contribution:** 3 good
**Rating:** 7
**Confidence:** 4

**Summary:**

This paper studies time complexity of parallel stochastic optimization, establishes lower bounds, and proposes algorithms achieving matching complexity results.

**Strengths:**

Based on my reading of the main paper (I haven't read the proofs), this is a solid theoretical work. The theoretical results improve widely used asynchronous SGD, and are even a bit surprising, especially the fact that ignoring some computations actually helps.

**Weaknesses:**

General comments:
- The organization of the paper is a bit reader unfriendly. The assumptions are mentioned at multiple places, starting with page 1, but are formally introduced almost at the end (page 8). This makes the reading tedious.

Minor comments:
- line 16, shouldn't the codomain of $f$ be $\mathbb R$ rather than $\mathbb R^d$?
- line 39: $L, \Delta$ have not been introduced so far in the paper
- Async. SGD (after line 52): step 2 gives the impression that only homogeneous data distribution across devices is considered in the paper. This is true for the main paper, but since the authors do have heterogeneous case results as well, they might consider reflecting that here.
- Lines 98-99: "this approach is not convenient" Maybe giving a short intuition why (which becomes clear on the next page), would help the reader.
- Typo in line 212

**Questions:**

Doubts:
- In (6), S_t is the set of indices that started computing gradients before time t. Shouldn't we consider k's which "finished" computation before t?
- Line 205: what does it mean by the freedom to interrupt oracles?

Questions:
- As the authors themselves remark in line 237, Rennala SGD goes contradictory to the idea of using all stochastic gradients. In that case, have the authors studied how it performs in practice in comparison to async SGD, especially with heterogeneity?
- In Theorem 7.4, we need a large batch size in the presence of sizable noise. Since in a lot of analyses, the ratio of learning rate and batch size is the crucial quantity, is it possible to choose $S=1$, and $\gamma = \Theta (\epsilon)$, and still get convergence? One would expect this to be closer to asynch SGD while achieving better rates, and performing better in practice.
- I haven't looked at Appendix A, but looking at the heterogeneous result in Table 1, the lower bound and Malenia SGD complexity have no dependence on the heterogeneity term. How come?
- The bound in (12) for when the workers start simultaneously: What is the explanation for this approach being slower?

---

> ### Author Rebuttal · Authors · 2023-08-06
>
> Thank you for the comments!
>
> > The organization of the paper is a bit reader unfriendly. The assumptions are mentioned at multiple places, starting with page 1, but are formally introduced almost at the end (page 8). This makes the reading tedious.
>
> We define assumptions about our main problem (1) only in Section 7.1. In the introduction (Section 1), we give the reader a brief description of the class of optimization problems that we consider. If we delete Lines 21-22, then it would be more difficult to explain the history of previous results in Section 1.1.
>
> > line 16, ...
> > line 39, ...
>
> Thank you, indeed, we will fix it.
>
> > Async. SGD (after line 52): step 2 gives the impression that only homogeneous data distribution across devices is considered in the paper. This is true for the main paper, but since the authors do have heterogeneous case results as well, they might consider reflecting that here.
>
> We will add a comment near line 52. Note that we have a long discussion in Sections A.2 and A.4. We do not avoid the fact that Async. SGD was considered in the heterogeneous case.
>
> > Lines 98-99: "this approach is not convenient" Maybe giving a short intuition why (which becomes clear on the next page), would help the reader.
>
> We tried to explain it in Lines 108-116. Unfortunately, we could not find a nice way to explain why "this approach is not convenient" without introducing our new protocol first.
>
> > In (6), S_t is the set of indices that started computing gradients before time t. Shouldn't we consider k's which "finished" computation before t?
>
> We consider all k's which "finished" computation before t. Indeed, let us fix an index $k$ and a time $t.$ Let us consider the gradient $g^k.$ From line 5 in Protocol 2, we know that the time of calculation of $g^k$ is $t^k.$ So if $t^k \leq t,$ then by definition of $S_t,$ we have that $k \in S_t.$
>
> > Line 205: what does it mean by the freedom to interrupt oracles?
>
> In Section F, we generalize our new protocol and assume that an algorithm can stop the calculations of workers at any time. In Theorem 6.4, we analyze the oracle from (7). Note that an algorithm should wait till the end of every calculation and can not stop/interrupt calculations in (7). In Section F, we answered to the question: "How will the complexity change if an algorithm can stop calculations at any time?." We show that it does not change the complexity.
>
> > As the authors themselves remark in line 237, Rennala SGD goes contradictory to the idea of using all stochastic gradients. In that case, have the authors studied how it performs in practice in comparison to async SGD, especially with heterogeneity?
>
> In Section J, we have experiments that support our theory. We show that Rennala SGD (that ignores previous stoch. gradients) has better time complexities than Async. SGD.
>
> > In Theorem 7.4, we need a large batch size in the presence of sizable noise. ...  is it possible to choose $S = 1$, and $\gamma = O(\epsilon)$, and still get convergence?
>
> Yes, it is possible. With this choice of parameters, Rennala SGD reduces to the classical SGD method with the fastest worker. However, it is better to choose $S = n,$ then one can show that Rennala SGD will have the suboptimal time complexity of Async. SGD. (we didn't add it to the paper because the choice of the parameter is suboptimal). But this a good question. We should probably add this clarification to the paper.
>
> > I haven't looked at Appendix A, but looking at the heterogeneous result in Table 1, the lower bound and Malenia SGD complexity have no dependence on the heterogeneity term. How come?
>
> Please take a look at the proof of Theorem A.3 for the heterogeneous case. It is very short and gives the answer to why Malenia SGD complexity has no dependence on the heterogeneity term. The main idea is that Melania SGD calculates unbiased stochastic gradients of the function $f$ (not of the function $f_i.$). So Melania SGD can work in the arbitrary heterogeneous setting!
>
> > The bound in (12) for when the workers start simultaneously: What is the explanation for this approach being slower?
>
> One should compare (12) and (11). First, the harmonic mean $\left(\frac{1}{m}\sum_{i=1}^m \frac{1}{\tau_i}\right) \leq \tau_m,$ so (11) $\leq $ (12). One can show that (11) $\ll$ (12) by taking $\tau_i = i$ (as a theoretical example to show the gap). Since $\frac{1}{m}\sum_{i=1}^m \frac{1}{i} \approx \frac{\ln m}{m},$ then
> $$(11) \approx \min_{m \in [n]} \left(\frac{m}{\ln m} \left(\frac{L \Delta}{\varepsilon} + \frac{\sigma^2 L \Delta}{\varepsilon^2 m} \right)\right) \leq \frac{\sigma^2 L \Delta}{\varepsilon^2} \frac{1}{\ln \frac{\sigma^2}{\varepsilon}},$$ where we take $m \approx \frac{\sigma^2}{\varepsilon}$
> and
> $$(12) = \min_{m \in [n]} \left(m \left(\frac{L \Delta}{\varepsilon} + \frac{\sigma^2 L \Delta}{\varepsilon^2 m} \right)\right) \geq \frac{\sigma^2 L \Delta}{\varepsilon^2}.$$
> Since $\frac{\sigma^2 L \Delta}{\varepsilon^2} \frac{1}{\ln \frac{\sigma^2}{\varepsilon}} \leq \frac{\sigma^2 L \Delta}{\varepsilon^2},$ we have (11) $\ll$ (12).
>
> Thank you for the comments!

---

> > ### Comment · Reviewer_Vdkt · 2023-08-18
> >
> > Thanks for the response. I have no further questions. I maintain my score (7) and support the paper's acceptance.

---

### Official Review · Reviewer_563Q · 2023-07-22

**Soundness:** 3 good
**Presentation:** 3 good
**Contribution:** 4 excellent
**Rating:** 7
**Confidence:** 4

**Summary:**

This paper studies the minimax complexities of distributed asynchronous stochastic optimization methods. By extending the oracle framework previously used in the literature, it establishes new (lower) lower bounds for parallel optimization methods. Based on the insights from their proof, they propose a new algorithm meeting these bounds. Notably, it is provably faster than previous synchronous and asynchronous methods in the homogeneous case, which is verified experimentally.

**Strengths:**

* This paper naturally extends the oracle protocol of previous work to cater for analysing parallel methods instead of sequential ones, which allows for a finer analysis of distributed methods.
* Efforts have been made in the writing to explain the introduced novelties step by step, which greatly helps understand and position the contributions.
* The insight taken from their new complexity proof that “ignoring stale gradients might actually help to converge faster” is interesting and highly relevant to the community, as it led to the development of a new state-of-art optimal method, which has provably and experimentally faster convergence rates than previous asynchronous algorithms in the homogeneous case.


**Weaknesses:**

* Many notations are introduced, which makes the paper a bit cluttered and cumbersome to read at times.
* No experiments comparing Renata & Asynchronous SGD to minibatch (synchronized) SGD are made to confirm the “provably better rates” claimed Section 8.


**Questions:**

* Your model assumes fixed delays per worker: could your analysis also hold with stochastic ones?
* You experimented with a fixed delay of $\sqrt i$ for worker $i$ (line 1163). Did you also experiment with real-life delays experienced naturally in the distributed asynchronous setting (as in Mischenko 2022 [1])? (For example, this would hint that your method also work with stochastic delays)
* Did you experiment with other convex functions other than the one introduced line 1159, and did you consistently observe that Renata leads to better results than Asynchronous SGD, or is this highly dependent on the convex functions used?
* Why Goyal et al 2017 [2] is cited line 45?

**Typos :**

* Line 212: our framework *is*.

**References :**

[1] Mishchenko, K., Bach, F., Even, M., andWoodworth, B, *Asynchronous SGD beats minibatch SGD under arbitrary delays*, In NeurIPS, 2022.

[2] Goyal, P., Dolla´r, P., Girshick, R., Noordhuis, P., Wesolowski, L., Kyrola, A., Tulloch, A., Jia, Y., and He, K., *Accurate, large minibatch SGD: Training imagenet in 1 hour.*

**Limitations:**

*

---

> ### Author Rebuttal · Authors · 2023-08-06
>
> Thank you for your positive review!
>
> > Your model assumes fixed delays per worker: could your analysis also hold with stochastic ones?
>
> > You experimented with a fixed delay of ... Did you also experiment with real-life delays experienced naturally in the distributed asynchronous setting...
>
> Great question. We were thinking about this setup when we were writing the paper. However, we decided to stop with the fixed computation model and understand it first. We didn't try to run ahead of the train. Note that the Rennala and Malenia SGD methods can work even with stochastic delays. See Theorem 7.4 and Theorem A.3. These theorems do not assume that the delays are constants.
> We believe that stochastic delays should be analyzed in future work. We will add a future work section where we will comment on this question. We believe that experiments with stochastic delays will be more appropriate in a paper that will consider such delays.
>
> > Did you experiment with other convex functions other than the one introduced line 1159...
>
> In this paper, we only experimented with quadratic optimization tasks. In Figure 3, we show that Asynchronous SGD is not robust to slow workers. We expect such behavior in other practical optimization problems also. Quadratic optimization tasks are the simplest and the best-understood problems in optimization, and even on these problems, Asynchronous SGD can show bad performance.
>
> > No experiments comparing Renata & Asynchronous SGD to minibatch (synchronized) SGD are made to confirm the “provably better rates” claimed Section 8.
>
> Note that \[1, Fig. 1\] showed that Asynchronous SGD is better than Minibatch SGD in their experiments. Unlike \[1\], we provide *theoretical* evidence that asynchronous methods are strictly better.
>
> \[1\]: Mishchenko K. et al. Asynchronous SGD Beats Minibatch SGD Under Arbitrary Delays

---

> > ### Comment · Reviewer_563Q · 2023-08-15
> >
> > * *"Unlike [1], we provide theoretical evidence that asynchronous methods are strictly better."* [[1]]( https://proceedings.neurips.cc/paper_files/paper/2022/file/029df12a9363313c3e41047844ecad94-Supplemental-Conference.pdf ) also provide *theoretical evidences*: this is the bulk of their paper (see for example page 2: *"we prove guarantees for Asynchronous SGD that match the guarantee for Minibatch SGD using exactly M times fewer updates, meaning that our Asynchronous SGD
> > guarantees are strictly better than the Minibatch SGD guarantees in terms of runtime."*) However, in addition to them, they also verify that it leads to faster convergence in *practice*, which is common practice in optimization. Thus, it would strengthen your claims to verify that your theoretical analysis also leads to faster convergence in practice.

---

> > > ### Author Response · Authors · 2023-08-15
> > >
> > > We agree that [1] shows better time complexity guarantees. We emphasize this in Lines 57-63 of our paper. However, [1] compares Asynchronous SGD and Minibatch SGD only. While our paper provides the lower bound for for any method that starts synchronous calculations in Section 8. And we compare the *lower bound* of any such method and the *upper bound* (which simultaneously is the lower bound) of Rennala SGD. In fact, Minibatch SGD is not the best method that starts synchronous calculations. An optimal method is $m$-Minibatch SGD (see details in Section G.1).
> > >
> > > > However, in addition to them, they also verify that it leads to faster convergence in practice, which is common practice in optimization.
> > >
> > > We agree; we will follow your advice, and we will add experiments with Minibatch SGD to Section H. We will also add small-scale experiments with real-life delays on standard machine learning tasks.
> > >
> > > [1]: Mishchenko K. et al. Asynchronous SGD Beats Minibatch SGD Under Arbitrary Delays

---

### Decision · Program_Chairs · 2023-09-21

**Decision:**

Accept (poster)

**Comment:**

This paper introduces novel lower bounds and addresses a crucial topic concerning the optimality of parallel/asynchronous methods. The content of the paper is thoroughly discussed, and although some reviewers noticed a minor flaw related to organization and experiments, it appears that the authors' rationale for introducing context before discussing related work and incorporating additional experiments to substantiate their claims is well-founded. The concerns raised by reviewer DBuG lack specificity, making it challenging to endorse their assessment. Considering these factors collectively and the support of other reviewers toward acceptance, I recommend the acceptance of this paper.